# PAX4 loss of function increases diabetes risk by altering human pancreatic endocrine cell development

Hwee Hui Lau [1,2,16], Nicole A. J. Krentz [3,4,15,16], Fernando Abaitua[4], Marta Perez-Alcantara [4], Jun-Wei Chan[1,2], Jila Ajeian[5], Soumita Ghosh[6], Yunkyeong Lee [3], Jing Yang[3], Swaraj Thaman[3], Benoite Champon[4], Han Sun [3], Alokkumar Jha[3], Shawn Hoon[7], Nguan Soon Tan [2,8], Daphne Su-Lyn Gardner [9], Shih Ling Kao[10,11], E. Shyong Tai[10,11,12], Anna L. Gloyn [3,4,6,13,17] ✉ & Adrian Kee Keong Teo [1,11,14,17] ✉

The coding variant (p.Arg192His) in the transcription factor *PAX4* is associated with an altered risk for type 2 diabetes (T2D) in East Asian populations. In mice, *Pax4* is essential for beta cell formation but its role on human beta cell development and/or function is unknown. Participants carrying the PAX4 p.His192 allele exhibited decreased pancreatic beta cell function compared to homozygotes for the p.192Arg allele in a cross-sectional study in which we carried out an intravenous glucose tolerance test and an oral glucose tolerance test. In a pedigree of a patient with young onset diabetes, several members carry a newly identified p.Tyr186X allele. In the human beta cell model, EndoC-βH1, *PAX4* knockdown led to impaired insulin secretion, reduced total insulin content, and altered hormone gene expression. Deletion of *PAX4* in human induced pluripotent stem cell (hiPSC)-derived islet-like cells resulted in derepression of alpha cell gene expression. In vitro differentiation of hiPSCs carrying *PAX4* p.His192 and p.X186 risk alleles exhibited increased polyhormonal endocrine cell formation and reduced insulin content that can be reversed with gene correction. Together, we demonstrate the role of PAX4 in human endocrine cell development, beta cell function, and its contribution to T2D-risk.

Diabetes is a chronic condition affecting more than 537 million people worldwide, giving rise to devastating complications and healthcare burden on society[1]. In an effort to identify novel disease-causing mechanisms and tractable targets for therapeutic development, numerous genome-wide studies have been performed across different ancestries to identify genetic variants that influence diabetes risk[2–4]. An Asian-enriched *PAX4* p.Arg192His (rs2233580) coding variant has been reproducibly associated with T2D risk [odds ratio of -1.75][2,5]. Additional studies revealed that 21.4% of 2886 people with early-onset T2D carried at least one *PAX4* p.Arg192His allele[6]. Participants carrying the p.His192 allele have a dose-dependent earlier age of T2D-onset[6] and have lower C-peptide levels[7], consistent with pancreatic beta cell dysfunction[8]. Earlier studies have reported other rare coding allele(s) in *PAX4* as a cause of monogenic diabetes[9]. However, the high frequency of the variants in the population and a lack of cosegregation with diabetes has led to discussion over whether they are causal for diabetes and if *PAX4* should be included in diagnostic testing panels for monogenic diabetes[10,11].

PAX4 is a paired-homeodomain transcription factor that has been shown to act as a repressor of insulin and glucagon promoters[12,13]. In

mice, *Pax4* is broadly expressed throughout the developing pancreas[13,14]. *Pax4* homozygous knockout mice die three days post-partum from hyperglycemia, caused by a near complete absence of insulin-producing beta cells[14]. Loss of *Pax4* in mice also leads to an increase in the number of alpha cells and an upregulation of the alpha cell gene *Arx*[14,15]. Conversely, *Arx*[-/-] mice upregulate *Pax4* and have more beta cells, leading to the model of mutual repression of *Pax4* and *Arx* to direct the development of alpha and beta cell lineages, respectively[15]. Similar mutual repression of *pax4* and *arx* has been detected in zebrafish[16]; however, unlike in mice, *Pax4* is not required for beta cell development[16]. In humans, it is currently unknown whether *PAX4* is required for endocrine cell formation. As *PAX4* variants have been reported as a potential cause of monogenic diabetes[9,17,18] and are associated with altered T2D risk, investigating their role in human endocrine cell formation may improve our understanding of the mechanism(s) underlying the genetic association and clarify the potential role of *PAX4* variants as a cause of monogenic diabetes.

Here we present detailed human in vivo and in vitro studies on two different *PAX4* coding alleles, the East Asian population enriched p.Arg192His and a novel protein-truncating variant (PTV) p.Tyr186X identified in a Singapore family with early onset diabetes. We generated three independent human induced pluripotent stem cell (hiPSC) models: (i) *PAX4*-knockout and *PAX4* variant SB Ad3.1 cell lines using CRISPR-Cas9 genome editing; (ii) donor-derived cells with p.Arg192Arg, p.Arg192His, p.His192His and p.Tyr186X genotypes; and (iii) genotype-corrected donor-derived cells and differentiated all lines into pancreatic islet-like cells (SC-islets) using two different protocols. Consistently, we found that *PAX4* deficiency and/or loss-of-function to result in derepression of alpha cell genes, leading to the formation of polyhormonal endocrine cells in vitro, with reduced total insulin content. This phenotype was confirmed independently in the human beta cell line EndoC-βH1 and could be reversed in donor-derived hiPSC lines through correction of diabetes-associated *PAX4* alleles. We conclude that, whilst PAX4 is not essential for in vitro stem cell differentiation to SC-islets, both *PAX4* haploinsufficiency and loss-of-function coding alleles increase the risk of developing diabetes by negatively impacting human beta cell development and insulin secretion. Our observations are consistent with rare *PAX4* alleles resulting in haploinsufficiency being insufficient to cause fully penetrant monogenic diabetes but increasing the risk for T2D.

## Results

### Participants carrying the *PAX4* p.Arg192His T2D-risk allele exhibit decreased beta cell function

We recruited a total of 183 non-diabetic individuals and assessed their pancreatic beta cell function by a frequently sampled intravenous glucose tolerance test (Fig. 1a). Participants carrying the T2D-risk allele (p.His192) had a decreased acute insulin response to glucose (AIRg, padj = 0.036) after adjusting for age, sex and BMI (Fig. 1b). Participants carrying the T2D-risk allele p.His192 were more insulin sensitive (Si, padj = 0.048). There were no differences in disposition index (DI, padj = 0.396) between the two groups (Fig. 1b). HOMA-B, a measurement of beta cell function, was significantly reduced in participants carrying p.His192 allele(s) after adjusting for age, sex and BMI (padj = 0.027). A subset of the recruited individuals [$n = 57$] then underwent an oral glucose tolerance test (OGTT) which revealed higher fasting and 2-h glucose levels, as well as a lower ratio of area under the curve (AUC) for insulin:glucose (Fig. 1c−e). HOMA-B was significantly poorer during the OGTT in participants carrying the p.His192 risk allele(s) (147.2) compared to controls (124.7) (padj = 0.027). There were no differences in fasting, 2-hour, or AUC glucagon (Supplementary Fig. 1a−c). However, there was a significant decrease in the difference between fasting and 2-hour glucagon (delta glucagon), suggesting participants carrying the

p.His192 have less glucagon suppression (Supplementary Fig. 1d). There was no difference in HOMA-IR (Supplementary Fig. 1e). As loss of *Pax4* impacts enteroendocrine cell formation in mice[19], we also measured GLP-1 in participants carrying the p.His192 and found no significant differences in GLP-1 level (Supplementary Fig. 1f−h). Together, the clinical data are consistent with increased T2D-risk via defects in pancreatic beta cell mass and/or function.

### Identification of a novel *PAX4* protein truncating variant p.Tyr186X in a family with early onset diabetes

A female proband (III-1) of Singapore Chinese ethnicity was diagnosed with early-onset diabetes at the age of 10 years (random glucose 306 mg/dL), verified to be GAD antibody negative, and had detectable C-peptide (1.2 nmol/L). Upon diagnosis, she was treated with a basal bolus insulin regimen and metformin for two weeks before being switched to metformin-alone treatment. Following lifestyle modifications, she lost weight and nine months post-diagnosis her HbA1c was 7.1% (mean basal glucose level of 156.6 mg/dL). The early diabetes onset, lack of evidence for type 1 diabetes, and persistence of diabetes despite weight loss prompted further assessment in the family (Fig. 1f). II-11 was diagnosed with gestational diabetes (GDM) at the age of 29 years when she was pregnant with the proband. At age 40 years, while being asymptomatic for diabetes, an OGTT confirmed a diagnosis of diabetes with a fasting glucose of 100.8 mg/dL and a 2-hour glucose of 205.2 mg/dL.

Genetic testing for monogenic diabetes with a custom Illumina Nextera rapid capture next-generation sequencing panel on an Illumina Miseq sequencing platform (*HNF4A, GCK, HNF1A, PDX1, HNF1B, NEUROD1, KLF11, CEL, PAX4, INS, ABCC8, KCNJ11*) was performed on members of the family who were recruited for the clinical study. A novel (not reported in gnomAD or ClinVar, date accessed Feb 2022), heterozygous *PAX4* mutation (c.555_557dup) predicted to result in a truncated protein (p.Tyr186X) was identified in the proband (III-1), the mother (II-11) and a female member of the family (II-7) (Fig. 1f). No rare coding variants were detected in the other genes tested.

The other female heterozygous p.Tyr186X variant carrier (II-7) had a history of GDM (age 26 years) and at the time of study (age 51 years) had impaired glucose tolerance (IGT). Family members (II-4, II-5, II-8, and II-9) with diabetes and another female family member without diabetes (II-3) did not carry the variant. Unfortunately, the proband's grandparents, both diagnosed with diabetes at the age of 40 years, declined to take part in the study.

Given the high prevalence of diabetes in the family and both maternal grandparents having diabetes, we evaluated measures of insulin resistance (HOMA-IR) and beta cell function (DI) in family members with and without the *PAX4* p.Tyr186X variant. Family member II-3, who does not have diabetes and does not carry the p.Tyr186X variant, has the highest beta cell function as measured by the DI whilst those carrying the *PAX4* variant (II-11 and II-7) have markedly reduced function (Fig. 1g). Of note, the family members with diabetes who do not carry the *PAX4* variant all displayed evidence of insulin resistance (HOMA-IR > 2) and low DI, consistent with T2D (Fig. 1g). Taken together, these findings are insufficient to provide support for the p.Tyr186X variant as the cause of monogenic diabetes in this family but are consistent with the *PAX4* variant being associated with decreased pancreatic beta cell function.

To further explore the role of rare coding variants in the *PAX4* gene on T2D risk, we accessed aggregated gene-level exome-sequencing association data from 52 K individuals deposited in the Common Metabolic Disease Portal (https://t2d.hugeamp.org) and in 281,852 individuals from UKBioBank (https://www.ukbiobank.ac.uk/). Burden and Sequence Kernel Association Test (SKAT) analyses[20] computed using a series of genotype filters and masks provided nominal evidence for a gene-level association that is independent of the p.Arg192His variant (Supplementary Table 1a, b).

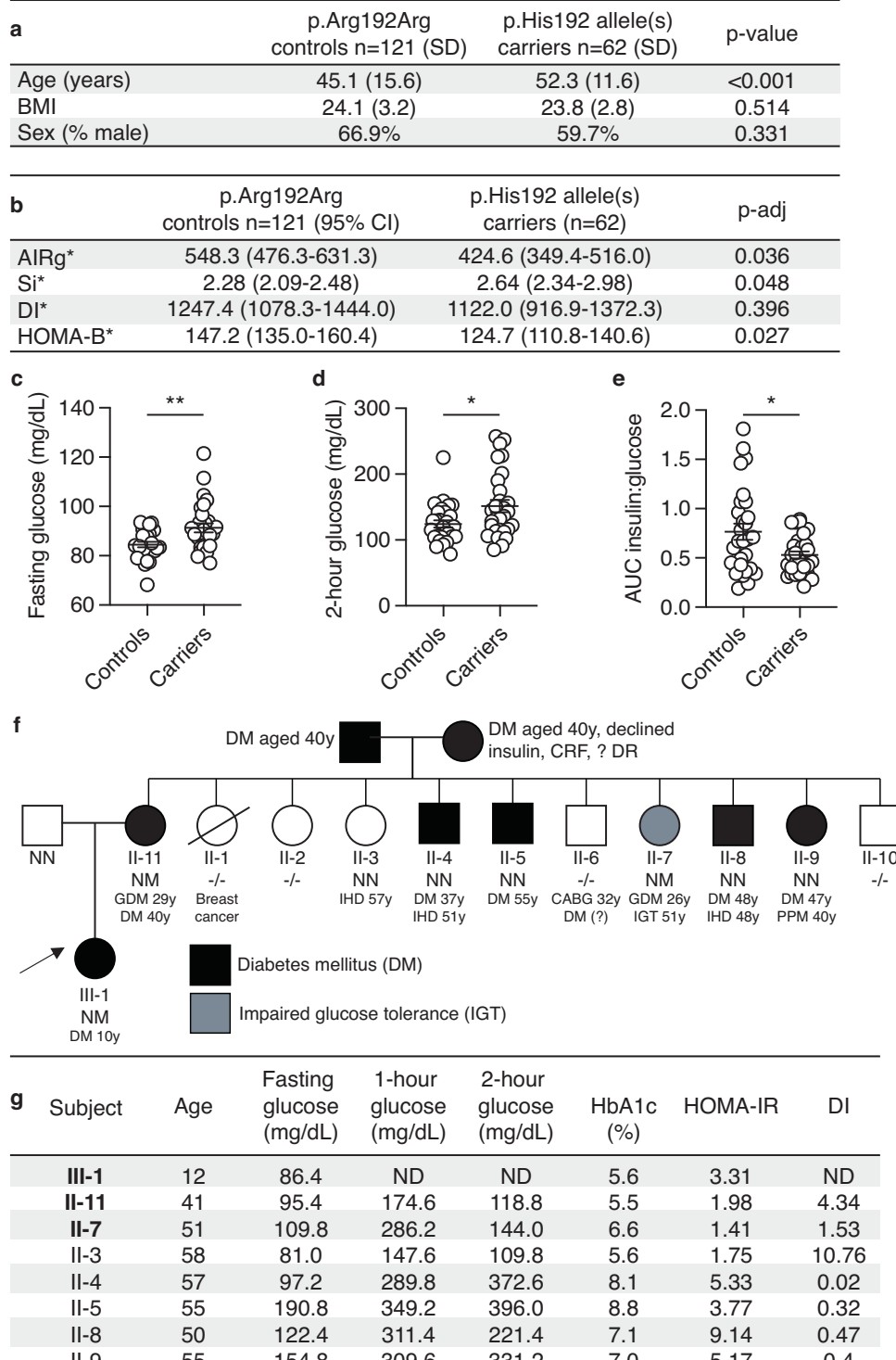

| a | p.Arg192Arg controls n=121 (SD) | p.His192 allele(s) carriers n=62 (SD) | p-value |
|---|---|---|---|
| Age (years) | 45.1 (15.6) | 52.3 (11.6) | <0.001 |
| BMI | 24.1 (3.2) | 23.8 (2.8) | 0.514 |
| Sex (% male) | 66.9% | 59.7% | 0.331 |

| b | p.Arg192Arg controls n=121 (95% CI) | p.His192 allele(s) carriers (n=62) | p-adj |
|---|---|---|---|
| AIRg* | 548.3 (476.3-631.3) | 424.6 (349.4-516.0) | 0.036 |
| Si* | 2.28 (2.09-2.48) | 2.64 (2.34-2.98) | 0.048 |
| DI* | 1247.4 (1078.3-1444.0) | 1122.0 (916.9-1372.3) | 0.396 |
| HOMA-B* | 147.2 (135.0-160.4) | 124.7 (110.8-140.6) | 0.027 |

| g | Subject | Age | Fasting glucose (mg/dL) | 1-hour glucose (mg/dL) | 2-hour glucose (mg/dL) | HbA1c (%) | HOMA-IR | DI |
|---|---|---|---|---|---|---|---|---|
| | **III-1** | 12 | 86.4 | ND | ND | 5.6 | 3.31 | ND |
| | **II-11** | 41 | 95.4 | 174.6 | 118.8 | 5.5 | 1.98 | 4.34 |
| | **II-7** | 51 | 109.8 | 286.2 | 144.0 | 6.6 | 1.41 | 1.53 |
| | II-3 | 58 | 81.0 | 147.6 | 109.8 | 5.6 | 1.75 | 10.76 |
| | II-4 | 57 | 97.2 | 289.8 | 372.6 | 8.1 | 5.33 | 0.02 |
| | II-5 | 55 | 190.8 | 349.2 | 396.0 | 8.8 | 3.77 | 0.32 |
| | II-8 | 50 | 122.4 | 311.4 | 221.4 | 7.1 | 9.14 | 0.47 |
| | II-9 | 55 | 154.8 | 309.6 | 331.2 | 7.0 | 5.17 | 0.4 |

## Loss of *PAX4* alters hormone gene regulation, reduces insulin secretion function and total insulin content in a human beta cell model

To evaluate the consequence of *PAX4* loss on beta cell function, we first performed siRNA- and shRNA-mediated knockdown of *PAX4* in human EndoC-βH1 cells (Fig. 2a). Transient knockdown of *PAX4* using siRNAs significantly reduced *PAX4* transcript expression (Fig. 2b) and glucose-stimulated insulin secretion (GSIS) in EndoC-βH1 cells (Fig. 2c). To model a chronic loss of *PAX4* expression, we generated stable knockdown *PAX4* EndoC-βH1 cells via lentiviral transduction of shRNA followed by antibiotic selection (Fig. 2a).

shPAX4 EndoC-βH1 cells had reduced *PAX4* transcript level compared to shScramble control cells (Fig. 2d). Long-term knockdown of *PAX4* completely abolished GSIS compared to shScramble control cells (Fig. 2e), accompanied by reduced total insulin content in shPAX4 EndoC-βH1 cells (Fig. 2f). While there was no difference in *INS* transcript (Fig. 2g), loss of *PAX4* increased *GCG* transcript by 2.7-fold in shPAX4 EndoC-βH1 cells (Fig. 2h), consistent with PAX4 being a repressor of *GCG* expression[13,21]. In agreement with *ARX* having a role in regulating beta cell development in humans[22], loss of *PAX4* led to a decrease in *ARX* expression in EndoC-βH1 cells (Fig. 2i).

**Fig. 1 | Reduced pancreatic beta cell function in participants carrying diabetes-associated *PAX4* variants. a, b** Mean age and BMI of heterozygous and homozygous carriers of p.His192 *PAX4* allele (*n* = 62) and homozygous p.Arg192Arg controls (*n* = 121) who underwent frequently sampled intravenous glucose tolerance tests to measure Acute Insulin Response to glucose (AIRg), insulin sensitivity (Si), disposition index (DI), and HOMA-B. Unadjusted p-value or adjusted *p*-value (adjusted for age, sex, and BMI) are indicated in the table for AIRg, Si, DI, and HOMA-B. *AIRg, Si, Disposition index and HOMA-B values were obtained by back-transforming Log10 values of the model-estimated means for controls and carriers. **c, d** Plasma glucose level (mg/dL) at the (**c**) fasting and (**d**) 2-hour time points during the oral glucose tolerance test (OGTT) of heterozygous or homozygous p.Arg192His carriers (*n* = 29) and p.Arg192Arg controls (*n* = 28). **e** Ratio of area under the curve (AUC) insulin to glucose during the 2-hour oral glucose tolerance

test. **f** Family pedigree of a Singaporean family with a novel p.Tyr186X *PAX4* variant. NN, wild-type; NM, heterozygotes; −/−, genotype not accessible. An arrow indicates the proband (III-1). Age of diagnosis for diabetes mellitus (DM), gestational diabetes mellitus (GDM), ischemic heart disease (IHD), coronary artery bypass grafting (CABG), impaired glucose tolerance (IGT), permanent pace-maker implantation (PPM), chronic renal failure (CRF), and diabetic retinopathy (DR). **g** Summary of measures of beta cell function between family members in (**f**) during a 2-h 75 g glucose oral glucose tolerance test. Carriers of the p.X186 allele are in bold. HbA1c, hemoglobin A1c; HOMA-IR, homeostatic model assessment of insulin resistance (value > 2 indicates insulin resistance); DI, disposition index (Matsuda); ND, not done. Data are presented as mean ± SEM in (**b–d**). Statistical analyses were performed using two-tailed unpaired *t* test. *$p < 0.05$, **$p < 0.01$. Source data is provided in the Source Data File.

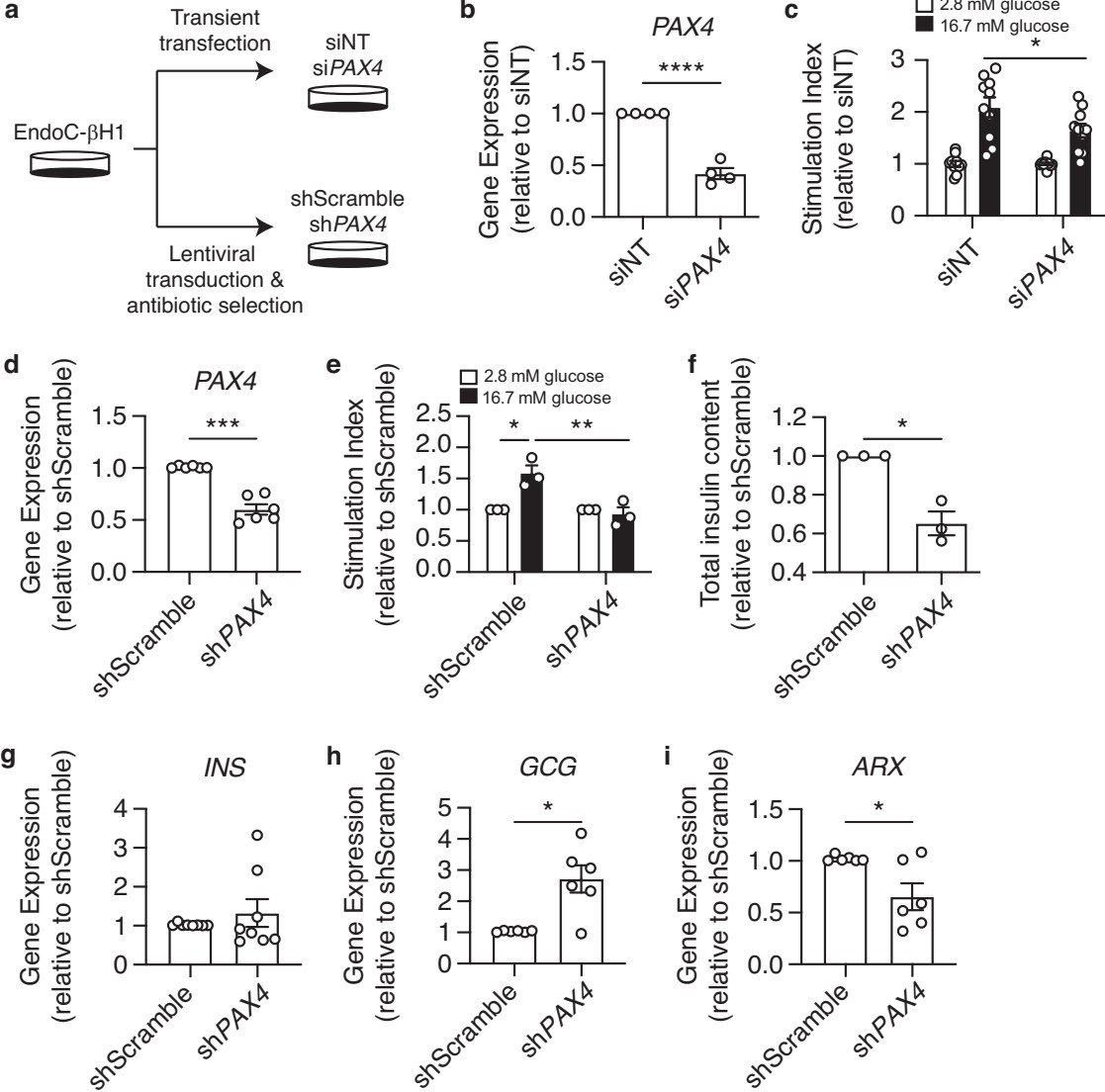

**Fig. 2 | *PAX4* knockdown and knockout impair glucose-stimulated insulin secretion and reduce insulin content in human EndoC-βH1 cells. a** Experimental design for *PAX4* knockdown approaches using siRNA and shRNA in EndoC-βH1 cells. **b** *PAX4* gene expression following transient transfection of si*PAX4* and non-targeting (siNT) control in EndoC-βH1 cells. **c** Glucose-stimulated insulin secretion of 2.8 mM and 16.7 mM glucose in siNT and si*PAX4* EndoC-βH1 cells, normalized to total protein then to 2.8 mM glucose. **d** *PAX4* gene expression in *PAX4*-knockdown (sh*PAX4*) and control (shScramble) EndoC-βH1 cells following six passages of antibiotic selection (*n* = 6). **e** Glucose-stimulated insulin secretion assay comparing sh*PAX4* and shScramble EndoC-βH1 cells, normalized to total DNA and then to

2.8 mM glucose (*n* = 3). **f** Relative fold change of total insulin content in sh*PAX4* and shScramble EndoC-βH1 cells, normalized to total DNA content (*n* = 3). **g** *INS* transcript expression in shScramble and sh*PAX4* EndoC-βH1 cells (*n* = 8). **h** *GCG* transcript expression in shScramble and sh*PAX4* EndoC-βH1 cells (*n* = 6). **i** *ARX* transcript expression in shScramble and sh*PAX4* EndoC-βH1 cells (*n* = 6). Data are presented as mean ± SEM. Each data point represents one independent experiment. Statistical analysis of two samples was performed by paired *t* test or a two-way ANOVA for comparison of multiple groups. *$p < 0.05$, ***$p < 0.001$, ****$p < 0.0001$. Source data is provided in the Source Data File.

## PAX4 knockout in hiPSC-derived islet cells causes derepression of alpha cell gene expression

While PAX4 transcript and protein can be detected in rat[23] and human islets[24], its expression is most abundant during embryonic development[13,14], suggesting that PAX4 variants may mediate disease risk early on during embryonic development. Homozygous Pax4 knockout mice die within three days of birth and have a near complete loss of pancreatic beta cells[14]. Whether PAX4 is similarly required for the formation of human beta cells is unknown. We generated PAX4 homozygous null hiPSC lines (PAX4^KO/KO) using CRISPR-Cas9 (Fig. 3a) and two single guide RNAs (sgRNAs) designed for exons 2 and 5 (encoding the paired-domain and homeodomain) of the PAX4 gene (Supplementary Fig. 2a). Using Sanger sequencing, we confirmed that two of the independent cell lines had a homozygous deletion for amino acids 64 through 200, whilst the other cell line was compound heterozygous for two premature stop codons at amino acids 61 and 74, respectively (Fig. 3b). Three independent, unedited hiPSC lines generated during the CRISPR-Cas9 process provided control PAX4 wildtype (PAX4^WT/WT) lines (Fig. 3a). All six hiPSC lines were differentiated towards SC-islets using a seven-stage protocol[25] (Fig. 3c; Protocol A). Flow cytometry analysis of CXCR4+ and SOX17+ definitive endoderm (DE) cells determined that there was no defect in the formation of DE (Supplementary Fig. 2b, c). As PAX4 is a transcription factor, RNA-seq analysis was used to determine the transcriptional consequence of PAX4 knockout. RNA-seq samples (n = 8 per genotype) were collected at DE (before PAX4 expression), pancreatic endoderm (PE), endocrine progenitor (EP) (at the peak of PAX4 expression) and SC-islets (Fig. 3d and Supplementary Data 1). The PAX4 transcript was significantly reduced in PAX4^KO/KO PE (padj = 4.25E−05), EP (padj = 6.15E−05) and SC-islets (padj = 1.27E−06) (Fig. 3d), and the remaining transcripts were missing exons 2 through 5 (Supplementary Fig. 2d).

Differential expression analysis using DESeq2 showed that the loss of PAX4 resulted in a derepression of an alpha cell gene signature (ARX, GCG, TTR) and a repression of the endocrine progenitor marker FEV[26,27] in SC-islets (Fig. 3e and Supplementary Fig. 2e–g). To confirm our results, we differentiated the same PAX4^WT/WT and PAX4^KO/KO hiPSC lines into SC-islets using a second protocol (Fig. 3c; Protocol B)[28]. Using Protocol B, we found a significant reduction in PAX4 transcript at the EP stage (Fig. 3f) and a larger number of differentially expressed genes (Fig. 3g and Supplementary Data 2). Gene ontology (GO) biological process analysis of the differentially expressed genes in PAX4^KO/KO SC-islets revealed a number of GO terms that included the alpha cell gene ARX (Fig. 3h).

Using a curated list of genes involved in beta and alpha cell lineages[29], we observed directionally consistent, albeit non-significant, derepression of alpha cell genes (ARX, GCG, TTR) (Fig. 3i). The expression of delta cell gene (SST), epsilon cell gene (GHRL) and PP cell gene (PPY) was also derepressed in SC-islets derived from PAX4^KO/KO lines (Fig. 3i). Some genes that are involved in beta cell maturation and hormone secretion (MAFA, ISL1, GRHL3, SLC17A6, PCSK2, EYA2) were downregulated with the loss of PAX4 (Fig. 3i). Importantly, PAX4^WT/WT and PAX4^KO/KO lines differentiating into SC-islets repress pluripotency genes and activate genes involved in endocrine cell fate in a similar manner (Supplementary Fig. 3), giving rise to similar proportions of DE, EP, and SC-islets (Supplementary Fig. 4a–c). These observations suggest that, unlike in mouse, PAX4 is not essential for human beta cell differentiation in vitro and its loss did not detrimentally impact the differentiation trajectory of hiPSCs towards pancreatic endocrine cells. Rather, PAX4 loss-of-function results in a dysregulation of key endocrine maturation genes in hiPSC-derived SC-islets.

## Donor-derived hiPSCs carrying the PAX4 p.Arg192His and p.Tyr186X alleles have defects in endocrine cell differentiation in vitro

Having established the effect of PAX4 loss during in vitro beta cell differentiation, we next generated donor-derived hiPSCs from participants carrying PAX4 variant and differentiated them into SC-islets. Skin biopsies and/or blood samples from recruited donors without diabetes were used to derive hiPSCs of the following genotypes: homozygous for the PAX4 p.Arg192 and p.Tyr186 alleles (wildtype), heterozygous for either the p.Arg192His or p.Tyr186X alleles, and homozygous for the p.His192 allele (p.His192His) (Fig. 4a). To account for possible line-to-line heterogeneity in hiPSC-based studies[30], three independent hiPSC lines were generated from two donors for wildtype cells, four lines from two donors for p.Arg192His, five lines from two donors for p.His192His, and three lines from one donor (II-7; Fig. 1e) for the p.Tyr186X variant (Fig. 4a). All hiPSC lines were characterized via pluripotency immunostaining, teratoma assay, karyotyping, and genotypes were confirmed by Sanger sequencing.

We simultaneously differentiated all 15 hiPSC lines into pancreatic SC-islets using Protocol B and performed qPCR, flow cytometry and immunostaining analyses. PAX4 transcript expression was unchanged in carriers of the PAX4 p.His192 allele but elevated in EPs derived from the p.Tyr186X carrier (Fig. 4b), consistent with transcriptional compensation for the PTV. Heterozygous and homozygous carriers of the PAX4 p.His192 allele had no measurable differences in INS, GCG, or SST gene expression at the EP or SC-islet stages (Fig. 4b–i). Similar to the PAX4^KO/KO hiPSCs (Fig. 2h), the PAX4 p.Tyr186X hiPSC lines exhibited a derepression of the GCG gene in both EPs and SC-islets (Fig. 4d, h), suggesting that the PTV is loss-of-function. Flow cytometry analyses of EPs and SC-islets found no significant differences in INS+, PDX1 + GCG+ and SST+ protein expression (Supplementary Fig. 5a–d). On the other hand, while immunostaining of endocrine hormones found no significant differences in GCG+ or INS+ cells (Fig. 4j-l), there was a significant increase in the number of polyhormonal (C-PEP+/GCG+) SC-islets from PAX4 variant hiPSC lines (Fig. 4j, m). SC-islets from the PAX4 p.Arg192His and p.Tyr186X hiPSCs had lowered total insulin content (Fig. 4n), which is consistent with the in vivo clinical data from participants carrying PAX4 variants (Fig. 1). Together, these data suggest that both PAX4 alleles result in a loss-of-function due to reduced PAX4 gene dosage and/or altered PAX4 transcriptional activity, negatively affecting pancreatic beta cell differentiation.

## PAX4 p.Arg192His and p.Tyr186X alleles reduce the expression and/or function of PAX4 protein

The PAX4 protein consists of two functional domains, paired and homeodomain, that are responsible for DNA binding, and two nuclear localization sequences (NLS) (Fig. 5a)[31,32]. Both p.Arg192His and p.Tyr186X variants are located within the functional homeodomain of the PAX4 protein (Fig. 5a). As the crystal structure of PAX4 protein has not been elucidated, we obtained the predicted three-dimensional molecular arrangement of PAX4 protein (AF-O43316-F1-model_v2) from the AlphaFold database[33,34]. The p.Arg192 residue is located within a hydrophobic pocket (Supplementary Fig. 6a), suggesting that substitution to an uncharged histidine may alter the DNA-binding function of the PAX4 protein. The PAX4 p.Tyr186X (c.557−559 GTA duplication) variant causes a frameshift, leading to the introduction of a premature stop codon at amino acid position 186 (Fig. 5a and Supplementary Fig. 6a). Transcripts containing PTVs, such as PAX4 p.Tyr186X, may undergo nonsense mediated decay (NMD), resulting in haploinsufficiency[35]. To test whether the PAX4 variants undergo NMD, we performed allelic-specific qPCR following treatment with the NMD inhibitor cycloheximide (CHX)[36,37]. Treatment of hiPSC-derived SC-islets with CHX overnight stabilized the p.X186 allele (Fig. 5b) but had no effect on p.His192 (Fig. 5c), confirming NMD of p.X186 and

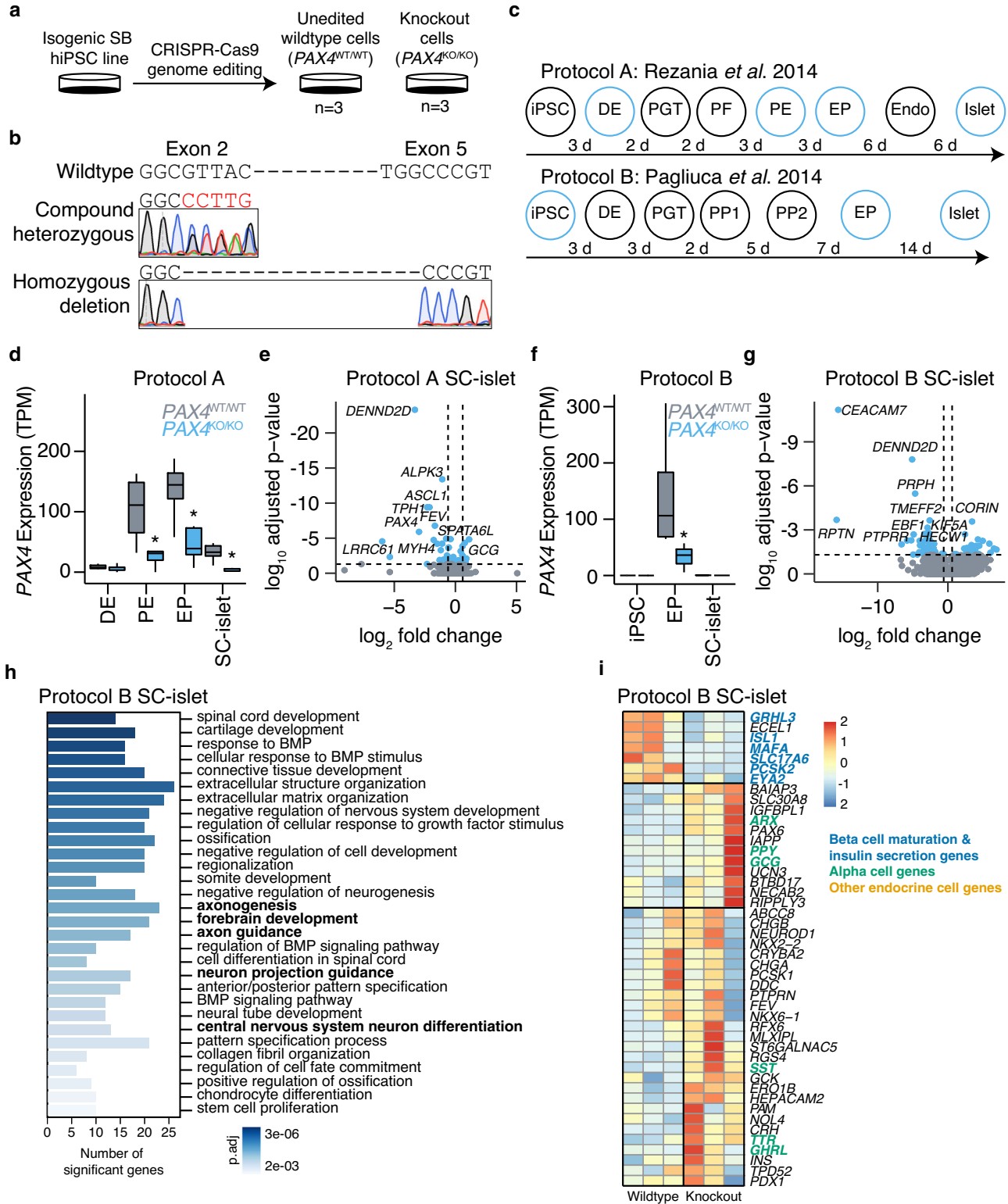

supporting *PAX4* haploinsufficiency as the mechanism for the p.Tyr186X variant.

To understand the consequence of *PAX4* variants on protein function, we performed a series of in vitro assays using overexpression of tagged WT and mutant (p.His192 and p.X186) PAX4 protein (Supplementary Fig. 6b). Western blot analyses demonstrated successful PAX4 overexpression detected by PAX4 antibody[38] and V5 tag expression (Supplementary Fig. 6c). Overexpression of the *PAX4* variants in AD293 cells confirmed that the p.His192 allele does not prevent

nuclear localization and that any p.X186 protein that escaped NMD remained trapped in the cytoplasm due to the loss of downstream NLS (Supplementary Fig. 6d). We observed fewer PAX4 antibody-positive cells in p.X186 transfected AD293 cells, despite no difference in overall transfection efficiency (% GFP+), consistent with decreased stability of any truncated protein produced by the PTV (Supplementary Fig. 6e). Treating AD293 cells overexpressing the *PAX4* constructs with the proteasomal inhibitor MG132 revealed an accumulation of the p.X186 protein compared to wildtype or p.His192, demonstrating that the

**Fig. 3 | *PAX4* knockout human induced pluripotent stem cell lines have defects in endocrine cell formation during in vitro differentiation. a** CRISPR-Cas9 genome editing was used to generate three independent hiPSC *PAX4* homozygous knockout cell lines (*PAX4*^KO/KO) and three unedited wildtype control cell lines (*PAX4*^WT/WT). **b** Sanger sequencing for exon 2 and exon 5 of compound heterozygous P*AX4*^KO/KO and homozygous deletion *PAX4*^KO/KO hiPSC lines. **c** Schematic outline of seven-stage protocol A and six-stage protocol B from human induced pluripotent stem cells (hiPSC), definitive endoderm (DE), primitive gut tube (PGT), posterior foregut (PF) or pancreatic progenitor 1 (PP1), pancreatic endoderm (PE) or pancreatic progenitor 2 (PP2), endocrine progenitor (EP), endocrine (Endo), towards islet-like cells (SC-islets). RNA-seq samples were collected at the end of stages highlighted in blue. **d** Expression of *PAX4* in transcripts per million

(TPM) in *PAX4*^WT/WT and *PAX4*^KO/KO cells at DE, PE, EP and SC-islets derived from Protocol A. **e** Volcano plot of differentially expressed genes in *PAX4*^KO/KO versus *PAX4*^WT/WT SC-islets derived from Protocol A. The top ten differentially expressed genes are highlighted. **f** Expression of *PAX4* in TPM in *PAX4*^WT/WT and *PAX4*^KO/KO cells at hiPSCs, EPs and SC-islets derived from Protocol B. **g** Volcano plot of differentially expressed genes in *PAX4*^KO/KO versus *PAX4*^WT/WT SC-islets derived from Protocol B. The top ten differentially expressed genes are highlighted. **h** Gene Ontology (GO) analysis of differentially expressed genes in *PAX4*^KO/KO compared to *PAX4*^WT/WT SC-islets from Protocol B. Bolded are the terms that include *ARX*. **i** Heatmap of relative gene expression for pancreatic endocrine genes in SC-islets derived from Protocol B.

overexpressed truncated protein is subject to proteasomal degradation (Supplementary Fig. 6f, g).

It was previously reported that *PAX4* p.His192 results in defective transcriptional repression of human *INS* and *GCG* gene promoters[13]. In EndoC-βH1 cells, overexpression of both WT and p.His192 PAX4 proteins resulted in significant repression of *INS* promoter activity (Fig. 5d). Although the p.X186 variant most likely results in NMD and haploinsufficiency, any translated protein was unable to repress *INS* promoter activity (Fig. 5d). WT PAX4 protein did not repress the *GCG* gene promoter in EndoC-βH1 cells (Fig. 5e) but did so in the rodent alpha cell model αTC1.9 (Fig. 5f), consistent with cell-type specific regulation of gene expression by PAX4. Both *PAX4* p.His192 and p.X186 resulted in a derepression of the *GCG* promoter in beta cells (Fig. 5e) and a loss of repression activity in alpha cells (Fig. 5f). Luciferase assays for *INS* gene promoter activity were unchanged in EndoC-βH1 cells following *PAX4* (sh*PAX4*) knockdown (Fig. 5g). However, sh*PAX4* EndoC-βH1 cells had significantly increased *GCG* promoter activity (Fig. 5h), consistent with the loss-of-repression of *GCG* promoter activity observed in the presence of p.His192 or p.X186. Taken together, our studies demonstrate that *PAX4* p.Arg192His and p.Tyr186X variant proteins have altered expression and/or transcriptional activity.

### hiPSC-derived EPs have a distinct metabolic gene signature and exhibit a bioenergetics switch from glycolysis to oxidative phosphorylation

To evaluate the overall impact of diabetes-associated *PAX4* gene variants on the global transcriptome of human pancreatic cells, RNA-seq was performed on *PAX4* variant carrier donor-derived hiPSCs across four differentiation time points using Protocol B: hiPSCs, PP2 cells, EPs and SC-islets (Supplementary Data 3). Uniform Manifold Approximation and Projection (UMAP) analyses of a total of 153 RNA samples demonstrated that samples were clustered based on differentiation day (i.e., largest source of variation is developmental time point) (Fig. 6a), indicating that the differentiation protocol is robust in directing the hiPSCs toward SC-islets. Volcano plots comparing the *PAX4* p.Arg192His, p.His192His or p.Tyr186X against wildtype *PAX4* donor-derived hiPSCs demonstrated that most differentially expressed genes were upregulated at the EP stage (Fig. 6b), coinciding with the peak of *PAX4* expression (Fig. 3d, f). Principal Component Analysis (PCA) revealed that EPs derived from *PAX4* variants clustered more closely to each other than to wildtype *PAX4* (Fig. 6c), suggesting that the two *PAX4* variants shared transcriptional similarity. Gene enrichment analyses of relevant biological processes of the differentially expressed genes in the *PAX4* p.His192His and p.Tyr186X EPs revealed an association with metabolic processes and cellular response to stress (Fig. 6d). Between the *PAX4* p.His192His and p.Tyr186X genotypes, there were 2012 and 452 genes in common within the "metabolic processes" and "cellular response to stress", respectively, most of which were elevated in expression in the *PAX4* variant lines (Supplementary Fig. 7a, b).

To further assess a potential defect in metabolism, we performed a Seahorse XFe96 Glycolysis Stress Test. The glycolysis stress test showed that EPs carrying one or two p.His192 risk alleles had lower measurements of glycolytic function, including non-glycolytic acidification, glycolytic capacity, and glycolytic reserve (Fig. 6e, f). In addition, EPs carrying the p.Tyr186X risk allele had decreased glycolytic reserve and non-glycolytic acidification (Fig. 6f). Next, we hypothesized that EPs would seek alternative metabolic processes to compensate for the reduction in energy production, such as oxidative phosphorylation through mitochondrial respiration. Mitochondrial function was measured via oxygen consumption in EPs using the Seahorse XFe96 analyzer. There was an increase in oxidative phosphorylation activity in EPs harboring *PAX4* variants (Fig. 6g), including basal respiration, non-mitochondrial $O_2$ consumption, ATP production, and $H^+$ (proton) leak (Fig. 6h). Overall, EPs carrying *PAX4* diabetes risk alleles demonstrated a bioenergetic switch from glycolysis to oxidative phosphorylation.

To investigate if the altered metabolic gene expression and bioenergetics profile contributed to beta cell maturation from the EP stage, we treated differentiating cells with the antioxidant N-acetylcysteine (NAC)[39] from EP (when *PAX4* expression peaks) to SC-islet stage and extracted total insulin for assessment. *PAX4* p.His192His and p.Tyr186X carrying SC-islets revealed only a modest upregulation in total insulin content (Supplementary Fig. 8a, b), suggesting that the alleviation of oxidative stress is insufficient to rescue the total insulin content in SC-islets. We postulate that the metabolic signature observed in our donor-derived hiPSC model reflects the physiological status of the EPs rather than being the immediate cause for the dysregulation of beta cell development and maturation.

### Correction of *PAX4* risk alleles in donor-derived hiPSCs with CRISPR-Cas9 rescues dysregulated endocrine gene expression and metabolic phenotypes

Next, we used CRISPR-Cas9 to correct the donor-derived hiPSC lines and to generate engineered *PAX4* variant hiPSC lines. We designed sgRNA#3 to target the donor-derived homozygous p.His192His line and provided the homology-directed repair (HDR) template to correct the rs2233580 T2D-risk allele (Fig. 7a). The II-11 donor-derived hiPSC line that is heterozygous for a GTA duplication was corrected with sgRNA#4 and an HDR template (Fig. 7b). From the CRISPR-Cas9 genome editing pipeline, we generated two corrected p.Arg192Arg non-risk and two uncorrected p.His192His hiPSC lines (Fig. 7c). From the II-11 donor-derived line, four corrected p.Tyr186Tyr and four uncorrected p.Tyr186X hiPSC lines were derived (Fig. 7c). To control for the different genetic backgrounds of donors, we genome engineered SB Ad3.1 hiPSCs that are homozygous for the risk alleles (p.His192His and p.X186X). All the corrected and uncorrected donor derived hiPSC lines and the engineered *PAX4* variant hiPSC lines were differentiated towards SC-islets using Protocol B (Fig. 7c, Supplementary Data 4 and 5). Correction of the p. His192 allele but not the p.Tyr186X allele

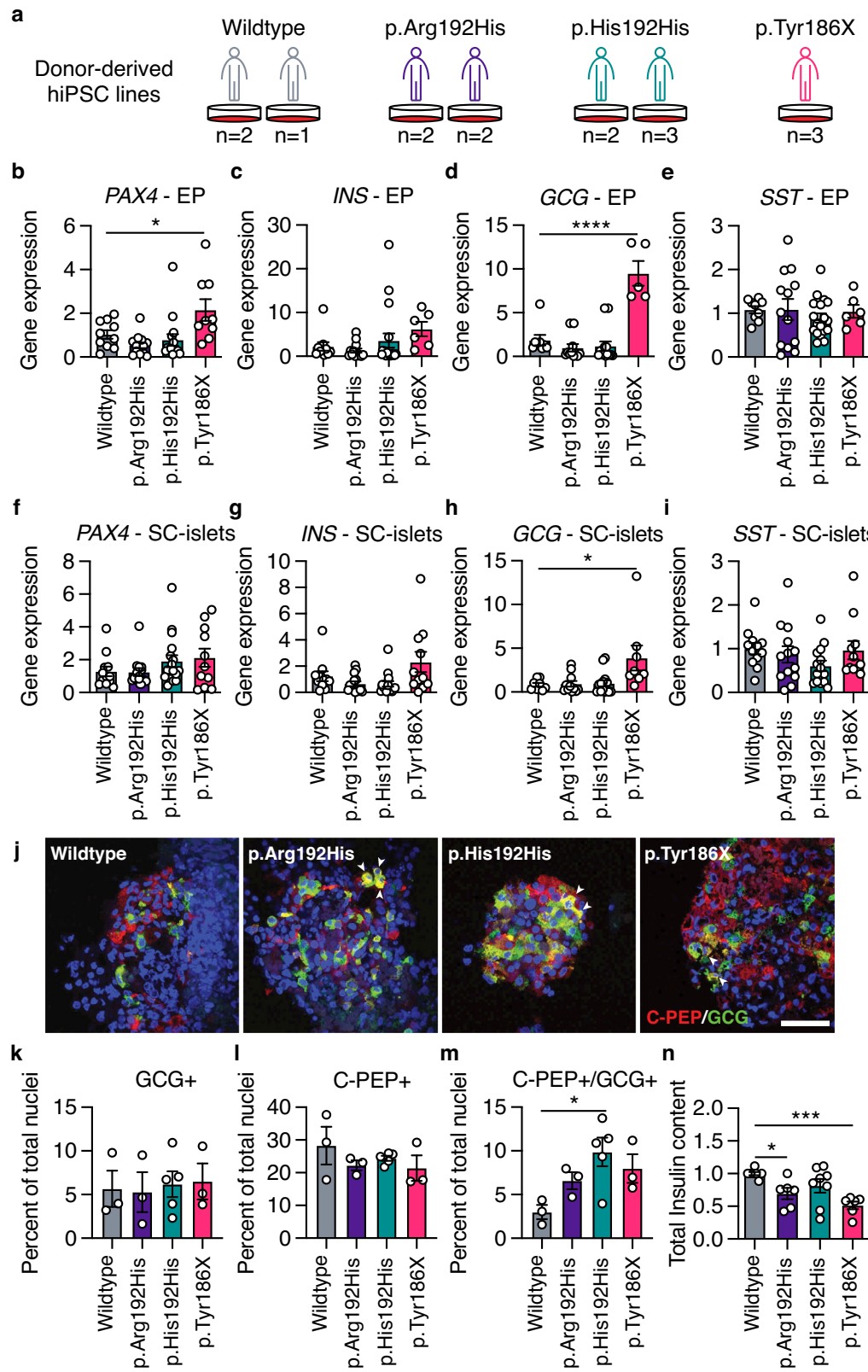

rescued glycolytic reserve (Supplementary Fig. 9). Importantly, correcting the *PAX4* p.His192His and p.Tyr186X mutations decreased the formation of polyhormonal (C-PEP⁺/GCG⁺) cells (Supplementary Fig. 10) and significantly increased the total insulin content of the SC-islets (Fig. 7d, e), indicating that the *PAX4* variants were a direct cause of reduced insulin content in the donor-derived SC-islets.

## Discussion

Rodent models have demonstrated that *Pax4* plays an important role in beta cell specification during early pancreas development[14,40]. However, differences between rodent and human islets in architecture[41] and gene expression[42] make it challenging to extrapolate data based on rodent studies directly to humans. For instance, heterozygous *Pax4* knockout mice do not develop diabetes[14] but *PAX4*

**Fig. 4 | PAX4 p.Arg192His and p.Tyr186X donor-derived hiPSCs have perturbations in differentiation towards SC-islets. a** Donor-derived hiPSC lines were generated from the following genotypes: three lines from two wildtype donors; four lines from two p.Arg192His donors; five lines from two p.His192His donors; and three lines from one p.Tyr186X donor. **b–e** Transcript expression of (**b**) *PAX4*, (**c**) *INS*, (**d**) *GCG* and (**e**) *SST* in hiPSC-derived endocrine progenitor (EP) cells using differentiation Protocol B. **f–i** Transcript expression of (**f**) *PAX4*, (**g**) *INS*, (**h**) *GCG* and (**i**) *SST* in hiPSC-derived islet-like cells (SC-islets) using differentiation Protocol B. **j** Representative immunofluorescence images of hiPSC-derived islet-like cells with C-peptide in red, glucagon in green, and nuclei in blue. Arrows indicate C-PEP + /

GCG+ double-positive cells. Scale bar: 50 μm. **k–m** Quantification of immunofluorescence images for the percentage of cells expressing (**k**) GCG (monohormonal), (**l**) C-PEP (monohormonal) or (**m**) C-PEP +/GCG+ (polyhormonal). **n** Total insulin content normalized to total DNA from SC-islets. Data are presented as mean ± SEM. $n = 4$ independent differentiation experiments were performed in (**b–i**). $n = 3$ independent differentiation experiments were performed in (**k–n**). Statistical analyses were performed using one-way ANOVA. *$p < 0.05$, ***$p < 0.001$, and ****$p < 0.0001$. Differentiation protocol B was used to derive data in Fig. 4. Source data is provided in the Source Data File.

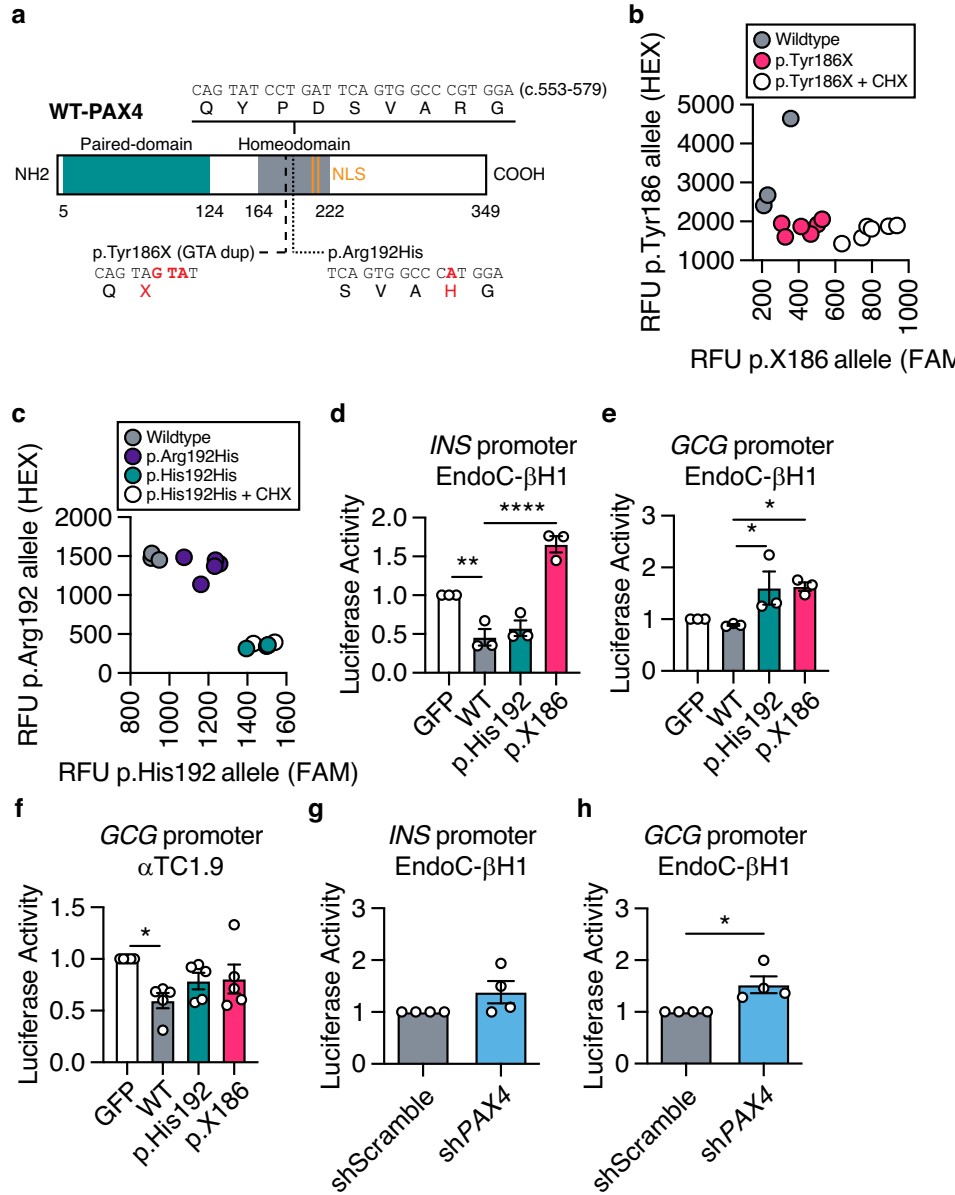

**Fig. 5 | Molecular function of PAX4 variants in human EndoC-βH1 cells. a** Illustration of human full-length wildtype (WT) PAX4 protein. Functional domains are depicted in green (paired-domain) and gray (homeodomain). Nuclear localization sequences (NLS; orange) are located at amino acids 206–210 and 212–216. Part of the homeodomain sequence that contains the *PAX4* variants (c.553–579) and the amino acid changes of p.Tyr186X and p.Arg192His are highlighted. **b** Allele-specific qPCR of *PAX4* transcript for p.Tyr186 and p.X186 alleles following cycloheximide (CHX) treatment. **c** Allele-specific qPCR of *PAX4* transcript for p.Arg192

and p.His192 alleles following CHX treatment. **d, e** Luciferase activity of (**d**) *INS* and (**e**) *GCG* gene promoters in EndoC-βH1 cells overexpressing WT-PAX4, p.His192 and p.X186 ($n = 3$). **f** Luciferase activity of the *GCG* gene promoter in αTC1.9 cells overexpressing WT-PAX4, p.His192 and p.X186 ($n = 5$). **g, h** Luciferase activity of (**g**) *INS* and (**h**) *GCG* gene promoters in shScramble and sh*PAX4* EndoC-βH1 cells ($n = 4$). Data are presented as mean ± SEM. Statistical analyses were performed by $t$ test or two-way ANOVA. $n > 3$. *$p < 0.05$, **$p < 0.01$, ****$p < 0.0001$. Source data is provided in the Source Data File.

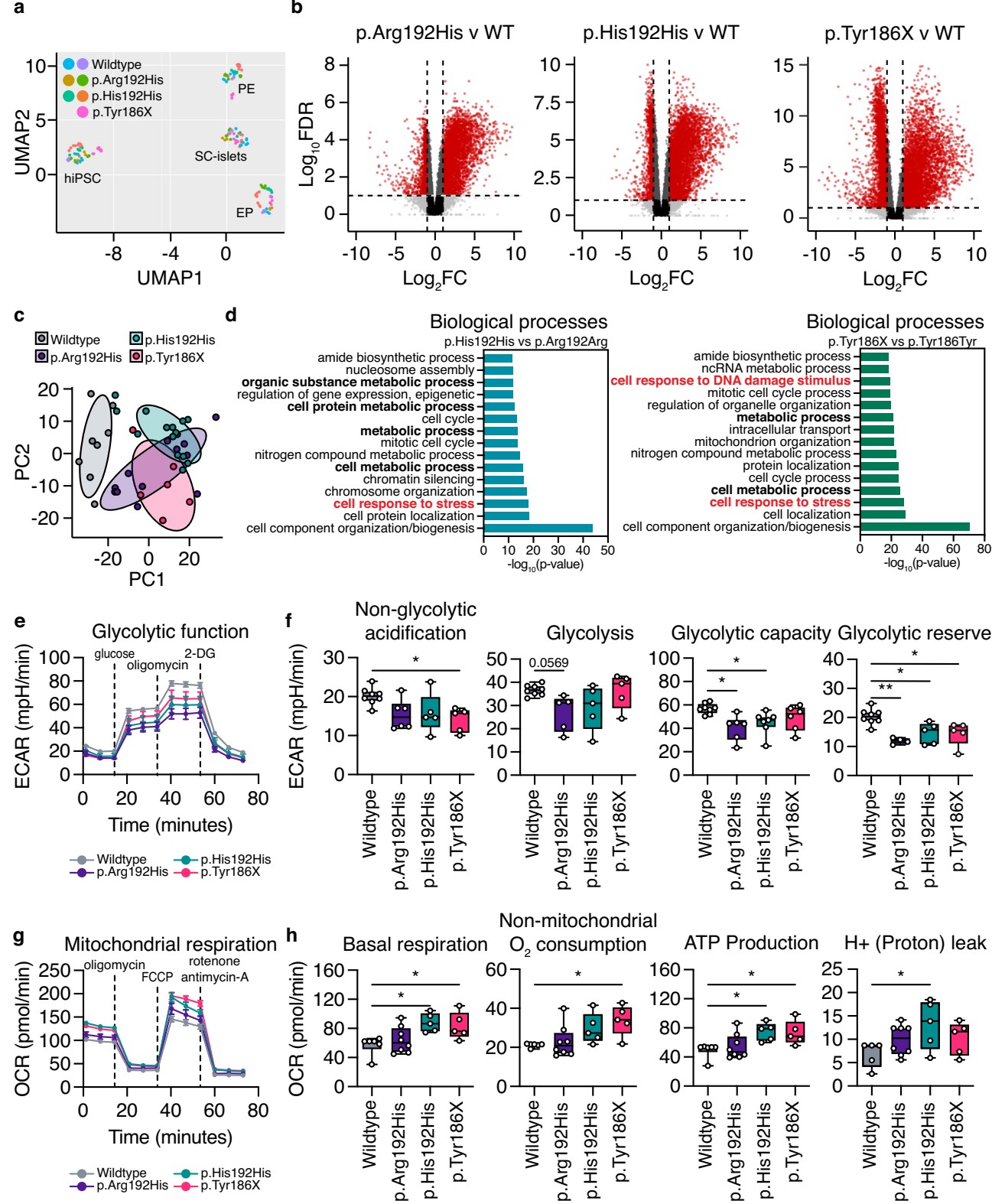

variants causing altered transcriptional activity are strongly associated with increased diabetes risk in humans[9,17,43,44]. These observations suggest that human beta cells could be more sensitive to changes in *PAX4* gene dosage.

While *PAX4* p.Arg192His has been identified as one of the most reproducible variants uniquely associated with East Asian T2D, the role of *PAX4* or its variant p.Arg192His in human beta cell development has not been addressed. Our study capitalized on access to East Asian

carriers of T2D *PAX4* risk alleles to study their effect on human beta cell function in vivo and used donor-derived hiPSCs as a versatile platform to interrogate the role of *PAX4* during human pancreas development in vitro[45–48]. Our clinical phenotyping of the participants carrying *PAX4* p.Arg192His allele(s) demonstrated decreased pancreatic beta cell function based on AIRg, HOMA-B, and lowered DI despite the donors being insulin sensitive based on HOMA-IR measures. Whilst our investigation of the impact of a novel variant predicted to result in a

**Fig. 6 | Metabolic seahorse assays revealed alterations in glycolysis function and oxidative phosphorylation in the presence of the *PAX4* p.Arg192His or p.Tyr186X variants. a** Uniform Manifold Approximation and Projection (UMAP) of 153 RNA samples at the hiPSC, PE, EP and SC-islet stages of in vitro differentiation using Protocol B. **b** Volcano plots [Log$_{10}$FDR and log$_2$(FC)] demonstrating pairwise comparisons of p.Arg192His, p.His192His, and p.Tyr186X against wildtype, respectively. Red circles represent transcripts with log2FC <−2 or >2 and *p* < 0.05. **c** Principal component analysis (PCA) of RNA-seq data for *PAX4* donor hiPSC-derived EP cells. PC1: 35%; PC2: 11%. (**d**) Gene ontology (GO) analysis of differentially expressed genes in EP comparing p.His192His against p.Arg192Arg (*PAX4*^WT/WT^) or p.Tyr186X against p.Tyr186Tyr (*PAX4*^WT/WT^), FC < 0.67 or FC > 1.5. The bars denote −log10(*p*-value) with FDR < 0.05. **e** Extracellular acidification rate (ECAR) measurements of wildtype, p.Arg192His, p.His192His, and p.Tyr186X EP cells following a sequential addition of glucose, oligomycin, and 2-deoxyglucose (2-DG). **f** Non-

glycolytic acidification, glycolysis, glycolytic capacity, and glycolytic reserve measurements during the ECAR of wildtype, p.Arg192His, p.His192His, and p.Tyr186X EP cells. **g** Oxygen consumption rate (OCR) measurements of wildtype, p.Arg192His, p.His192His, and p.Tyr186X EP cells following a sequential addition of oligomycin, FCCP, rotenone and antimycin-A. **h** Basal respiration, non-mitochondrial O$_2$ respiration, ATP production, and H+ (proton) leak of wildtype, p.Arg192His, p.His192His, and p.Tyr186X EP cells. Box and whisker plots illustrating median (center), quartiles (25th and 75th percentile), maximum and minimum of all data points. *n* = 3 differentiation experiments were performed for glycolysis stress test seahorse assays. *n* = 4 differentiation experiments were performed for mito stress test seahorse assay. Statistical analyses were performed by one-way ANOVA. **p* < 0.05, ***p* < 0.01. Differentiation protocol B was used to derive data in Fig. 6. Source data is provided in the Source Data File.

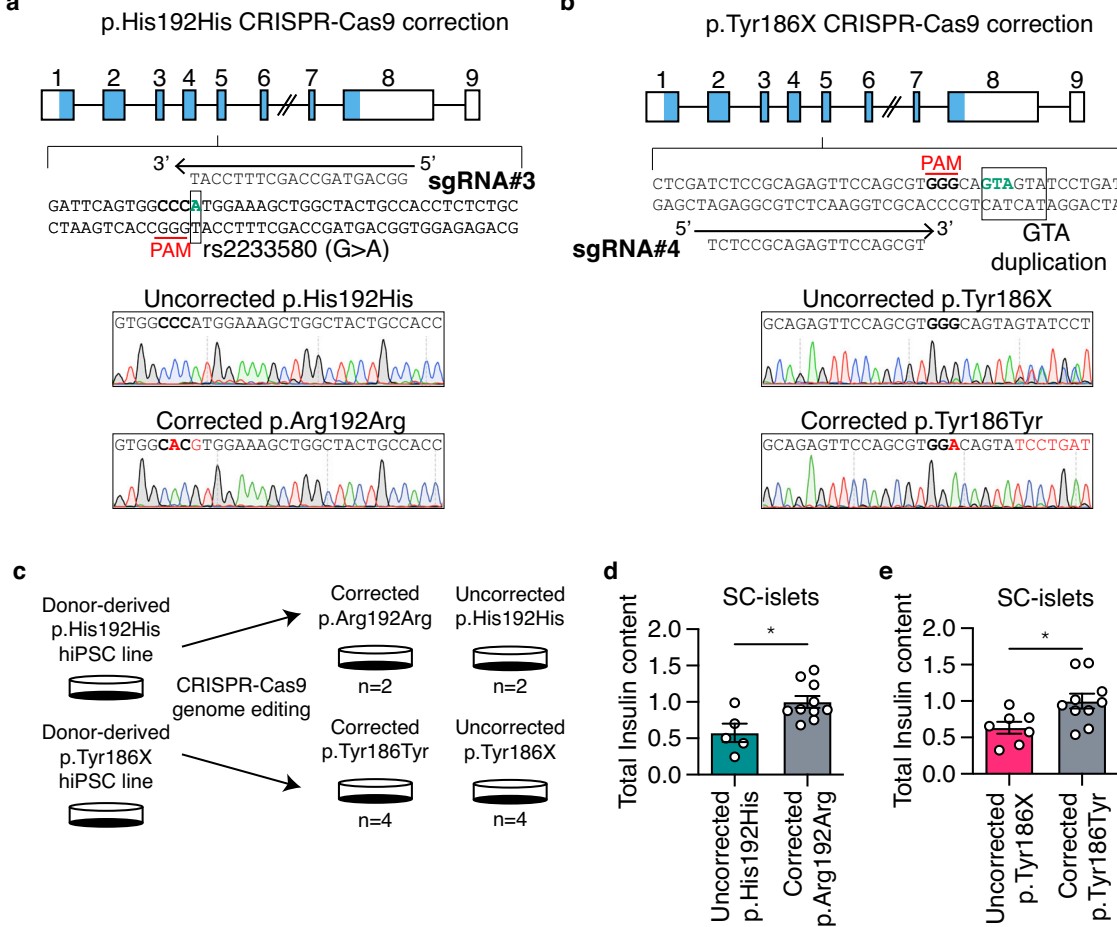

**Fig. 7 | CRISPR-correction of p.Arg192His or p.Tyr186X allele demonstrated rescue in beta cell identity and total insulin content. a, b** CRISPR-Cas9 gene editing strategy to correct the (**a**) p.His192His genotype to p.Arg192Arg (rs2233580) and (**b**) p.Tyr186X genotype to p.Tyr186Tyr (GTA duplication). Protospacer adjacent motif (PAM) sequence is bolded and the respective variant, and duplication are labeled in green. Representative Sanger sequencing results for the corrected allele are depicted with nucleotide changes labeled in red.

**c** Experimental design for CRISPR-Cas9 mediated gene correction and differentiation strategies. **d, e** Total insulin content of SC-islets derived from (**d**) uncorrected p.His192His and corrected p.Arg192Arg and (**e**) uncorrected p.Tyr186X and corrected p.Tyr186Tyr. *n* = 4 differentiation experiments were performed. Data are presented as mean ± SEM. Statistical analyses were performed by two-tailed unpaired Student's *t* test, **p* < 0.05. Differentiation protocol B was used to derive data in Fig. 7. Source data is provided in the Source Data File.

loss of PAX4 function (p.Tyr186X) demonstrated that *PAX4* haploinsufficiency is insufficient to cause monogenic diabetes, it was consistent with a negative impact on beta cell function. This finding is further supported by large sequencing studies that collectively show an association of rare alleles in *PAX4* with an increased risk for diabetes or elevated HbA1c levels.

While donor hiPSC-derived beta cells can be used to study human pancreas development in vitro, this experimental model suffers from

the following challenges: (1) hiPSC line-to-line variability, (2) the heterogenous nature of differentiated islet-like cells and (3) incomplete functional maturity of differentiated beta-like cells[49]. To circumvent these challenges while leveraging the benefits of this model, we rigorously applied two differentiation protocols to multiple donor-derived and genome-edited hiPSC lines. Our three independent sets of RNA-seq data using two protocols[25,28] in multiple hiPSC models (*PAX4*^KO/KO^ knockout, donor-derived and engineered hiPSCs carrying

*PAX4* variants, and donor-derived gene-corrected hiPSCs)[30] demonstrated that all differentiating cells shared a similar trajectory towards pancreatic islet-like cells. Knockout of *PAX4* did not result in the ablation of human beta cells, but rather, resulted in compromised beta cells with elevated expression of multiple endocrine hormone markers and lowered expression of genes associated with beta cell functional maturation. These observations were similarly replicated across donor-derived hiPSCs of three independent genotypes (p.Arg192His, p.His192His and p.Tyr186X), whereby SC-islets carrying *PAX4* alleles demonstrated increased polyhormonal gene expression and reduced total insulin content. Our molecular assessments confirmed p.X186 to undergo NMD, while the p.His192 resulted in altered transcriptional activity. Contrary to rodent models, our human in vivo and in vitro findings indicate that differentiating human beta cells are sensitive to the functional (haploinsufficiency; loss-of-function) *PAX4* gene dosage required to maintain beta cell identity, insulin production and secretion. Our data are consistent with a recent study on *HNF1A* deficiency[50] and support a model where *PAX4* T2D-risk alleles mediate disease risk by biasing endocrine precursor cells towards an alpha cell fate.

Transcriptomic assessment of EPs identified altered metabolic signatures in carriers of p.Arg192His or p.Tyr186X allele(s). Indeed, metabolic stress can compromise beta cell identity and has been proposed to be one of the mechanisms underlying beta cell exhaustion in T2D[51,52]. Unfortunately, the use of NAC to alleviate oxidative stress was insufficient to rescue the total insulin content in *PAX4* variant-expressing SC-islets. SC-islets derived from gene-corrected hiPSCs demonstrated a rescue in total insulin content, affirming the direct contribution of p.Arg192His or p.Tyr186X to decreased insulin content. While we were unable to determine the direct cause of the altered metabolic signature observed in EPs or whether the variants were a direct cause of the metabolic signature, it is tempting to hypothesize that the inferior beta cells resulting from the *PAX4* variants, compounded with cellular metabolic stress, hasten the eventual progression toward T2D development.

Our hiPSC-derived beta cells were not robustly functional under high glucose and secreted insulin only when treated with 30 mM potassium chloride (Supplementary Fig. 11). To circumvent the limitations of our hiPSC model in deriving functionally mature beta cells in vitro, we included the study of *PAX4* in the human beta cell line EndoC-βH1. Knockdown of *PAX4* led to a derepression of the *GCG* gene promoter and elevated *GCG* gene expression in beta cells. *PAX4* expression requires cooperative activation by several key transcription factors specific to pancreatic beta cells[13] to specify PAX4 exclusively in beta cells. In rodents, the maintenance of pancreatic beta cell identity requires a continual repression of non-beta cell gene expression[51,52]. The expression of multiple hormonal markers, including GCG, is one of the hallmarks of immature cells that could have diminished function in endocrine hormone secretion. The reduced *PAX4* levels resulting in the loss-of-repression of non-beta cell gene expression could possibly explain the co-expression of GCG in C-PEP-expressing SC-islets carrying the p.His192 or p.X186 allele(s) that is reversed with correction of the *PAX4* gene (Supplementary Fig. 10). We have observed that participants carrying p.Arg192His or p.Tyr186X alleles secrete less insulin during GSIS (Fig. 1), and this was recapitulated in our si*PAX4* and sh*PAX4* EndoC-βH1 cells (Fig. 2). The insulin content within pancreatic islet cells has a strong correlation with the amount of insulin secreted during GSIS[53]. The reduced total insulin content observed in our hiPSC and EndoC-βH1 models and subsequent impaired GSIS in EndoC-βH1 cells collectively suggest a role for PAX4 in maintaining beta cell identity and regulating insulin secretion function.

A limitation of our study is the description of a single family with a *PAX4* PTV, limiting the confidence with which conclusions can be drawn from our observations of an effect of *PAX4* haploinsufficiency in humans. We sought to strengthen our findings through the aggregation of exome-sequencing data from multiple publicly available datasets, which both provided nominal evidence for the role of rare coding variation in elevated diabetes risk and support mounting evidence that *PAX4* is not a monogenic diabetes gene[11].

The loss of beta cell identity and the acquisition of polyhormonal cells have been reported in the pancreatic islets of individuals with diabetes[54,55]. The transdifferentiation of metabolically stressed beta cells to express GCG has been proposed by several groups as a mechanism underlying beta cell failure in T2D[55–57]. In the current study, we demonstrate how coding gene variants in *PAX4* can influence pancreatic beta cell development, identity, and function, thereby predisposing East Asian carriers to higher risks of developing T2D.

## Methods
Our clinical data collection for this study was approved by the Singapore National Healthcare Group Domain Specific Review Board (NHG DSRB 2013/00937), and written informed consent was obtained from all participants. Participants were given monetary compensation for the time during which they participated in this study. The use of consented de-identified human cells (NHG DSRB 2013/00937) to generate hiPSCs for this study is covered by A*STAR IRB 2020-096. This study follows the principles of Declaration of Helsinki. SB Ad3.1 hiPSC, Lonza CC-2511, tissue acquisition number 23447, was purchased to be used in this study. We were not responsible for obtaining approval of its use and consent.

### Clinical studies
We recruited by genotype 183 individuals without diabetes (60 heterozygous for the p.His192 allele, 2 homozygous for the p.His192 allele, and 121 controls) from two sources: (1) We recruited individuals by genotype who did not have diabetes mellitus from the multi-ethnic cohort maintained by the Saw Swee Hock School of Public Health (Singapore Population Health Studies; information accessible at https://blog.nus.edu.sg/sphs/). A subset of participants underwent whole exome sequencing as part of another study. From this subgroup, 52 individuals who were heterozygous and 2 individuals who were homozygous for the p.His192 allele were recruited to participate in this study. In addition, we recruited 72 individuals (matched for age to the carriers) who were homozygous for the p.Arg192. (2) A cross-sectional study of individuals without diabetes recruited to identify biomarkers of beta cell function conducted at the same time in a metabolic phenotyping unit using the same study methodology provided an addition 57 individuals (8 heterozygous for the p.His192 allele and 49 individuals homozygous for the p.Arg192 allele). All participants were recruited and studied between 7 February 2013 and 21 August 2015. For both studies, the inclusion criteria were Chinese ethnicity, age between 21 and 80 years, non-smoker or no use of nicotine or nicotine-containing products for at least 6 months. Participants with a known history of diabetes mellitus, screening HbA1c greater than 6.5% or fasting plasma glucose greater than 7.0 mmol/L were excluded. Participants with weight loss greater than 5% of body weight in the preceding six months, major surgery in the last three months, a history of malignancy, estimated creatinine clearance based on the MDRD formula less than 60 mL/min, current corticosteroid use, or any clinically significant endocrine, gastrointestinal, cardiovascular, hematological, hepatic, renal, respiratory disease, or pregnancy were also excluded. Data on demographics and medical history were obtained through an interviewer-administered questionnaire. Height and weight were measured. All participants underwent confirmation of genotype by polymerase chain reaction-restriction fragment-length polymorphism analysis.

### Frequently sampled intravenous glucose tolerance test
A 3-hour intravenous glucose tolerance test was performed after an overnight 10- to 12-hour fast. Participants were required to abstain from strenuous physical activity, alcohol, and caffeinated beverages

24 h before the procedure. A bolus of intravenous 50% glucose (0.3 g/kg body weight) was given within 60 s into the antecubital vein. Regular insulin (Actrapid; NovoNordisk, Copenhagen, Denmark) was administered as a bolus injection at 20 min at a dose of 0.03 units/kg body weight. Blood was sampled from the contralateral antecubital vein at −15, −10, −5, 0, 2, 3, 4, 5, 6, 8, 10, 14, 19, 22, 25, 30, 40, 50, 70, 100, 140 and 180 min for assessment of plasma glucose (YSI 2300 STATPLUS; YSI Incorporated, Life Sciences, Yellow Springs, OH, USA) and insulin (Advia Centaur; Siemens Health-care Diagnostics, Hamburg, Germany). AIRg (acute insulin response to glucose) and Si (insulin sensitivity) were estimated using mathematical modeling methods (MINMOD Millennium, ver. 6.02). Disposition index (DI) was calculated as AIRg x Si. HOMA-B was computed using the formula: [20 × fasting insulin (µIU/ml)]/[fasting glucose (mmol/mL)−3.5].

## Oral glucose tolerance test (OGTT)

Fifty-seven participants (28 heterozygous and 1 homozygous for the p.His192 allele, and 28 homozygous for the p.Arg192 allele) agreed to return for a 3-hour oral glucose tolerance test. The test was performed after an overnight 10 to 12-h fast. A 75-gram glucose drink in 200 mL of water was administered orally over 5 min. Blood samples were collected via an intravenous cannula at −10, 0, 10, 20, 30, 45, 60, 75, 90,120, 150 and 180 min for glucose (YSI 2300 STATPLUS; YSI Incorporated, Life Sciences, Yellow Springs, OH, USA), insulin (Advia Centaur; Siemens Health-care Diagnostics, Hamburg, Germany), glucagon (Human Glucagon ELISA; BioVendor R&D, Shizuoka, Japan) and GLP-1 (Glucagon-like peptide-1 total ELISA; IBL International, Hamburg, Germany). HOMA-B was calculated using the formula: 20 × fasting insulin (µIU/mL)/fasting glucose (mmol/mL)−3.5. HOMA-IR was computed using the formula: fasting insulin (µIU/ml) × fasting glucose (mmol/mL)/22.5.

## Statistical analysis of clinical data

Analyses were carried out using SPSS software version 28.0 (SPSS Inc., Chicago, IL, USA). AIRg, Si, DI, HOMA-B, and HOMA-IR were log transformed to reduce skewness. Independent t-test and Chi-square tests were used to compare continuous and categorical variables between carriers and controls, respectively. A multiple linear regression model was used with adjustment by age, sex and BMI. Data are shown as the means (SD) or means (95% confidence intervals) and a $p$-value of <0.05 was considered statistically significant. Our study had >80% power to detect a 35% difference in AIRg assuming a one-sided alpha of 0.05 between carriers and non-carriers of the p.His192 allele. The recall-by-genotype dataset was enriched for individuals who carried the p.His192 allele. These individuals were also older. Given that older individuals could have more impaired beta cell function, this could lead to bias. To address this source of bias, all analyses were adjusted for age.

## Cell culture

All mammalian cells were routinely tested to be mycoplasma free using a MycoAlert™ PLUS mycoplasma detection kit (Lonza Bioscience, LT07-710). All mammalian cells were cultured in a 5% $CO_2$ humidified incubator at 37 °C. Unless otherwise stated, cells were passaged using 0.25% trypsin. Alpha TC clone 9 mouse pancreatic adenoma cells (αTC1.9) (ATCC, CRL-2350™, RRID:CVCL_0150) were cultured in low glucose DMEM (1.0 g/L) supplemented with an additional 1.0 g/L glucose (final glucose concentration to be 2.0 g/L), 10% FBS, 15 mM HEPES (ThermoFisher Scientific, 15630080), 1% NEAA (Gibco, 11140-50), 0.02% Bovine serum albumin (BSA) (Sigma-Aldrich, A9418) and 1.5 g/L sodium bicarbonate (ThermoFisher Scientific, 25080094). AD293 (Agilent, 240085, RRID:CVCL_KA63) and 293FT (Invitrogen, R70007, RRID:CVCL_6911) human embryonic kidney cell lines were cultured in high glucose DMEM supplemented with 10% FBS and 1% NEAA. EndoC-

βH1 cells[58] (Univercell Biosolutions EndoC-βH1, RRID:CVCL_L909) were cultured according to the manufacturer's recommendations. Briefly, tissue culture plates were precoated with high glucose DMEM supplemented with 2 µg/mL fibronectin (Sigma-Aldrich, F1141) and 1% ECM (Sigma-Aldrich, E1270) at least 30 min prior to cell plating. Low glucose DMEM (Gibco, 11885084) supplemented with 2% BSA, 10 mM nicotinamide (Sigma-Aldrich, N0636 or N3376), 2 mM GlutaMAX™ (Gibco, 35050061), 50 µM beta mercaptoethanol, 5.5 µg/mL transferrin (Sigma-Aldrich, T8158) and 6.6 ng/mL sodium selenite (Sigma-Aldrich, 214485). Cells were passaged weekly with 0.05% or 0.25% Trypsin and neutralized with 20% FBS in DPBS and plated at a density of 70,000 cells/cm$^2$. The SB Ad3.1 hiPSC line derived from human skin fibroblasts from a Caucasian donor with no reported diabetes (Lonza CC-2511, tissue acquisition number 23447) was obtained from the Human Biomaterials Resource Centre, University of Birmingham. The SB line and donor-derived hiPSC lines generated herein were cultured in TeSR™-E8™ or mTeSR-1™ medium (StemCell Technologies, 05990 or 85850) with daily media changes. hiPSCs were passaged twice weekly using ReLeSR™ (StemCell Technologies, 05872) or Accutase (Gibco, A1110501) according to the manufacturer's instructions. Culture plates were precoated with 0.1% gelatin in cell culture grade water for at least 10 min and then with MEF media for at least 48 h prior to plating or with Corning Matrigel hESC-Qualified Matrix (VWR International, BD354277) for at least an hour prior to plating.

## Generating donor-derived hiPSC lines

Skin punch biopsies were obtained from the upper forearm of recruited participants and cultured in low glucose DMEM supplemented with 10% heat inactivated FBS and 1% MEM non-essential amino acids (Gibco, 11140-50) to obtain fibroblasts. A Human Dermal Fibroblast Nucleofector™ Kit (Lonza Bioscience, VDP-1001) was used for episomal reprogramming of fibroblasts. Cells were trypsinized and washed with DPBS and 500,000 cells were resuspended in Nucleofector™ Solution according to the manufacturer's instructions. The following Yamanaka factors from Addgene were added at 1 µg to the cell suspension: pCXLE-hOCT3/4-shp53-F (plasmid #27077), pCXLE-hSK (plasmid #27078), and pCXLE-hUL (plasmid #27080). The nucleofection program P22 was used to transfect cells. At the end of the nucleofection, cells were plated onto mitotically inactivated CF1-MEF (plated one day in advance) (Lonza Bioscience, GSC-6201G) and cultured in DMEM/F12 (Gibco, 10565018) media supplemented with 20% KnockOut™ serum replacement (Gibco, 10828010), 1% NEAA, and 10 ng/mL FGF-2 (Miltenyi Biotec, 130-093-842). Media were replaced daily until hiPSC colonies emerged.

Peripheral blood mononuclear cells (PBMCs) were extracted from donor blood using a BD Vacutainer® CPT™ Mononuclear Cell Preparation Tube (BD Biosciences, 362753). The white buffy coat layer containing PBMCs was collected, washed twice with DPBS, and centrifuged at 300xg for 10 min to pellet the cells. One to two million cells were seeded and cultured in expansion media: IMDM media (Gibco, 12440053) supplemented with 10% FBS, 50 µg/mL of L-ascorbic acid (Sigma-Aldrich, A8960), 50 ng/mL of Stem Cell Factor (RnD Systems, 255-SC-010), 10 ng/mL IL-3 (StemCell Technologies, 78040), 2 U/mL Erythropoietin, 40 ng/mL IGF-1 (BioVision, 4119), 1 µM dexamethasone (Sigma-Aldrich, D8893) and 0.2% Primocin (Invivogen, ant-pm-1). PBMCs were reprogrammed following the manufacturer's instructions using CytoTune™-iPS 2.0 Sendai Reprogramming Kit (Invitrogen™, A16517) to obtain hiPSCs. For reprogramming, 200,000 PBMCs were plated in 12-well plates, and Sendai viruses [hKOS (MOI5), hc-Myc (MOI5), and hKlf4 (MOI3)] were added to culture media supplemented with 8 µg/mL of Polybrene (Sigma-Aldrich, TR-1003). Media was replaced the next day. Cells were collected and plated onto mitotically inactivated CF1-MEFs (pre-seeded one day in advance) and cultured in DMEM/F12 media supplemented with 20% KOSR (Gibco, 10828010), 1% NEAA (Gibco, 11140-50) and 10 ng/ml FGF-2 (Miltenyi Biotec, 130-093-

842), supplemented with 50 μg/mL of L-ascorbic acid, 50 ng/mL of Stem Cell Factor, 10 ng/mL IL-3, 2 U/mL Erythropoietin, 40 ng/mL IGF-1, and 1 μM dexamethasone for the first two days. The reprogrammed PBMCs were then maintained in basal media without growth factor and small molecule supplementation until hiPSC colony formation. hiPSC colonies were handpicked and cultured with a TeSR™-E8™ Kit. Each colony was designated to be one hiPSC line and expanded for cryo-preservation, with two to three independent lines per donor.

Immunofluorescence staining was performed on all hiPSC lines used in this study to confirm the expression of pluripotency markers OCT3/4, SOX2, NANOG, SSEA-4 and TRA1-60. One representative donor-derived hiPSC line from each donor was submitted for kar-yotyping (Cytogenetic laboratories, Singapore General Hospital) and for teratoma assay (A*STAR Biological Resource Centre (BRC) Animal Facility). Teratomas were then sent to Advanced Molecular Pathology Laboratory (AMPL, A*STAR) for paraffin block processing, sectioning and H&E staining. Derivation of all three germ layers (definitive endoderm, mesoderm and ectoderm) was confirmed using light microscopy. A summary of all the hiPSCs generated in this study can be found in Supplementary Table 2.

### CRISPR-Cas9 genome editing of hiPSCs

To generate $PAX4^{KO/KO}$ SB Ad3.1 hiPSC lines, a strategy was designed to mirror the well-studied $Pax4^{-/-}$ mice where almost all of the functional domains were replaced with a beta galactosidase-neomycin resistance cassette[14]. To delete the majority of the paired and homeodomains, sgRNAs were designed targeting exon 2 (sgRNA#1: CTAGGGCGT-TACTACCGCAC) and exon 5 (sgRNA#2: TATCCTGATTCAGTGGCCCG) of $PAX4$ gene (ENST00000341640.6). Successfully edited clones were detected using genotyping PCR and primers flanking the exon 2 through 5 deletion. To generate the p.Arg192His and p.Tyr186X vari-ants in the SB Ad3.1 hiPSC line, sgRNA#2 was electroporated with HDR template with either rs2233580 (G > A) mutation or GTA duplication, respectively. To correct the donor-derived p.His192His and p.Tyr186X hiPSCs, sgRNA#3 (GGCAGTAGCCAGCTTTCCAT) or sgRNA#4 (TCTCCGCAGAGTTCCAGCGT) were electroporated with an HDR repair template, respectively. sgRNAs were synthesized following the manufacturer's instructions using the EnGen sgRNA Synthesis Kit, $S.$ $Pyogenes$ (NEB, E3322), followed by DNase treatment and RNA pur-ification using the RNA Clean & Concentrator Kit (Zymo Research, R1017). Ribonucleoprotein (RNP) complexes were formed by com-bining 20 μM (681 ng) sgRNA, 20 μM Cas9 (NEB, M0646T) and Buffer R (ThermoFisher, MPK109R) in a total volume of 6 μL and incubating at room temperature for 15 min. The RNP complex was then combined with 250,000 hiPSCs in 15 μL of Buffer R and incubated on ice for 5 min. Ten microliters of the RNP+cell mixture was electroporated in two separate electroporations using the Neon™ Transfection System 10 μL Kit (ThermoFisher, MPK1025). Electroporated cells were seeded into Matrigel-coated plates with mTeSR media and 10 μM Y-27632 (StemCell Technologies, 72302). Forty-eight hours after electropora-tion, hiPSCs were plated at low-density (5000 cells/60 mm dish) on Matrigel-coated plates with mTeSR and 10 μM Y-27632. The resulting colonies were handpicked and expanded for further genotyping and quality control measures, including karyotyping and pluripotency staining. Primers were designed to amplify and Sanger sequence the top predicted off-target sites for each of the four gRNAs. No mutations were detected in any of the sequenced potential off-targets sites. A summary of all the hiPSCs generated in this study can be found in Supplementary Information.

### hiPSC differentiation into SC-islets

For differentiation experiments using Protocol A, hiPSCs were cultured in mTeSR™1 with daily media changes and passaged using Accutase. Cells were plated at $10^6$ cells/well in Growth Factor Reduced Matrigel (Corning, 356230)-coated CellBind 12-well tissue culture plates (Corning, 356230 and 3336) in mTeSR1 (StemCell Technologies, 85850) supplemented with 10 μM of Y-27632 dihydrochloride (AbCam, ab120129). The following morning, the medium was changed to mTeSR™1, and differentiation was started 24 h after plating. Directed differentiation protocol was adapted from Rezania et al. and basal differentiation media (using MCDB-131) was formulated accordingly[25]. Media was changed daily to basal media supplemented with growth factors and small molecules (Supplementary Table 3) with the fol-lowing modifications: Activin A and CHIR 99021 were used for Stage 1; all stages were performed in planar culture; and stages 6 and 7 were both 6 days in length.

For differentiation experiments using Protocol B, hiPSCs were plated and maintained in 10 cm plates until 80–90% confluency. Fol-lowing, hiPSCs were washed with DPBS and dissociated into single cells using TrypLE™ Express (Gibco, 12605-010). Cells were seeded at a density of 1 million cells per mL of mTeSR™1 kit supplemented with 10 μM of Y-27632 (StemCell Technologies, 72303) on non-treated 6 well plate. Cells were then incubated in tissue culture incubator on an orbital shaker with a shaking speed set at 80 rpm over a duration of 24–48 h before the start of differentiation by changing to differentia-tion media supplemented with growth factors and/or small molecules. Directed differentiation protocol was adapted from Pagliuca et al. and basal differentiation media (S1, S2, S3, S5 and S6) were formulated accordingly[28]. Fresh differentiation media supplemented with growth factors and small molecules were added at stipulated timepoints for directed differentiation over a duration of 35 days. The details of the culture medium and key reagents used for Protocol B can be found in Supplementary Table 3.

### Cloning

$PAX4$ plasmid (pLenti6.2/V5-DEST-PAX4, HsCD00329734) was pur-chased from DNASU plasmid repository. Full-length $PAX4$ sequence was confirmed via Sanger sequencing before subcloning into an engineered lentiviral vector pCDH-MCS-EF1-GFP to include a 5' Flag tag and a 3' V5 tag within the multiple cloning site (MCS). The full-length $PAX4$ sequence was subcloned into the pCDH-MCS-EF1-GFP vector for protein expression. Refer to Supplementary Table 4 for the list of primers used for cloning in this study. Cloning primers hPax4FLXbaI1F (forward) and hPax4FLV5Xho1R (reverse) were used to amplify full-length $PAX4$ sequence. PCR was performed using Phusion™ High-Fidelity DNA Polymerase (ThermoScientific, F530) for sequence amplification. Thermal cycling conditions were set according to the manufacturer's instructions. Restriction enzyme digestion was per-formed on the pCDH-MCS-EF1-GFP and amplified $PAX4$ sequence independently using XbaI (New England Biolabs, R0145) and XhoI (New England Biolabs, R0146) according to the manufacturer's instructions. Digested products were resolved using gel electrophoresis and gel extraction was performed using Purelink™ Quick Gel Extraction Kit (Invitrogen, K210012). Ligation was performed using Quick Ligation Kit (New England Biolabs, M2200S) according to the manufacturer's instructions. The ligated plasmids were transformed into home-made competent cells propagated from Stbl3™ competent cells (Invitrogen, C7373-03) and sequentially amplified in LB broth for plasmid extrac-tion using PureLink™ HiPure Plasmid Filter Maxiprep Kit (Invitrogen, K210017). To introduce p.Arg192His and p.Tyr186X mutations into the pCDH-$PAX4$ plasmid, site-directed mutagenesis (SDM) primers were designed (Supplementary Table 4) and SDM was performed according to the procedures described.

For gene promoter cloning, human genomic DNA extracted from AD293 cells was used as template. The basic luciferase vector, pGL4.10 (Promega) was used as cloning vector for gene promoters. Briefly, primers targeting the promoter region (-1 nucleotide from ATG translational start site) were designed (Supplementary Table 4). Tar-geted promoter regions were amplified, digested with restriction enzyme, ligated and transformed into competent cells similarly as

described in the previous section. With the exception of the insulin gene promoter, all other gene promoters used in this study were amplified from human gDNA and subcloned into pGL4.10 at the multiple cloning site. The pGL4.10 *INS* promoter plasmid was synthesized by IDT (gBlocks™ Gene Fragments). Length of the various gene promoters used is as follows: *PAX4* −1384 bp, *INS* −1499 bp, *GCG* −1068 bp, *SST* −718 bp.

The design of shRNA sequence to knockdown *PAX4* gene was referenced to Genetic Perturbation Platform (Broad Institute), Clone ID TRCN0000015989. The shRNA targets the coding sequence CGGATCCTTAAGGTATCTAAT within *PAX4* gene (Supplementary Table 4). The shRNA was ligated into pLKO.1 vector using Quick Ligation Kit (New England Biolabs, M2200S) according to the manufacturer's instructions. Successfully ligated sh*PAX4* plasmid was amplified for subsequent experiments.

### Lentiviral-mediated sh*PAX4* stable line generation

Third generation lentivirus system was used for this study. Lentivirus plasmids used for virus production: pRC/CMV-Rev (Rev), pHDM-HIVgpm (Gag/Pol) and pHDM-G (Vsv-g). Non-targeting shScramble and shRNA targeting human *PAX4* (sh*PAX4*) gene were subcloned into pLKO.1 vector for lentiviral packaging in 293FT cells. For the generation of stable lines, EndoC-βH1 cells were plated onto 10 cm plates. The cells were then transduced with pLKO.1 shScramble or sh*PAX4* lentiviruses in the presence of 8 μg/mL polybrene. After 72 h, the transduced cells were cultured in EndoC-βH1 media supplemented with 500 μg/mL of G418 antibiotic (Invivogen, ant-gn-1). In parallel, one plate of untreated EndoC-βH1 cells (plated at the same density) was cultured in the same antibiotic-supplemented media as a control. Media were replenished routinely during this selection process. The surviving EndoC-βH1 cells were expanded to obtain stable lines.

### EndoC-βH1 gene silencing using siRNAs

Knockdown studies in EndoC-βH1 cells were performed using Lipofectamine RNAiMAX® transfection protocol and 15 nM SMART pool ON-TARGETplus siRNAs (Horizon Discovery Biosciences, si*NT*: D-001810-10-05, si*PAX4*: L-012240-00-0005) diluted in Opti-MEM reduced serum-free medium (ThermoFisher Scientific, 31985062) and 0.4% RNAiMAX® (ThermoFisher Scientific, 13778150). Silencing efficiency was determined by qPCR from samples collected during GSIS, five days post-transfection.

### Glucose-stimulated insulin secretion (GSIS) assay

si*NT* and si*PAX4* EndoC-βH1 cells were seeded in 48-well plates at a density of 180,000 cells six days prior to GSIS. Cells were gently washed three times with pre-warmed secretion assay buffer (114 mM sodium chloride, 4.7 mM potassium chloride, 1.2 mM calcium chloride, 1.2 mM potassium phosphate, 1.16 mM magnesium sulfate, 25 mM sodium bicarbonate, 0.2% fatty acid-free BSA (Proliant, 68700), 20 mM HEPES, adjusted to pH 7.3). Cells were then incubated in secretion assay buffer for 1 h before being stimulated with 2.8 mM or 16.7 mM glucose for 40 min. The supernatant was collected at the end of 40 min for insulin secretion measurements and insulin content was collected using RIPA buffer. AlphaLISA human insulin research kit (Perkin Elmer, AL204C) was used to measure insulin secretion and content. Total protein measurements were determined using a Pierce BCA Protein Assay Kit (Life Technologies, PI23227). The stimulation index was calculated by normalizing to total protein (quantified using BCA) and relative to 2.8 mM glucose.

sh*PAX4* and shScramble EndoC-βH1 cells were seeded in 12-well plate prior to GSIS assay. Before the GSIS assay, cells were gently washed three times with warm Krebs Ringer bicarbonate (KRB) buffer (125 mM sodium chloride, 4.74 mM potassium chloride, 1 mM calcium chloride, 1.2 mM potassium phosphate, 1.2 mM magnesium sulfate, 5 mM sodium bicarbonate, 0.1% fatty acid-free BSA, 25 mM HEPES,

adjusted to pH 7.5 ( ± 0.2) with 1 M sodium hydroxide). After that, cells were subjected to normalization at 2.8 mM glucose for 1 h before being stimulated at 2.8 mM and 16.7 mM glucose for 30 min each sequentially. At the end of each stimulation step, KRB buffer was collected for human insulin ELISA (Mercodia, 10-1113-10). The stimulation index was computed by insulin secreted at 16.7 mM divided by insulin secreted at 2.8 mM glucose. Total insulin was extracted from each sample after the whole process of GSIS was completed.

### Dynamic glucose-stimulated insulin secretion

hiPSC-derived SC-islets were collected at the end of differentiation using Protocol B and the insulin secretion capacity was assessed using an automated perifusion system (Biorep). With a flow rate set at 100 μL/min, SC-islets were subjected to 60 min of perfusion with 2.8 mM glucose in KRB buffer before sequential stimulation with 2.8 mM (6 min), 16.7 mM (40 min), 2.8 mM (16 min) and 30 mM KCL (6 min). Secreted insulin was collected as flowthrough for quantitation.

### Total insulin content extraction

At the end of the 35-day directed differentiation using Protocol B, SC-islets from each hiPSC line were handpicked, and 400 μL of acid/ethanol solution was added. Cells were vigorously vortexed and subjected to repeated pipetting to break up cell clumps. SC-islets were then incubated at 4 °C overnight before total insulin extraction. For EndoC-βH1 cells, 500,000 cells were seeded onto each well of a 12-well plate. At the end of the experiment, cells were washed thrice with DPBS before adding 500 μL of acid/ethanol solution to each well and incubated at 4 °C overnight prior to insulin extraction. After overnight incubation, the insulin extracts were centrifuged at 300 g for 5 min. The top aqueous layer containing insulin was collected and subjected to human insulin ELISA assay (Mercodia, 10-1113-10) while the bottom layer (containing cell pellet) was boiled to dryness at 80 °C on a heat block. The dried cell pellet was resuspended in water for total DNA quantification. All data involving total insulin content quantification were normalized to total DNA. For donor hiPSC-derived SC-islets, the average total insulin content extracted from various cell lines from each donor in one experiment is represented as a single data point on the graph. For genome-edited cells, each data point represents the average of total insulin extracted from one cell line in one experiment. For EndoC-βH1 cells, each data point represents the average of total insulin extracted in independently sampled triplicates in one experiment.

### Gene expression analysis

Total RNA was extracted using MN NucleoSpin RNA Kit (Macherey-Nagel). RNA was quantified and reverse transcribed to cDNA using High-capacity cDNA reverse transcription kit (Applied Biosystems, 4368813). qPCR was performed using iTaq™ Universal SYBR ® Green Supermix (Bio-rad, 172−5124). Thermal cycling was performed using CFX384 Touch Real-Time PCR System (Bio-rad). Relative quantification of each gene expression was normalized to *ACTIN*, calculated by the $2^{-ddCt}$ method. The sequences of qPCR primers used in this study are available in Supplementary Table 5.

The above method was used for all gene expression studies, except for siRNA EndoC-βH1 experiments. For those samples, RNA was extracted using RNeasy Mini Kit (Qiagen, 74104), reverse transcribed using GoScript Reverse Transcription System (Promega, A5001), and qPCR was performed using TaqMan Gene Expression Master Mix and Assays (Applied Biosystems, 4369106; *PAX4*: Hs00173014_m1; *TBP*: Hs00427620_m1) on QuantStudio 7 Real-Time PCR Instrument.

### Taqman allelic discrimination assay

Custom Taqman® Assay Design Tool (ThermoScientific) was used to design probes specific for either the wildtype (p.Arg192 or p.Tyr186) or *PAX4* variant transcripts (p.His192, ANT2HTM or p.X186, ANU7DDJ). A

template sequence of approximately 600 bp around the SNP of interest was used as a reference to design custom assay probe. For Taqman qPCR assays, cDNA from EPs was used as the template. A 5 μL assay with TaqMan® SNP Genotyping MasterMix (Applied Biosystems, 4351384) was prepared according to the manufacturer's instructions. Relative fluorescence units (RFUs) from the HEX probe (wildtype allele) and the FAM probe (either p.Arg192 or p.X186 allele) were analyzed using the CFX384 Touch Real-Time PCR System (Bio-rad).

## Immunofluorescence staining

For pluripotency staining, each donor-derived hiPSC line was seeded onto a few wells of a precoated 12-well plate. AD293 and EndoC-βH1 cells were seeded onto uncoated and coated coverslips in 12-well plates, respectively. For overexpression studies, transfection was performed on seeded cells using Lipofectamine™ 2000 transfection reagent (Invitrogen, 11668-019) or FuGENE® 6 transfection reagent (Promega, E2691) according to the manufacturer's instructions. hiPSC-derived EPs and SC-islets were collected and sent to Advanced Molecular Pathology Laboratory (AMPL, A*STAR) for cryo-embedding, cryo-block processing and sectioning. For IHC, cryosections were thawed and dried at room temperature before staining. Cells were washed thrice with DPBS and fixed with 4% paraformaldehyde (WAKO, 163-20145) for 20 min. Blocking and cell membrane permeabilization were performed using DPBS supplemented with 5% Donkey serum (Merck Millipore, S-30) and 0.1% Triton-X-100 (Merck Millipore, 9410) for 1 h at 4 °C. Cells were incubated with primary antibodies overnight at 4 °C. Cryosections were then incubated with corresponding secondary antibodies for 1 h at room temperature. For nuclear staining, cryosections were incubated with DAPI (1:5000) (Sigma-Aldrich, D9542) in DPBS for 20 min before mounting onto glass slides for imaging with Olympus Fluoview Inverted Confocal microscope. Refer to Supplementary Table 6 for the list of antibodies used and their respective dilution factors for IHC.

## SDS-PAGE and Western blot

Cells were washed with DPBS and lysed in M-PER™ (Mammalian protein extraction reagent) (ThermoScientific, 78501) in the presence of protease inhibitor cocktail (Sigma-Aldrich, P8340), phosphatase-2 inhibitor (Sigma-Aldrich, P5726), and phosphatase-3 inhibitor (Sigma-Aldrich, P0044). Protein was quantified using Pierce™ BCA protein assay kit (ThermoScientific, 23227) according to the manufacturer's instructions before being separated with SDS-PAGE and transferred to PVDF membrane. Protein blots were first blocked with 5% milk in TBST (1X Tris-Buffered Saline, 0.1% Tween 20) for 1 h before incubating with primary antibody for either 2 h at room temperature or overnight at 4 °C. Blots were washed and then incubated with the respective HRP-conjugated secondary antibody for 1 h. Chemiluminescence signals were visualized after incubation with Super Signal™ West Dura Extended Duration Substrate (ThermoScientific, 34076). Refer to Supplementary Fig. 12 for uncropped Western blot images. Refer to Supplementary Table 6 for the list of antibodies used and their respective dilution factors for western blotting.

## Flow cytometry

DE cells generated with Protocol A were collected at the end of Stage 1 using Accutase. For extracellular staining, cells were washed twice with 1X Flow Cytometry Staining Buffer (RnD, FC001). Cells were blocked in Flow Cytometry Staining Buffer with FC block for 5 min before adding human anti-CXCR4 antibody for 45 min. Cells were then washed twice with Flow Cytometry Staining Buffer. For intracellular staining, cells were fixed using BD CytoFix Buffer (BD Biosciences, 554655) for 20 min on ice before washing twice with PBS. Using BD Perm/Wash Buffer (BD Biosciences, 554723), fixed cells were permeabilized for 30 min on ice, washed three times, and human anti-SOX17 antibody was incubated for 1 h at 4 °C before a final wash step in PBS. Stained cells were acquired

on SH800 Cell Sorter (Sony), and data analysis was performed using FlowJo™ 10.6.0.

DE, EPs and SC-islets were collected on D3, D20 and D35, respectively, following differentiation with Protocol B before being dissociated into single cells using TrypLE™ Express (Gibco, 12605-010). Cells were passed through a 40 μm cell strainer and single cells were fixed with 4% PFA for 20 min on ice. Antigen blocking and cell permeabilization were performed using DPBS supplemented with 5% FBS (HyClone, SV30160.03) and 0.1% Triton-X-100 for 30 min on ice. Cells were stained with primary antibodies for 1 h at room temperature. The cells were then washed three times with DPBS and incubated with corresponding secondary antibodies for 1 h at room temperature. Flow cytometry analyses were performed with BD® LSR II Flow Cytometer (BD Biosciences) and data was analyzed using FlowJo™ software (BD Biosciences). Refer to Supplementary Table 6 for the list of antibodies used and their respective dilution factors for flow cytometry. Refer to Supplementary Fig. 13 for the gating strategy that was applied to all flow cytometry analyses in this study.

## Luciferase assays

Cells were plated in triplicate one day prior to co-transfection with 0.5 μg of pCDH-overexpression constructs encoding *PAX4* or its variants, 0.4 μg of pGL4.10 luciferase vector and 10 ng of TK Renilla vector. Transfection was performed using either Lipofectamine 2000 (Invitrogen, 11668-019) or FuGENE® 6 transfection reagent (Promega, E2691). Cells were lysed with lysis buffer at the end of transfection (24 h for AD293, 48 h for αTC1.9 and 72 h for EndoC-βH1 cells). Luciferase assay was performed using Dual-Glo® Luciferase Reporter Assay Kit (Promega, E2920) following the manufacturer's instructions. The luciferase firefly activity was normalized against the Renilla readings within each well to account for variation in transfection efficiency across replicate wells. Each triplicate was normalized to the mean of the pCDH-MCS-EF1-GFP-empty control.

## Seahorse metabolic assays

EPs on D19 of directed differentiation using Protocol B were dissociated into single cells using TrypLE™ Express (Gibco, 12605-010) before passing through 40 μm cell strainer. 80,000 or 120,000 cells were plated with S5 differentiation medium[28] supplemented with 10 μM of Y-27632 onto pre-coated Seahorse microplate one day prior to analysis. The same number of cells were seeded across all cell lines within each experiment. On the day of glycolysis stress test (Agilent Seahorse XF Glycolysis Stress Test Kit), cells were washed with unbuffered serum-free assay medium (DMEM 5030, Sigma-Aldrich; supplemented with 2 mM L-glutamine). Following, the cells were incubated in assay medium in a non-CO$_2$ incubator at 37 °C for 1 – 2 h before measurements were taken. Extracellular acidification rates (ECAR) were measured using Seahorse XFe96 analyzer (Seahorse Bioscience) at pre-set timings prior to and following sequential injections of 10 mM glucose, 1.5 μM oligomycin and 50 mM 2-deoxy-glucose (2-DG). The same number of cells was seeded across the various genotypes for individual experiments. Four to eight technical replicates were seeded for each cell line. Each data point on graph represents the average of all replicates from one cell line. For the Mito Stress Test (Agilent Seahorse XF Cell Mito Stress Kit), cells were washed with unbuffered serum-free assay medium supplemented with 20 mM glucose (keeping the glucose level consistent with S5 differentiation medium), 2 mM pyruvate and 2 mM L-glutamine prior to incubation in assay medium in a non-CO$_2$ incubator at 37 °C for 1 – 2 h before analysis. Oxygen consumption rates (OCR) were measured using Seahorse XFe96 analyzer prior to and following sequential injections of 1.5 μM oligomycin, 1 μM FCCP and 0.5 μM rotenone/antimycin-A. Similarly, the same number of cells was seeded across the various genotypes for individual experiments. Four to eight technical

replicates were seeded for each cell line. One data point on the graph represents the average of all replicates from one cell line.

## RNA sequencing and analysis

Total RNA was extracted from samples generated using differentiation Protocol A at the end of Stage 1 (DE), 4 (PE), 5 (EP), and 7 (SC-islets) using RNeasy Mini Kit following the manufacturer's instructions. Polyadenylated transcripts were isolated using NEBNext PolyA mRNA Magnetic Isolation Module (New England Biolabs, E7490). Sequencing libraries were prepared using the NEBNext Ultra Directional RNA Library Kit with 12 cycles of PCR and custom 8 bp indexes (New England Biolabs, E7420). Libraries were multiplexed and sequenced on the Illumina NovaSeq 6000 as 150-nucleotide paired-end reads. Reads were mapped to human genome build hg19 (GRCh37) using STAR v.2.5[59], with GENCODE v19 (https://www.gencodegenes.org/human/release_19.html) as the transcriptomic reference. featureCounts from the Subread package v1.5 (http://subread.sourceforge.net/) was used to perform gene-level quantification. Differential expression analysis was performed per stage using DESeq2[60] comparing *PAX4*$^{WT/WT}$ and *PAX4*$^{KO/KO}$ cell lines. First, the model was fit using a likelihood ratio test with genotype as a factor of interest and experiment as a covariate. From this, genes not within the top 5000 most significant genes were used as an empirical control (affected only by unwanted experimental variation) and the estimated factor of unwanted variation (k = 1) was calculated using the RUVg function from RUVSeq[61]. To identify differentially expressed genes, DESeq2 was performed using a likelihood ratio test and including the factor from RUV as a covariate along with the technical replicate (experiment). Significance was determined by padj <0.05. Sashimi, TPM, and volcano plots were generated using ggplot2 in RStudio.

Total RNA was extracted from samples generated using differentiation Protocol B on day 0 (hiPSC), day 13 (PP2), day 20 (EP) and day 35 (SC-islets). Poly-A mRNA (10 – 100 ng) was used to construct multiplexed strand-specific RNA-seq libraries (NEXTflexTM Rapid Directional RNA-SEQ Kit, dUTP-Based, v2). The quality of individual libraries was assessed and quantified using Agilent 2100 Bioanalyzer and Qubit 2.0 fluorometer before pooling for sequencing using a HiSeq 2000 (1×101 bp read). Prior to cluster formation, pooled libraries were quantified using the KAPA quantification kit (KAPA Biosystems). The processing of raw RNA sequencing data was performed in collaboration with Molecular Engineering Laboratory (A*STAR) to remove low-quality sequence reads. Filtered read sequences were mapped onto human genome (hg19). Fragments per kilobase million (FPKM) were used to calculate differential expression between patient lines using DESeq2. Using UMAP (Uniform Manifold Approximation and Projection) and PCA (Principal Component Analysis) dimension reduction clustering for all four timepoints, 11 out of 164 samples were classified as outliers and excluded from clustering analyses.

For the generation of PCA plot for EPs differentiated using protocol B, the TPM read counts for each gene within the transcriptome were first normalized to calculate a standardized score. To derive a standardized score, we applied the following formula: $\log_{10}$(((GOI's TPM counts for sample of interest +1)/(average TPM counts for GOI across all samples) + 1), where GOI represents a gene of interest within the whole transcriptome. Using this standardized score, PCA analysis was performed in R via the prcomp() function. The ggplot2 package was used to plot the final PCA biplot based upon the PC1 and PC2 loadings obtained from prcomp(). Finally, the stat_ellipse() function was used to cluster the transcriptome of cells with the various *PAX4* genotypes based upon a confidence interval of 90%.

## Statistical analysis

Statistical analyses were performed using GraphPad Prism version 9. Data are presented as the standard error of the mean (SEM). Unless otherwise specified, unpaired Student's *t* tests were performed to compare the means of two groups, and one-way ANOVA was performed to compare the means among three or more groups. Statistical significance was set at $p < 0.05$.

## Reporting summary

Further information on research design is available in the Nature Portfolio Reporting Summary linked to this article.

## Data availability

Data reported in this study were deposited into public data repositories, with accession numbers provided. Protocol A *PAX4*$^{KO/KO}$ RNA-seq data: EGAS00001006036. Data is available through the EGA - https://ega-archive.org. Applicants should request access to the data using the data access agreement form (available from EGA or through https://med.stanford.edu/genomics-of-diabetes/datasets.html). Upon receipt of completed forms, a decision on access will be made by the data access committee (DAC) within 14 days. The chair of the DAC will inform EGA and EGA will release the data to the account listed in the data access agreement. Forms should be returned to Anna L. Gloyn, Stanford University (agloyn@stanford.edu). Protocol B RNA-seq data: GSE203265. The processed RNA-seq data generated from this study are available as Supplementary Data 1-5. Source data are provided with this paper.

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

## Acknowledgements

The authors thank A/P Yee Joo Tan (IMCB, A*STAR) for her support with antibodies and Daniela Moralli (University of Oxford) for her support with hiPSC karyotyping. We thank the Oxford Genomics Centre at the Wellcome Centre for Human Genetics (funded by Wellcome Trust grant reference 203141/Z/16/Z) for the generation and initial processing of the sequencing data. Data used in this manuscript has received funding from the Innovative Medicines Initiative Joint Undertaking under Grant Agreement number 115439 (StemBANCC), resources of which are composed of financial contribution from the European Union's Seventh Framework Programme (FP7/2007–2013) and EFPIA companies in kind contribution. H.H.L. is supported by the Institute of Molecular and Cell Biology (IMCB) Scientific Staff Development Award (SSDA) for her part-time Ph.D. N.A.J.K. is supported by the Stanford Maternal and Child Health Research Institute Postdoctoral Fellowship. A.L.G. is a Wellcome Senior Fellow in Basic Biomedical Science. A.L.G. is funded by the Wellcome (200837) and National Institute of Diabetes and Digestive and Kidney Diseases (NIDDK) (U01-DK105535, U01-DK085545, UM1DK126185, U01DK123743, U24DK098085) and the Stanford Diabetes Research Center (NIDDK award P30DK116074). A.K.K.T. is supported by IMCB, A*STAR, Lee Foundation Grant SHTX/LFG/002/2018, FY2019 SingHealth Duke-NUS Surgery Academic Clinical Programme Research Support Programme Grant, Precision Medicine and Personalized Therapeutics Joint Research Grant 2019, the 2nd A*STAR-AMED Joint Grant Call 192B9002, HLTRP/2022/NUS-IMCB-02, Paris-NUS 2021-06-R/UP-NUS (ANR-18-IDEX-0001), OFIRG21jun-0097, CSASI21jun-0006, MTCIRG21-0071, SDDC/FY2021/EX/93-A147, FY 2022 Interstellar Initiative Beyond grant, H22G0a0005 and I22D1AG053.

## Author contributions

Conceptualization: E.S.T., A.L.G., A.K.K.T. Data curation: M.P.A., H.S., A.J. Formal Analysis: H.H.L., N.A.J.K., M.P.A., H.S., A.J., A.L.G., A.K.K.T. Funding acquisition: E.S.T., A.L.G., A.K.K.T. Investigation: H.H.L., N.A.J.K., F.A., J.W.C., J.A., S.G., Y.L., J.Y., S.T., B.C., S.H., N.S.T., D.G., S.L.K., A.L.G., A.K.K.T. Methodology: H.H.L., N.A.J.K., F.A., M.P.A., J.A., A.L.G., A.K.K.T. Project administration: E.S.T., A.L.G., A.K.K.T. Resources: H.H.L., N.A.J.K., F.A., J.A., S.H., D.G., S.L.K., E.S.T., A.L.G., A.K.K.T. Software: M.P.A., A.J. Supervision: E.S.T., A.L.G., A.K.K.T. Validation: H.H.L., N.A.J.K. Visualization: H.H.L., N.A.J.K. Writing—original draft: H.H.L., N.A.J.K., A.L.G., A.K.K.T. Writing—review & editing: All authors approved.

## Competing interests

A.L.G.'s spouse is an employee of Genentech and holds stock options in Roche. A.K.K.T. is a co-founder of BetaLife Pte Ltd. The remaining authors declare no competing interests.

## Additional information

¹Stem Cells and Diabetes Laboratory, Institute of Molecular and Cell Biology (IMCB), Agency for Science, Technology and Research (A*STAR), Proteos, Singapore. ²School of Biological Sciences, Nanyang Technological University, Singapore, Singapore. ³Division of Endocrinology, Department of Pediatrics, Stanford University School of Medicine, Stanford, CA, USA. ⁴Wellcome Centre for Human Genetics, University of Oxford, Oxford, UK. ⁵Oxford Centre for Diabetes Endocrinology and Metabolism, University of Oxford, Oxford, UK. ⁶Cancer Science Institute of Singapore, National University of Singapore, Singapore, Singapore. ⁷Molecular Engineering Laboratory, Institute of Molecular and Cell Biology (IMCB), Agency for Science, Technology and Research (A*STAR), Proteos, Singapore. ⁸Lee Kong Chian School of Medicine, Nanyang Technological University, Singapore, Singapore. ⁹Department of Endocrinology, Singapore General Hospital, Singapore, Singapore. ¹⁰Department of Medicine, National University Hospital and National University Health System, Singapore, Singapore. ¹¹Department of Medicine, Yong Loo Lin School of Medicine, National University of Singapore, Singapore, Singapore. ¹²Saw Swee Hock School of Public Health, National University of Singapore, Singapore, Singapore. ¹³Stanford Diabetes Research Center, Stanford University, Stanford, CA, USA. ¹⁴Department of Biochemistry, Yong Loo Lin School of Medicine, National University of Singapore, Singapore, Singapore. ¹⁵Present address: Faculty of Pharmaceutical Sciences, University of British Columbia, Vancouver, BC, Canada. ¹⁶These authors contributed equally: Hwee Hui Lau, Nicole A. J. Krentz. ¹⁷These authors jointly supervised this work: Anna L. Gloyn, Adrian Kee Keong Teo. ✉e-mail: agloyn@stanford.edu; ateo@imcb.a-star.edu.sg

