## [Peer Review File · Nature Communications]

REVIEWER COMMENTS

Reviewer #1 (Remarks to the Author):

This study by Lau&Krentz et al. contains comprehensive clinical and experimental studies that overall support the title of the paper. The results are mostly clear and well described. However, there are several problems related with Fig.7 (the rescue experiments by mutation correction).

Major comments:

1. Differentiation of iPSC into islet-like cells: Two well-known but related protocols were used (refs 22 and 25). The study comes from two collaborating laboratories. Does this mean that one lab used protocol A and the other one B, or were both protocols established at both sites, or only one site? Was this planned and coordinated, or were the results of two labs combined retrospectively? It would be fair to make this clear.

2. It would also be important to present at least the elementary general characterization of the differentiation efficiencies using the two protocols in the Supplementary materials. How do the yields of Pdx1/Nkx6.1 endocrine progenitors at stage 4 compare? What about the yield of beta and alpha cells? And insulin secretion in response to glucose plus other secretagogues (like GLP1, KCl)?

It is also very important to clearly indicate which protocol was used for each result presented (missing at least for Fig. 7).

3. Figure 4 k-n shows clearly the important effects of the LOF mutations on the insulin content and presence of polyhormonal cells. Why is this information lacking for the Pax4 KO lines? This is an important part of the basic characterization that is missing (see point 3), and it would be particularly important to include this characterization for the Pax4 KO lines in comparison with the isogenic WT controls.

4. An improved protocol has recently been published by Balboa et al (PMID 35241836), showing further maturation and the loss of polyhormonality during extended final stage. It would be interesting to see the impact of Pax4 LOF (both the KO and the mutations studied) on the C-Pep/GCG double positive cells after extended maturation.

5. The mutation correction results presented in Fig. 7 are very difficult to understand:

- The genotypes of the lines are expressed confusingly. Clearly, in panel a, it is the p.Arg192His mutation that is corrected (and not His192His)? Similar confusing labels can be found also in other panels.
- Panels d and e: Labelling of the lanes is very confusing. I believe that in (d) the same labelling applies for both two lanes on the left and right (uncorrected vs corrected), and in (e) the same label is supposed to cover 4 lanes (?). This is really difficult to understand, so please clarify.
- Furthermore, I simply do not understand what the “isogenic BLC” stand for? The term “isogenic” in my understanding should mean that the cells have the same background genotype, i.e- are derived from the same donor. So, what are these? And how should one interpret the result when the color scale is completely different from the “donor-derived BLC”. I find it hard to identify obvious patterns in the expression that would make sense (this applies also the replicate clones particularly in panel d).
- Based on a lot of evidence provided in the manuscript, a hallmark of Pax4 LOF should be an increased alpha-cell expression pattern, represented by higher ARX expression. However, in panels d-e it appears that ARX is consistently higher in the corrected than in uncorrected cells?

Minor:

1. Title: The title could be a bit more specific; instead of “endocrine cell”, rather “islet cell” or “pancreatic endocrine cell”; and instead of “influences”, rather “increases”). As a result, perhaps an appropriate title could be: “Pax4 loss of function increases diabetes risk by altering human pancreatic endocrine cell development”.

2. Line 195: HbA1c was 7.1%/8.7 mmol/L. If the intention is to express the HbA1c in molar units, I suppose this should be around 54 mmol/mol. But perhaps the authors mean that the HbA1c of 7.1% corresponded to a mean b-glucose level of 8.7 mmol/L. Please clarify.

3. Figure 1f: In the table the glucose values are obviously in mmol/L, but the column title has mg/dL.

Reviewer #2 (Remarks to the Author):

The study by Lau/Krentz and colleagues expand our current knowledge on the functional role of the transcription factor PAX4 in human islet development, and function, and how mutations, including a new one identified herein in a Singapore family (p.X186), may contribute to type 2 diabetes risk. Using both clinical and experimental approaches that include the generation of iPSCs-derived beta-like cells obtained from individuals with this new mutation combined with CRISP and RNAseq technologies, the authors demonstrate that PAX4 is essential to repress the alpha genetic program which favors beta cell identity hallmarked by insulin content and functional insulin secretion. The study is of interest with a massive amount of data that will be resourceful for the scientific community.

General comments:

Mouse studies have shown that overexpression of the Pax4 diabetes-linked mutant variant R129W in adult mice beta cells sensitize to stress and apoptosis (Hu He et al., Diabetes 2011 and Mellado Gil et al., Cell Death Dis 2016). Did the authors contemplate this venue in BLC that may rationalize reduced beta cell function and insulin content in carriers of PAX4 mutations?

Please remove references from abstract

Figure 1e: Please label in figures proband III-1 and 2.

Figure 2b to h: Please provide the actual relative values rather than a fold change.

Figure 3g: Please add the reference to this panel in line 281. Furthermore, please indicate that the bold GO terms are the ones that include ARX (I guess).

Figure 5b and c: The authors claim that CHX stabilizes the p.X186 allele while having no effect on p.His192 (line 338-341). To this reviewer's interpretation, CHX has no effect on either (red and white dots in b and green and white dots in c). Please clarify as this may have a major impact on the conclusion that NMD of p.X186 is the mechanism of haploinsufficiency.

Extended Data Fig 4: It is unclear whether the immunofluorescence staining in (d) and the Western blot in (f) were performed with the PAX4 or V5 antibody. Please provide this information as the PAX4 antibody may not recognize the truncated p.186X variant.

Figure 7d: Please clarify the origin of each lane. For example, uncorrected refers to a line derived from the CRISPR correction that failed while p.His192His is the initial line used for CRISPR correction? The

same goes for lanes 3 and 4. If so it may be worth emphasizing that CRISPR globally alters the transcriptome landscape. It also may be worth highlighting in bold genes which are mentioned in the text. Interestingly PDX1 was increased in the corrected version.

Figure 7e: Columns 1, 3,5 and 8 lack labeling. The conclusion drawn by the authors is truly a leap of faith for the donor-derived BLC. In fact, the corrected version appears to have fewer beta cell-enriched factors as compared to the uncorrected version. Nonetheless, the isogenic data is clear-cut and conclusive.

Reviewer #3 (Remarks to the Author):

The authors report interesting data with attempts to illustrate functionality of PAX4 identified from GWAS. Cutting-edge technologies were applied including application of beta-like cells (BCLs) derived from patient iPSCs and precise genome-editing by CRISPR/Cas9. Such attempts are much desired for GWAS follow-up studies; however, the present study draws conclusions without sound causal evidence, and suffers from major methodological concerns.

1. Lack of GSIS study in BCLs

This is a fundamental aspect when using BCLs derived from patient iPSCs. If the BCLs do not respond to glucose on insulin secretion, the authors cannot draw firm conclusion on the effect of any genetic manipulation (KD or genome editing) to have any causal effect on beta cell function. This aspect was completely ignored throughout the study, and there is no illustration on how BCLs were characterized as BCLs.

2. PAX protein expression in all tissues is extremely low or absent (please check in various public databases), including in the pancreas. Have the authors taken this into consideration when designed the study?

3. Page 8 line 172, reduction of HOMA-B was no longer significant after adjusted for age, gender and BMI? What about only BMI, or age, or gender, or any combination? How would the authors interpret the data?

4. In Fig 2, gene expression data was presented as “relative to siNT/scramble”. As detailed in the “Methods” session in Page 41, only 1 housekeeping gene ACTIN was used in gene expression experiments. At least 2 housekeeping genes should be used, and what are the Ct values of PAX4? According to public databases, in general PAX4 gene expression is very low in all tissues.

5. CRISPR guides: what is the deletion? Between the 2 guides? There is no description of the confirmation of deletion (e.g. by Sanger). The authors did not address off target effects either.

6. PAX4 Arg192His variant (chr7:127253550, rs2233580) is located in Exon 5, the location of Tyr186X is also in Exon 5. CRISPR sgRNA#2 cutting site is around R192, A191 and V190I (theoretically 3-4 bps before or after the PAM seq). What is exactly the deleted region, does it cover the two SNP loci? (Line 259: deletion for AA 64-200?) What is the rationale to design a big deletion fragment in relation to the two GWAS loci?

7. It would be better to use PAX4 wt instead of PAX4^{+/+}, which may be confused with insertion mutation.

8. Page 13, why did the authors choose protocol B over A, and applied protocol B for the rest of the study? PAX4 expression was absent in BLC by protocol B. Volcano plots in panel d and f showed completely different gene profiles generated from protocol A and B, except for DENND2D, any follow-up study on DENND2D? In panel f, PAX4 is again absent, not among the top genes that were affected by PAX4 kd, which makes it further questionable the choice of protocol B.

9. Page 14, line 290-292, this is an overstatement: "PAX4 is not required for human beta cell differentiation in vitro. Rather, PAX4 loss-of-function results in derepression of alpha cell genes and a dysregulation of key endocrine maturation genes in hiPSC-derived BLCs." As stated on page 13 Line 283, nonsignificant.

10. What is the ethnic origin of the hiPSC lines used in Fig 3? Furthermore, the PAX4 variants were identified in Asian populations, and EndoC cell line is of French origin. (<https://www.ncbi.nlm.nih.gov/pmc/articles/PMC3163974/>). Indeed, many human derived cell lines do carry ethnic-specific signatures. Any thoughts?

11. What is the consequence of increased GCG gene expression in translation? In protein expression detection? Fig 4 IF stained for GCG. The mammalian proglucagon gene (Gcg) encodes three glucagon like sequences, glucagon, glucagon-like peptide-1 (GLP-1), and glucagon-like peptide-2. Have the authors looked at other potential translational products?

12. Line 320-322, this is an overstatement. In his192 allele carriers, no change in PAX4 or INS, GCG, SST transcription was detected in this study. Line 322, "negatively affecting endocrine cell differentiation", this is a bold conclusion based on only GCG staining, without further data to back up endocrine cell differentiation.

13. PAX4 primers: hPax4F5 (AGGACACGGTGAGGGTCTGGT) shows no blot result (please correct me if I was wrong); hPax4R5 (CAGTGGTTCCAGGGCAGGCA) is located in Exon 8. Tyr186X is supposed to introduce premature stop codon at AA186 (line 333-334). How is the amplification region in relation to the variants?

14. What is the epitope of PAX4 antibody?

15. ChIP should be performed to demonstrate interaction between PAX4 and gene promoter regions.

16. Fig 5h shows that PAX4 kd had no effect on INS promoter activity, how to explain Fig 2g where it shows increased INS gene expression by PAX4 kd?

17. Fig 6e and g, there is no significant difference between the variants. Should be carefully interpreted in data presentation.

18. Line 411, these changes are not significant in Extended data Fig6

19. Fig 7f and g, Extended Data Fig 7, wt is missing in all experiments for control

Point-by-point response

Reviewer #1 (Remarks to the Author):

This study by Lau&Krentz et al. contains comprehensive clinical and experimental studies that overall support the title of the paper. The results are mostly clear and well described. However, there are several problems related with Fig.7 (the rescue experiments by mutation correction).

We thank Reviewer #1 for his/her support of our manuscript. We have addressed the concerns relating to Fig. 7 below. Thank you.

Major comments:

1. Differentiation of iPSC into islet-like cells: Two well-known but related protocols were used (refs 22 and 25). The study comes from two collaborating laboratories. Does this mean that one lab used protocol A and the other one B, or were both protocols established at both sites, or only one site? Was this planned and coordinated, or were the results of two labs combined retrospectively? It would be fair to make this clear.

Thank you for providing us the opportunity to clarify how this collaborative study was performed, as we think it is a major strength of our study. Our laboratories (Teo and Gloyn) and collaborators embarked on this study together and coordinated our efforts so that each team led on parts of the study where they had the most expertise. The Teo group led on the generation of hiPSC-derived donor cells and differentiation of all cell lines using Protocol B. The Gloyn group focused on CRISPR-Cas9 genome correction of donor-derived cells and generation of isogenic PAX4 KO and variant hiPSC lines. The bulk of the differentiations were performed by Adrian Teo's group using Protocol B, whilst Anna Gloyn's group used Protocol A for differentiation of the PAX4 KO cells. Both labs performed experiments on EndoC-βH1 cells.

The decision to predominantly use Protocol B was made after we did the head-to-head comparison of the two protocols (Gloyn Protocol A and Teo Protocol B) on the PAX4 KO cells. We did not observe significant differences between the two protocols transcriptionally (**Supplementary Fig. 3**), with both showing peak *PAX4* expression in the endocrine progenitor stage (**Figs. 3c and 3e**). As the Teo lab was tasked with performing the bulk of the *in vitro* stem cell differentiations and had extensive experience using Protocol B (Low *et al.*, 2021; Nguyen *et al.*, 2021), the remaining differentiations were performed using Protocol B. We hope the reviewers agree that one of the strengths of our study is in the combined expertise of our two groups, along with the clinical data from our collaborators, which allowed us to more comprehensively address how genetic variation at the *PAX4* locus contributes to T2D risk.

2. It would also be important to present at least the elementary general characterization of the differentiation efficiencies using the two protocols in the Supplementary materials. How do the yields of Pdx1/Nkx6.1 endocrine progenitors at stage 4 compare? What about the yield of beta and alpha cells?

The focus of our collaborative study was to elucidate the biological mechanism of how coding variants in *PAX4* impact T2D risk by altering pancreatic beta cell development and/or function. As such, we did not intend to compare the differentiation efficiencies of Protocols A and B in deriving pancreatic beta-like cells. Because the differentiations were performed by two different labs and not done in parallel, we don't think our experimental setup is the best way to perform such protocol comparison studies.

We agree with the reviewer that the general characterization of the differentiation efficiencies is important. For Protocol A, we had previously characterized the differentiation towards the definitive endoderm stage (**Supplementary Fig. 2b-c**). For Protocol B, we had previously characterized the differentiation efficiency of BLCs on the donor-derived hiPSC lines (**Fig. 4j-m**) by immunofluorescence and cell quantification. Using this quantitative technique, we identified significant differences in the proportion of total nuclei that co-expressed C-PEP and GCG (**Fig. 4m**), with homozygous carriers of the p.His192 risk allele having significantly increased co-expression. To address the reviewer's concerns on the yield of beta and alpha cells in BLCs generated from donor-

derived hiPSCs carrying *PAX4* variants, we performed new flow cytometry experiments using Protocol B and evaluated the percentage of beta (INS+, PDX1+), alpha (GCG+) and delta (SST+) cells using flow cytometry (Supplementary Fig. 5). The efficiencies of generating beta (30-40%), alpha (~10%), and delta cells (15-20%) were comparable to the adapted protocol (Pagliuca *et al.*, 2014). Again, we did not see any distinct differences in the percentage of cell types when comparing the *PAX4* variant-carrying cells to wild-type donors.

A caveat of this data is that the wild-type cell lines are hiPSCs derived from independent donors. As we know that genetic background of individual donors can contribute to transcriptional differences (Rouhani *et al.*, 2014), the definitive experiment is to compare the effect of the genotype at the *PAX4* locus using CRISPR-corrected control cells. Therefore, we have now performed additional flow cytometry analyses on uncorrected p.Tyr186X vs corrected p.Tyr186Tyr hiPSC lines derived from the same donor, as well as uncorrected p.His192His vs corrected p.Arg192Arg hiPSC lines derived from the same donor (Supplementary Fig. 4). The new data lend support to the main message of our study, that *PAX4* variants/*PAX4* KO did not detrimentally impact the differentiation trajectory of hiPSCs towards pancreatic endocrine cells but impacted the maturation of BLCs (increased polyhormonal expression and reduced total insulin content; Fig. 4).

Supplementary Fig. 4

Supplementary Fig. 5

And insulin secretion in response to glucose plus other secretagogues (like GLP1, KCI)?

The functionality of hiPSC-derived BLCs is known to be variable and is highly dependent on the hiPSC line used. Previously, we stimulated the hiPSC-derived BLCs from Protocol B with high glucose and determined they were not functionally mature (static GSIS, data not shown). **We performed new experiments to evaluate dynamic GSIS using a perfusion system** from a selected set of cell lines (*PAX4*^{WT/WT} vs *PAX4*^{KO/KO}; uncorrected p.Tyr186X vs corrected p.Tyr186Tyr; and uncorrected p.His192His vs corrected p.Arg192Arg). Unfortunately, our hiPSC-derived BLCs using Protocol B were still not robustly functional under high glucose and secreted insulin only when treated with 30 mM potassium chloride (**Now Supplementary Fig. 11**). As expected, we also observed heterogeneity across cell lines carrying the same genotype, making it challenging to draw any conclusions from the data.

Supplementary Fig. 11 | Dynamic glucose-stimulated insulin secretion performed on beta-like cells. Donor hiPSC-derived beta-like cells carrying (a) uncorrected p.His192His (two lines); (b) corrected p.Arg192Arg (two lines); (c) uncorrected p.Tyr186X (one line); and (d) corrected p.Tyr186Tyr (one line) were stimulated at 2.8 mM (6 min), 16.7 mM (40 min), 2.8 mM (16 min) and 30 mM KCL (6 min) sequentially. Each graph represents data obtained from one hiPSC line.

To overcome the immature state of our hiPSC-derived BLCs, we included an alternative human beta cell line derived from human fetal pancreas – EndoC- β H1 – to demonstrate the consequence of *PAX4*

perturbation on beta cell function (Fig. 2). Importantly, in line with our observations in BLCs derived from hiPSCs, EndoC-βH1 cells knocked down for PAX4 also exhibited reduced maturity as witnessed by reduced total insulin content, which likely contributed to the impaired GSIS (Fig. 2c,e-f), and replicated the clinical phenotype of the recruited subjects (Fig. 1a,d, lower insulin secreted in response to glucose). We have added the following sentences to the discussion section (lines 523-526):

“Our hiPSC-derived beta cells were not robustly functional under high glucose and were demonstrated to secrete insulin only when treated with 30 mM potassium chloride (Supplementary Fig. 11). To circumvent the limitations of our study in deriving fully functional beta cells in vitro, we included the study of PAX4 in the human beta cell line EndoC-βH1.”

It is also very important to clearly indicate which protocol was used for each result presented (missing at least for Fig. 7).

We have now clearly indicated the protocol used in each experiment in the revised figure legends.

3. Fig. 4 k-n shows clearly the important effects of the LOF mutations on the insulin content and presence of polyhormonal cells. Why is this information lacking for the Pax4 KO lines? This is an important part of the basic characterization that is missing (see point 3), and it would be particularly important to include this characterization for the Pax4 KO lines in comparison with the isogenic WT controls.

We thank the reviewer for this comment and agree that it is important to characterize the effect(s) of PAX4 LOF in the KO lines. We have now performed **additional flow cytometry characterization for the PAX4 KO lines** using Protocol B (three control and three PAX4-KO lines; isogenic SB origin) and have inserted the data under **Supplementary Fig. 4**). The additional characterization revealed no distinct differences between control and PAX4 KO lines in their efficiencies in differentiating towards definitive endoderm (SOX17⁺/CXCR4⁺), endocrine progenitors (INS⁺, NKX6.1⁺ and PDX1⁺), as well as beta-like (INS⁺ and NKX6.1⁺) cells. However, due to the heterogeneous nature of hiPSC models, we were not able to detect any significant difference when comparing the total insulin content between the control and PAX4 KO lines (Figure below, data not included in the manuscript). The observations stay true with the key message of our study. Unlike in rodents, PAX4 KO did not detrimentally impact the differentiation trajectory of hiPSCs towards pancreatic endocrine cells as supported by **Supplementary Fig. 4a-c**. This has now been acknowledged in the results section (lines 293-300):

“Importantly, PAX4^{WT/WT} and PAX4^{KO/KO} lines differentiating into BLCs repress pluripotency genes and activate genes involved in endocrine cell fate in a similar manner (Supplementary Fig. 3), giving rise to similar proportions of DE, EP, and BLC (Supplementary Fig. 4a-c). These observations suggest that, unlike in mouse, PAX4 is not essential for human beta cell differentiation in vitro and its loss did not detrimentally impact the differentiation trajectory of hiPSCs towards pancreatic endocrine cells. Rather, PAX4 loss-of-function results in a dysregulation of key endocrine maturation genes in hiPSC-derived BLCs.”

Total insulin content in PAX4 wild-type and knockout hiPSC-derived beta-like cells.

4. An improved protocol has recently been published by Balboa et al (PMID 35241836), showing further maturation and the loss of polyhormonality during extended final stage. It would be interesting to see the impact of Pax4 LOF (both the KO and the mutations studied) on the C-Pep/GCG double positive cells after extended maturation.

We thank the reviewer for this suggestion. The Balboa *et al.* protocol involves initial differentiation as monolayer (definitive endoderm to posterior gut tube), transiting to microwell culture (from pancreatic to endocrine progenitor) and further maturation as suspension culture over a duration of up to six weeks (total differentiation duration adding up to over 60 days). In Balboa *et al.*, hESC (human embryonic stem cell) H1 line was predominantly used to optimize the differentiation protocol to achieve beta-like cells with remarkable insulin secretion profile. While robust insulin secretion was still observed, differentiation of two other hiPSC lines with the same protocol resulted in variable amounts of insulin being secreted (Balboa *et al.*, 2022; Supplementary Figure 1h). The difficulty of adapting differentiation protocols to new hiPSC lines is consistent with our labs' experience in using hiPSCs as a platform for disease modeling (Chan and Teo, 2020; Perez-Alcantara *et al.*, 2018). To overcome the variability in hiPSC differentiation *in vitro*, we used a combination of donor-derived hiPSCs (16 lines), CRISPR-corrected donor-derived isogenic hiPSCs (10+ lines), SB Ad3.1 (commercially available and then genome edited at the *PAX4* locus, 10+ lines), and subjected them to differentiation using two established protocols (Protocols A and B). As can be appreciated, the scale of this approach to model disease using *in vitro* differentiation is time consuming and expensive, but is necessary to overcome variability and to include proper controls. Importantly, our donor hiPSCs were generated from individuals without diabetes and we have begun to make several of these lines available to the wider research community (<https://skip.stemcellinformatics.org/en/>).

We agree that the Balboa *et al.* protocol represents a substantial update in our ability to generate functional beta-like cells *in vitro*. However, we hope that the reviewer will agree that it is out of scope of our current manuscript for the following reasons:

- 1) Our current study already used two different differentiation protocols. To adopt another differentiation protocol for the number of hiPSC lines we have would be a tremendous effort.
- 2) While Protocols A and B do not generate functional beta-like cells, we have complemented these studies with clinical data from human carriers of *PAX4* alleles (**Fig. 1**), which show lower insulin secreted in response to glucose.
- 3) Consistent with the clinical data, *PAX4* perturbation in the human beta cell line EndoC- β H1 (**Fig. 2**) reduced total insulin content, which likely contributed to the impaired GSIS (**Fig. 2e-f**) and replicated the clinical phenotype of the recruited subjects (**Fig. 1**).

5. The mutation correction results presented in Fig. 7 are very difficult to understand:

- The genotypes of the lines are expressed confusingly. Clearly, in panel a, it is the p.Arg192His mutation that is corrected (and not His192His)? Similar confusing labels can be found also in other panels.

We thank the reviewer for highlighting this point and we apologize for the lack of clarity. In panel a, it is indeed p.His192His (homozygote for His allele encoded by A nucleotide) that is being corrected into p.Arg192Arg (wildtype: **CGT**). For simplicity, we have only shown the one allele/double stranded DNA sequence, but the genotype at both alleles is being corrected from A -> G. We have now added an additional schematic indicating what the corrected genotype is. We hope this improves the clarity of the CRISPR-correction strategy.

- Panels d and e: Labelling of the lanes is very confusing. I believe that in (d) the same labelling applies for both two lanes on the left and right (uncorrected vs corrected), and in (e) the same label is supposed to cover 4 lanes (?). This is really difficult to understand, so please clarify.

We apologize for the confusion. In panels d and e, each column represents one cell line. Yes, the same labeling applies for both lanes on the left (meaning that two lines carrying the uncorrected genotype, p.His192His, were displayed) and the "corrected" label applies to the two lanes on the right (two lines with corrected genotype, p.Arg192Arg, wildtype). Similarly, there are four lines with uncorrected genotype (p.Tyr186X, mutant) and four lines with corrected genotype (p.Tyr186Tyr, wildtype) displayed in panel e. We have added additional lines to demarcate which columns correspond to which genotype.

- Furthermore, I simply do not understand what the "isogenic BLC" stand for? The term "isogenic" in my understanding should mean that the cells have the same background genotype, i.e. are derived from the same donor. So, what are these? And how should one interpret the result when the color

scale is completely different from the “donor-derived BLC”. I find it hard to identify obvious patterns in the expression that would make sense (this applies also the replicate clones particularly in panel d).

In panels d and e, each column represents one cell line. For panel d, the first lane (on the left) refers to the parental line carrying the p.His192His variant (donor-derived hiPSC), second lane is parental lane that underwent CRISPR correction but retained the p.His192His mutation (CRISPR control line), while third and fourth lanes represent successfully corrected lines that are now of the wildtype genotype p.Arg192Arg. As there is only one line each for the parental and control line, we placed them together under the same label as the two lines are carrying “Uncorrected p.His192His” variant.

In panel e, there are two donor-derived hiPSC lines naturally carrying the p.Tyr186X variant (lanes 1 and 4) and two lines that underwent CRISPR correction but retained the p.Tyr186X genotype (lanes 2 and 3, CRISPR control lines). Lanes 5 to 8 refer to CRISPR-corrected donor-derived hiPSCs. We noticed higher heterogeneity in hiPSCs derived from donor II-7 (as in panel e), making it harder to draw a conclusion. Again, we placed the parental and CRISPR control lines under the same label as they are carrying the same “Uncorrected p.Tyr186X” variant. Due to the high heterogeneity, “isogenic BLCs” were introduced to provide better clarity.

In panels d and e, the lanes under “isogenic BLC” originate from SB Ad3.1 cell line, with p.His192His/p.Tyr186X mutation knocked-in (on the left) and wildtype control (on the right). We termed them “isogenic” because the two lines (mutation knocked-in and control) were derived from the same source SB Ad3.1 hence sharing isogenic background. As we only have one line each for the *PAX4* variant knocked-in and control line, even though **the same color scale had been applied**, there will only be two colors to represent the directional expression of the genes (red and blue, in this case).

Comparing the “Uncorrected” columns composed of parental and CRISPR controls, as can be appreciated, panels d and e highlighted the heterogeneous nature of hiPSCs even in lines that were derived from the **same donors**. We appreciate the reviewer for recognizing the heterogeneous nature of hiPSCs even though the cells share an isogenic background.

To address the reviewer’s concern in missing obvious patterns in the transcript expression, **we performed additional experiments** to support the hypothesis that the dysregulated gene expression profiles in *PAX4* variant carrying BLCs contributed to impaired beta cell maturation. We have incorporated the new data under Supplementary Fig. 10 to demonstrate that BLCs with corrected genotypes had reduced number of polyhormonal cells (C-PEP⁺/GCG⁺) and hence had higher total insulin content (Fig. 7f-g).

“Importantly, correcting the PAX4 p.His192His and p.Tyr186X mutations decreased the formation of polyhormonal (C-PEP⁺/GCG⁺) cells (Supplementary Fig. 10) and significantly increased the total insulin content of the BLCs (Fig. 7f-g), indicating that the PAX4 variants were a direct cause of reduced insulin content in the donor-derived BLCs.”

- Based on a lot of evidence provided in the manuscript, a hallmark of Pax4 LOF should be an increased alpha-cell expression pattern, represented by higher ARX expression. However, in panels d-e it appears that ARX is consistently higher in the corrected than in uncorrected cells?

It is an interesting observation that *ARX* is consistently upregulated in the BLCs differentiated from donor-derived CRISPR-corrected lines. We took reference from a study by Prof. Timothy Kieffer’s team (Gage *et al.*, 2015), which reported an unexpected reduction of insulin-positive cells despite a significantly higher expression of *PAX4* seen in *ARX*-knockout pancreatic progenitors. Upon re-introduction of *ARX* into the *ARX*-knockout endocrine cells, Kieffer’s team observed a restoration of insulin-positive cells and hence confirmed the importance of *ARX* in beta cell development on top of its well-reported role in the regulation of alpha cell development.

We thank the reviewer for identifying the expression of *ARX* to be consistently higher in the gene corrected BLCs and we are equally curious to find out the effect of *PAX4* LOF in human endocrine cells. **We have now performed additional experiments to assess the expression of ARX in our shPAX4 EndoC-βH1 human beta cell line.** We observed significantly higher expression of *GCG* but lower *ARX* expression when *PAX4* was knocked down (Fig. 2i). The new data has been

incorporated/replaced Fig. 2d, 2g, 2h and 2i. Together with the findings by Kieffer's team that emphasized a role for *ARX* in both human beta and alpha cell lineage, our results are consistent with the expression of *ARX* not being a direct representative of alpha cell fate, at least in these human cell models.

Fig. 2

Minor:

1. Title: The title could be a bit more specific; instead of “endocrine cell”, rather “islet cell” or “pancreatic endocrine cell”; and instead of “influences”, rather “increases”). As a result, perhaps an appropriate title could be: “Pax4 loss of function increases diabetes risk by altering human pancreatic endocrine cell development”.

We thank the reviewer for the suggestion, we have now amended the title as suggested.

“PAX4 loss of function increases diabetes risk by altering human pancreatic endocrine cell development.”

2. Line 195: HbA1c was 7.1%/8.7 mmol/L. If the intention is to express the HbA1c in molar units, I suppose this should be around 54 mmol/mol. But perhaps the authors mean that the HbA1c of 7.1% corresponded to a mean b-glucose level of 8.7 mmol/L. Please clarify.

We have now clarified this in the revised manuscript (line 196). We have updated the unit for all blood glucose measurements to ‘mg/dL’, following American standard.

“Following lifestyle modifications, she lost weight (from 53.1 kg, BMI 25.3 kg/m² to 49.5 kg, BMI 23.6 kg/m²) and nine months post-diagnosis her HbA1c was 7.1% (mean basal glucose level of 156.6 mg/dL).”

3. Fig. 1f: In the table the glucose values are obviously in mmol/L, but the column title has mg/dL.

We thank the reviewer for pointing this out. We have now corrected the table title accordingly.

g	Subject	Age	BMI	Fasting glucose (mg/dL)	1-hour glucose (mg/dL)	2-hour glucose (mg/dL)	HbA1c (%)	HOMA-IR	DI
	III-1	12	21.2	86.4	ND	ND	5.6	3.31	ND
	II-11	41	24.1	95.4	174.6	118.8	5.5	1.98	4.34
	II-7	51	23.8	109.8	286.2	144.0	6.6	1.41	1.53
	II-3	58	27.0	81.0	147.6	109.8	5.6	1.75	10.76
	II-4	57	24.9	97.2	289.8	372.6	8.1	5.33	0.02
	II-5	55	24.3	190.8	349.2	396.0	8.8	3.77	0.32
	II-8	50	33.6	122.4	311.4	221.4	7.1	9.14	0.47
	II-9	55	25.4	154.8	309.6	331.2	7.0	5.17	0.4

Reviewer #2 (Remarks to the Author):

The study by Lau/Krentz and colleagues expand our current knowledge on the functional role of the transcription factor PAX4 in human islet development, and function, and how mutations, including a new one identified herein in a Singapore family (p.X186), may contribute to type 2 diabetes risk. Using both clinical and experimental approaches that include the generation of iPSCs-derived beta-like cells obtained from individuals with this new mutation combined with CRISP and RNAseq technologies, the authors demonstrate that PAX4 is essential to repress the alpha genetic program which favors beta cell identity hallmarked by insulin content and functional insulin secretion. The study is of interest with a massive amount of data that will be resourceful for the scientific community.

We thank Reviewer #2 for the positive comments, interest and support for our study.

General comments:

Mouse studies have shown that overexpression of the Pax4 diabetes-linked mutant variant R129W in adult mice beta cells sensitize to stress and apoptosis (Hu He et al., Diabetes 2011 and Mellado Gil et al., Cell Death Dis 2016). Did the authors contemplate this venue in BLC that may rationalize reduced beta cell function and insulin content in carriers of PAX4 mutations?

Pax4 is known to be expressed during pancreas development in the endocrine progenitors. In this study, we focused on the major effect of human PAX4 variants during pancreatic differentiation, rather than its role in adult beta cells and apoptosis. It is possible that the two PAX4 variants we studied could also play a role in stress and apoptosis. However, as coding variants will impact the protein from the point at which it begins to be expressed (pancreas development), any impact in mature beta cells would be in addition to, and perhaps caused by, PAX4's important role during development.

Please remove references from abstract

We have now removed all references from the abstract.

Figure 1e: Please label in figures proband III-1 and 2.

We thank the reviewer for highlighting this point. We have added label in Fig. 1e to identify the proband.

Figure 2b to h: Please provide the actual relative values rather than a fold change.

We thank the reviewer for this comment. We acknowledge the importance of Ct values particularly in knockdown studies. We have addressed this in the revised manuscript.

The siRNA studies were performed by the Gloyn team. In their hands, the Ct value of PAX4 ranged between ~28-32. As a comparison, the expression of INS in these cells amplified at an average CT of ~13, and the housekeeping gene TBP amplified at ~25. Unfortunately, the range in PAX4 expression across passages and knockdown experiments makes it difficult to plot as relative values. As you can see from the graph below, there was a 50% reduction in PAX4 expression following knockdown across all four experimental replicates. However, there is a greater than five-fold difference in the expression of PAX4 in the non-targeted siRNA control (siNT). As such, we have opted to present fold knockdown compared to siNT (effectively normalizing for the variability in PAX4 expression across experiments).

PAX4 expression relative to *TBP* in siRNA treated EndoC-βH1 cells.

The shRNA studies were performed by the Teo team. In their hands, the average Ct value for *PAX4* gene was ~26.7 in shSCR; ~27.2 in sh*PAX4* cells, with an average *ACTIN* Ct value of 20.4. We have consolidated and included additional information on the Ct values of *PAX4* and additional housekeeping genes (*ACTIN*, *GAPDH*, *TUBB3*, *PPIA*) in the figure below for better clarity for the reviewer (figures will not be included in the manuscript).

Figure 3g: Please add the reference to this panel in line 281. Furthermore, please indicate that the bold GO terms are the ones that include ARX (I guess).

Thank you for pointing out our oversight. We have cited this in the revised manuscript and updated the figure legend to explain that bold GO terms include ARX.

Figure 5b and c: The authors claim that CHX stabilizes the p.X186 allele while having no effect on p.His192 (line 338-341). To this reviewer's interpretation, CHX has no effect on either (red and white dots in b and green and white dots in c). Please clarify as this may have a major impact on the conclusion that NMD of p.X186 is the mechanism of haploinsufficiency.

We thank the reviewer for pointing this out. We have included a schematic here to explain how we concluded that p.X186 undergoes NMD. With treatment of CHX (which inhibits NMD) you see a rightward shift and stabilization of the p.X186 allele (increase from ~400 to ~800) but there is no stabilization of the p.His192 allele (as indicated by the overlap of green and white samples).

Supplementary Fig 4: It is unclear whether the immunofluorescence staining in (d) and the Western blot in (f) were performed with the PAX4 or V5 antibody. Please provide this information as the PAX4 antibody may not recognize the truncated p.186X variant.

The PAX4 antibody (RnD, AF2614) is polyclonal and the binding epitope(s) of the antibody is proprietary information not revealed by the company. We have evaluated this antibody on cells overexpressing PAX4 and its variant proteins when doing preliminary antibody screening to identify one that works well. In the figure below, we have demonstrated using western blot that the AF2614 antibody is able to detect both the p.192His and p.186X proteins. As V5 is located at the 3' end of PAX4 in the expression plasmid, p.186X introduced a premature stop codon that prevented the expression of V5. This is reflected in the missing band observed for V5 under the p.X186 lane (refer to the figure below). To clarify further, Supplementary Fig. 6f was blotted with PAX4 antibody instead, so that all the variant proteins can be captured. We thank the reviewer for identifying this point. The antibody information is now clearly annotated under the figure legend.

Image of western blot of WT PAX4 (WT), p.His192 and p.X186 proteins overexpressed in AD293 cells. PAX4 and V5 antibodies were used to assess protein expression 24 h post transfection.

Supplementary Fig. 6f: western blot using PAX4 RnD antibody.

Figure 7d: Please clarify the origin of each lane. For example, uncorrected refers to a line derived from the CRISPR correction that failed while p.His192His is the initial line used for CRISPR correction? The same goes for lanes 3 and 4. If so it may be worth emphasizing that CRISPR globally alters the transcriptome landscape. It also may be worth highlighting in bold genes which are mentioned in the text. Interestingly PDX1 was increased in the corrected version.

We apologize to the reviewer as our labeling on the original figure was unclear. We have now updated the figure to indicate that in panel d the first two columns in the donor derived BLCs include parental hiPSC line and a control line that went through the genome editing process but was unedited (both are labeled "Uncorrected p.His192His"). The second two columns are two independent hiPSC clones that were derived during the genome editing process and are corrected to be homozygous for the p.Arg192 allele ("Corrected p.Arg192Arg"). We have highlighted in bold the genes that were mentioned in the text.

Figure 7e: Columns 1, 3,5 and 8 lack labeling. The conclusion drawn by the authors is truly a leap of faith for the donor-derived BLC. In fact, the corrected version appears to have fewer beta cell-enriched factors as compared to the uncorrected version. Nonetheless, the isogenic data is clear-cut and conclusive.

We have now improved the labeling for Fig. 7e. There are two donor-derived hiPSC lines naturally carrying the p.Tyr186X variant (lanes 1 and 4) and two lines that underwent genome editing but remained unedited for the p.Tyr186X genotype (lanes 2 and 3) ("Uncorrected p.Tyr186X"). Lanes 5 to 8 refer to CRISPR-corrected donor-derived hiPSCs ("Corrected p.Tyr186Tyr").

Comparing the "Uncorrected" columns composed of parental and CRISPR controls, we agree with the reviewer that CRISPR editing could potentially alter the transcriptome landscape. However, we do not have sufficient cell lines to support this statement. Hence, we prefer to stay conservative in making this statement. We appreciate the reviewer for recognizing the heterogeneous nature of hiPSCs even through the cells share an isogenic background. This further substantiates the need to have various models to complement the findings derived from hiPSC-based models.

Reviewer #3 (Remarks to the Author):

The authors report interesting data with attempts to illustrate functionality of PAX4 identified from GWAS. Cutting-edge technologies were applied including application of beta-like cells (BCLs) derived from patient iPSCs and precise genome-editing by CRISPR/Cas9. Such attempts are much desired for GWAS follow-up studies; however, the present study draws conclusions without sound causal evidence, and suffers from major methodological concerns.

We thank Reviewer #3 for recognizing the cutting-edge technologies deployed in our study and the need for GWAS follow up studies. Below we respond to their comments and provide new data to support our conclusions.

1. Lack of GSIS study in BCLs

This is a fundamental aspect when using BCLs derived from patient iPSCs. If the BCLs do not respond to glucose on insulin secretion, the authors cannot draw firm conclusion on the effect of any genetic manipulation (KD or genome editing) to have any causal effect on beta cell function. This aspect was completely ignored throughout the study, and there is no illustration on how BCLs were characterized as BCLs.

The reviewer is correct that most studies deploying BCLs derived from patient hiPSCs do not result in BCLs that respond to glucose. For this very reason, and the fact our gene of interest is involved in development, we have focused our evaluation on the effect of PAX4 perturbation on islet cell development. Based on our observations across multiple human beta cell models, we demonstrate a developmental defect that results in increased polyhormonal cells with decreased total insulin content.

In response to Reviewers #1 and 2, we have **now carefully characterized both the endocrine progenitors and BCLs differentiated from the donor-derived hiPSCs using Protocol B**. In brief, the additional characterization revealed no distinct differences between the control and PAX4-KO/p.Tyr186X/p.His192His-carrying lines in their efficiencies to differentiate towards definitive endoderm (SOX17⁺/CXCR4⁺), endocrine progenitors (INS⁺, NKX6.1⁺ and PDX1⁺), as well as beta-like (INS⁺ and NKX6.1⁺) cells. The new data are now included in **Supplementary Fig. 4**. The efficiencies of generating beta-like cells (30-40% - INS, PDX1), alpha cells (~10%, GCG), and delta cells (15-20%, SST) were highly comparable to Pagliuca et al., 2014 (Protocol B), and not significantly different in wild-type versus PAX4 variants (**Supplementary Fig. 4**). While the flow cytometry analyses have demonstrated no significant difference in the cell populations, our immunostaining data revealed elevated expression of polyhormonal INS⁺/GCG⁺ cells (Fig. 4j-m) which led us to uncover that these cells are less mature and expressed lower total insulin content (Fig. 4n). **The new data lend support to the main message of our study that PAX4 diabetes associated variants and complete loss-of-function (PAX4-KO) do not detrimentally impact the differentiation trajectory of hiPSCs towards pancreatic endocrine cells (Supplementary Fig. 3a-b) but impact the maturation of BCLs** as witnessed by increased polyhormonal expression and reduced total insulin content (Fig. 4).

We also note the reviewer's concerns on the lack of evaluation of GSIS in BCLs in our study. The functionality of hiPSC-derived BCLs is known to be variable and highly dependent on the hiPSC line. We have previously stimulated the hiPSC-derived BCLs with high glucose but unfortunately they were non-functional (static GSIS, data not shown). We understand the request to include functional data from hiPSC-derived BCLs. Therefore **we have now performed new experiments and re-visited the insulin secretion capabilities of the BCLs** from a selected set of cell lines (PAX4^{WT/WT} vs PAX4^{KO/KO}; uncorrected p.Tyr186X vs corrected p.Tyr186Tyr; and uncorrected p.His192His vs corrected p.Arg192Arg), this time using a perfusion system to evaluate the dynamic GSIS in response to glucose and KCl. **We have now incorporated the new data under Supplementary Fig. 11.**

Unfortunately, our hiPSC-derived BCLs were still not robustly functional under high glucose stimulation and only secreted insulin when treated with 30 mM KCl (Supplementary Fig. 11). As expected, **we also observed heterogeneity across the cell lines carrying the same genotype (Supplementary Fig. 11), making it challenging to draw any robust conclusions from the data.** We therefore included a complementary human beta cell model derived from human fetal pancreas – EndoC-βH1, to evaluate the effects of PAX4 perturbation on insulin secretion (Fig. 2). Importantly, in line with our observations in BCLs derived from hiPSCs, our EndoC-βH1 model also exhibited

reduced maturity as demonstrated by reduced total insulin content which likely contributes to the impaired GSIS observed (Fig. 2e-f), recapitulating the clinical phenotype in subjects carrying the diabetes associated variants (Fig. 1a and 1d, lower insulin secreted in response to glucose). **This has now been acknowledged in the discussion section** (line 523-526):

“Our hiPSC-derived beta cells were not robustly functional under high glucose and secreted insulin only when treated with 30 mM potassium chloride (Supplementary Fig 11). To circumvent the limitations of the hiPSC model in deriving functionally mature beta cells in vitro, we included the study of PAX4 in the human beta cell line EndoC-βH1.”

We would like to emphasize that the **major focus of our study here is the role of human PAX4 and diabetes associated variants in this gene on human pancreatic differentiation**. Thus far, we have demonstrated a developmental defect that resulted in increased polyhormonal cells with decreased total insulin content. That said, we do acknowledge the limitations and difficulty in obtaining robustly functional hiPSC-derived BLCs (that we tried) with Protocols A and B used in this study.

Supplementary Fig. 11 | Dynamic glucose-stimulated insulin secretion performed on beta-like cells. Donor hiPSC-derived beta-like cells carrying (a) uncorrected p.His192His (two lines); (b) corrected

p.Arg192Arg (two lines); (c) uncorrected p.Tyr186X (one line); and (d) corrected p.Tyr186Tyr (one line) were stimulated at 2.8 mM (6 min), 16.7 mM (40 min), 2.8 mM (16 min) and 30 mM KCL (6 min) sequentially. Each graph represents data obtained from one hiPSC line.

2. PAX protein expression in all tissues is extremely low or absent (please check in various public databases), including in the pancreas. Have the authors taken this into consideration when designed the study?

We can confirm that we carefully considered the expression profile of PAX4 when designing our study. We were fortunate to have existing datasets for both RNA-seq and ATAC-seq from seven stages of pancreatic islet cell development from hiPSC, providing a rich resource to evaluate not only levels of *PAX4* expression during development but also chromatin activity at the locus. These data were key in our decision of which time points to focus on. *PAX4* expression peaks on D20 in the endocrine progenitor stage (Fig. 3c,e, EP stage) and decreases towards the BLC stage, recapitulating the expression pattern seen in rodents (Sosa-Pineda *et al.*, 1997). The expression of *PAX4* in mature human islets and in our cell model (EndoC-βH1) is indeed low. Others have reported that *Pax4* remains detectable in adult pancreatic beta cells (Brun *et al.*, 2004) and that *PAX4* expression in EndoC-βH1 is lower than human islets, but detectable (Tsonkova *et al.*, 2018). In our EndoC-βH1 models, the Ct values of *PAX4* via qPCR are approximately in the 26-32 range and vary across passages.

Taking reference from the rodent models, hiPSCs are an ideal model to evaluate PAX4 biology during human pancreatic beta cell development *in vitro*, given that access to embryonic/fetal endocrine tissues is limited and genetic manipulation of the tissue not possible. Here, we have demonstrated using hiPSC-derived BLCs that loss of PAX4 had no significant impact on the differentiation trajectory towards endocrine cells but played a role in driving BLCs towards their maturation. Yet, the major challenge of our study remains the derivation of fully functional cells that can be robustly evaluated for defects in GSIS. To circumvent this shortcoming, we made use of EndoC-βH1 cells to evaluate the effect of PAX4 LOF on insulin secretion. When PAX4 is downregulated (shPAX4), EndoC-βH1 cells also exhibited reduced maturity as demonstrated by reduced total insulin content which likely contributes to the impaired GSIS (Fig. 2e-f), replicating the clinical phenotype in variant carriers (Fig. 1a,d, lower insulin secreted in response to glucose). These findings support the key aim of our study to evaluate how PAX4 LOF can impact human beta cell development and in turn increase T2D risk.

3. Page 8 line 172, reduction of HOMA-B was no longer significant after adjusted for age, gender and BMI? What about only BMI, or age, or gender, or any combination? How would the authors interpret the data?

We thank the reviewer for bringing this to our attention. When revisiting the data, we discovered that they were not normally distributed and consequently they should have been log transformed. We have redone our analysis and HOMA-B now remains significant after adjustment for age, sex and BMI.

b	p.Arg192Arg controls n=121 (95% CI)	p.His192 allele(s) carriers (n=62)	p-adj
AIRg*	548.3 (476.3-631.3)	424.6 (349.4-516.0)	0.036
Si*	2.28 (2.09-2.48)	2.64 (2.34-2.98)	0.048
DI*	1247.4 (1078.3-1444.0)	1122.0 (916.9-1372.3)	0.396
HOMA-B*	147.2 (135.0-160.4)	124.7 (110.8-140.6)	0.027

* AIRg, Si, Disposition index and HOMA-B values obtained by back-transforming Lg10 values of the model-estimated means for controls and carriers.

“Carriers of the T2D-risk allele (p.His192) had a decreased acute insulin response to glucose (AIRg, padj=0.036) after adjusting for age, sex and BMI (Fig. 1b). Carriers of the T2D-risk allele p.His192 were more insulin sensitive (Si, p=0.048). There were no differences in disposition index (DI, p=0.396) between the two groups (Fig. 1b). HOMA-B, a measurement of beta cell function, was significantly reduced in p.His192 allele(s) carriers after adjusting for age, sex and BMI (padj=0.027).”

4. In Fig 2, gene expression data was presented as “relative to siNT/scramble”. As detailed in the “Methods” session in Page 41, only 1 housekeeping gene ACTIN was used in gene expression experiments. At least 2 housekeeping genes should be used, and what are the Ct values of PAX4? According to public databases, in general PAX4 gene expression is very low in all tissues.

We thank the reviewer for highlighting this, and we agree that additional housekeeping genes should be tested. **We have now repeated qPCR to include additional housekeeping genes (GAPDH and TUBB3) and the data is displayed below.** In brief, there is no significant difference within the expression of the three housekeeping genes between shScrambled and shPAX4 EndoC-βH1 cells. When normalized to each of the housekeeping gene, PAX4 expression in the shPAX4 EndoC-βH1 cells was consistently lower than that of shScrambled cells.

5. CRISPR guides: what is the deletion? Between the 2 guides? There is no description of the confirmation of deletion (e.g. by Sanger). The authors did not address off target effects either.

We apologize that this information was not adequately highlighted in the manuscript. We have added the following additional information in the results and methods section.

"We designed our loss-of-function strategy to mirror the well-studied Pax4^{-/-} mice where almost all of the functional domains were replaced with a beta galactosidase-neomycin resistance cassette (stated in the methods section). To delete the majority of the paired and homeodomains, sgRNAs were designed targeting exon 2 and exon 5 of PAX4 gene (ENST00000341640.6). Successfully edited clones were detected using genotyping PCR and primers flanking the exon 2 through 5 deletion. Sanger sequencing was used to confirm that two of the independent cell lines had a homozygous deletion for amino acids 64 through 200, whilst the other cell line was compound heterozygous for two premature stop codons at amino acids 61 and 74, respectively."

With respect to the potential off target effects, using Off-Spotter (Pliatsika and Rigoutsos, 2015), we did not detect any genomic sequences with 1 or 2 mismatches from either gRNAs. In addition, none of the predicted off-targets (up to 5 mismatches) are in genes relevant for islet cell biology.

6. PAX4 Arg192His variant (chr7:127253550, rs2233580) is located in Exon 5, the location of Tyr186X is also in Exon 5. CRISPR sgRNA#2 cutting site is around R192, A191 and V190I (theoretically 3-4 bps before or after the PAM seq). What is exactly the deleted region, does it cover the two SNP loci? (Line 259: deletion for AA 64-200?) What is the rationale to design a big deletion fragment in relation to the two GWAS loci?

We previously tried to generate a PAX4 knockout hiPSC line using a single gRNA to generate indels and a premature stop codon. Unfortunately, this strategy was unsuccessful in our hands as the deletion generated an in-frame mutation and resulted in the continued production of a PAX4 protein. Therefore, we decided to adjust our strategy to mimic that of the well-studied Pax4 null mouse model, which deletes a large region of the functional domains of the protein (exons 2 through 5). Two of our lines deleted amino acids 64-200 (so would delete both variants), while the third line introduced premature stop codons that would be predicted to undergo nonsense-mediated decay.

7. It would be better to use PAX4 wt instead of PAX4^{+/+}, which may be confused with insertion mutation.

We thank Reviewer #3 for this suggestion. We have edited the manuscript accordingly using PAX4^{WT/WT} for wild-type cells, and PAX4^{KO/KO} for knockout cells.

8. Page 13, why did the authors choose protocol B over A, and applied protocol B for the rest of the study? PAX4 expression was absent in BLC by protocol B.

This study is a collaboration between two labs that both perform *in vitro* stem cell differentiation using two different protocols. The Gloyne lab uses protocol A (Rezania *et al.*, 2014), and the Teo lab uses protocol B (Pagliuca *et al.*, 2014). At the initiation of the collaboration, both groups were keen to use their own protocols and saw enormous value in comparing their data. We compared the two differentiation protocols, showing that PAX4 expression peaks at the endocrine progenitor stage (Figs. 3c,e). Also, as illustrated in Supplementary Fig. 3 (a bioinformatic comparison of RNA-seq data from Protocols A and B), the two protocols generated highly similar expression profiles over the course of differentiation. As the Teo laboratory had access to the donors, generated the donor-derived hiPSCs, and was using Protocol B, we subsequently decided to combine efforts and perform all experiments with Protocol B after initial comparison experiments. We thank the reviewer for pointing out the lack of clarity. We have now indicated the protocol used for each experiment clearly.

Using hiPSCs as a model for beta cell development, we have demonstrated that PAX4 expression peaks at D20 endocrine progenitor stage (Fig. 3c,e, EP stage) and decreases towards BLC stage. These observations are consistent with the studies reported in rodents whereby rodent Pax4 begins to be expressed at E9.5, peaks between E13.5 and E15.5 to drive endocrine progenitors towards the insulin-producing beta cell fate before its expression decreases towards maturation (Sosa-Pineda *et*

et al., 1997). In fact, shortly after birth there are few Pax4-expressing cells in the mouse (Wang *et al.*, 2004). Therefore, it is an expected observation in which PAX4 expression is lower but not absent in BLCs.

Volcano plots in panel d and f showed completely different gene profiles generated from protocol A and B, except for DENND2D, any follow-up study on DENND2D? In panel f, PAX4 is again absent, not among the top genes that were affected by PAX KD, which makes it further questionable the choice of protocol B.

We thank the reviewer for pointing this out and apologize for the confusion which arises from how our data are presented. It is important to clarify that labelled genes in the volcano plots are the top ten genes that are differentially expressed genes (by p-value) (line 1360-1365). Since PAX4 expression is low in mature beta cells (Sosa-Pineda *et al.*, 1997), it is not unexpected that PAX4 does not appear among one of the most differentially expressed genes.

DENND2D is also a differentially expressed gene in the hiPSCs (Protocol B) and definitive endoderm (Protocol A). As DENND2D expression is altered before activation of PAX4, we hypothesize that the decreased expression of DENND2D is independent of PAX4. Notably, it is not predicted to be an off-target of the gRNAs.

9. Page 14, line 290-292, this is an overstatement: "PAX4 is not required for human beta cell differentiation in vitro. Rather, PAX4 loss-of-function results in derepression of alpha cell genes and a dysregulation of key endocrine maturation genes in hiPSC-derived BLCs." As stated on page 13 Line 283, nonsignificant.

We thank the reviewer for this point. We have toned down on the language and amended it accordingly in the revised manuscript.

"These observations suggest that, unlike in mouse, PAX4 is not essential for human beta cell differentiation in vitro and its loss did not detrimentally impact the differentiation trajectory of hiPSCs towards pancreatic endocrine cells. Rather, PAX4 loss-of-function results in a dysregulation of key endocrine maturation genes in hiPSC-derived BLCs."

10. What is the ethnic origin of the hiPSC lines used in Fig 3? Furthermore, the PAX4 variants were identified in Asian populations, and EndoC cell line is of French origin. (<https://www.ncbi.nlm.nih.gov/pmc/articles/PMC3163974/>). Indeed, many human derived cell lines do carry ethnic-specific signatures. Any thoughts?

The reviewer raises an interesting point regarding the genetic background of cell lines used in research. In our study we have been able to capitalize on numerous resources providing the opportunity to evaluate the role of PAX4 in cell lines of different ancestries showing a common cellular phenotype across them. The diabetes associated variants have been found in the Singapore population and we have modeled them both in patients and in an isogenic European hiPSC model and the immortalized fetal derived EndoC-βH1 cell line with consistent observations. The use of different models is a strength of our study.

11. What is the consequence of increased GCG gene expression in translation? In protein expression detection? Fig 4 IF stained for GCG. The mammalian proglucagon gene (*Gcg*) encodes three glucagon like sequences, glucagon, glucagon-like peptide-1 (GLP-1), and glucagon-like peptide-2. Have the authors looked at other potential translational products?

We thank the reviewer for raising this point. The loss of beta cell identity and the acquisition of polyhormonal cells have been reported in the pancreatic islets of individuals with diabetes (Cinti *et al.*, 2016; Talchai *et al.*, 2012). The transdifferentiation of metabolically stressed beta cells to express GCG has been proposed by several groups as a mechanism underlying beta cell failure in T2D (Brereton *et al.*, 2014; Talchai *et al.*, 2012; Wang *et al.*, 2014). The GCG antibody was carefully selected for the identification of alpha cells (Santa cruz, sc-7780, used by Kroon *et al.*, 2008; Ackermann *et al.*, 2018; Dooley *et al.*, 2016). GLP-1 and GLP-2 are predominantly secreted by the enteroendocrine cells (Iakoubov *et al.*, 2007). Therefore, we have not considered looking into the expression of GLP-1 and GLP-2 in our hiPSC-derived BLCs. However, in response to the reviewer's

concern, we did measure the GLP-1 levels during an OGTT in carriers of the p.Arg192His variant (Supplementary Fig. 1f-h, see below). There was no significant difference between the carriers and non-carriers of the variant.

Clinical assessment of GLP-1 in carriers of p.Arg192His PAX4 variant during two-hour OGTT. (F) Fasting, (G) 20-min and (H) AUC of GLP-1 secreted in subjects.

12. Line 320-322, this is an overstatement. In his192 allele carriers, no change in PAX4 or INS, GCG, SST transcription was detected in this study. Line 322, “negatively affecting endocrine cell differentiation”, this is a bold conclusion based on only GCG staining, without further data to back up endocrine cell differentiation.

We thank the reviewer for highlighting this. With new flow cytometry characterization data that we have incorporated as Supplementary Fig. 2 and 3, we have revised our manuscript as follows:

“Heterozygous and homozygous carriers of the PAX4 p.His192 allele had no measurable differences in INS, GCG, or SST gene expression (Fig. 4b-i).” There were no significant differences in the cell populations expressing INS, GCG, or SST at the EP or BLC stages (new Supplementary on FACS characterization). *“Together, these data suggest that both PAX4 alleles result in a loss-of-function due to reduced PAX4 gene dosage and/or altered PAX4 transcriptional activity, negatively affecting pancreatic beta cell differentiation.”*

13. PAX4 primers: hPax4F5 (AGGACACGGTGAGGGTCTGGT) shows no blat result (please correct me if I was wrong); hPax4R5 (CAGTGGTTCAGGGCAGGCA) is located in Exon 8. Tyf186X is supposed to introduce premature stop codon at AA186 (line 333-334). How is the amplification region in relation to the variants?

The primers were designed against the transcript (not the genome) and the hPax4F5 spans the exon 5/6 boundary. The primer pair amplifies a 191-bp sequence that is 3' to both variants (see image below).

Homo sapiens paired box 4 (PAX4), transcript variant 1, mRNA
 >NM_001366110.1:407-1462 Homo sapiens paired box 4 (PAX4), transcript variant 1, mRNA

```

ATGCATCAGGACGGGATCAGCAGCATGAACCAGCTTGGGGGGCTCTTTGTGAATGGCCGGCCCTGCCTCTGGATAC
CCGGCAGCAGATTGTGCGGCTAGCAGTCAGTGAATGCGGCCCTGTGACATCTCACGGATCCTAAGGTATCTAATG
GCTGTGTGAGCAAGATCCTAGGGCGTTACTACCCACAGGTGCTTGGAGCCAAAGGGCATTGGGGGAAGCAAGCC
ACGGCTGGCTACACCCCTGTGGTGGCTCGAATTGCCAGCTGAAGGGTGAGTGTCCAGCCCTCTTTGCCTGGGAAA
TCCAACGCCAGCTTTGTGCTGAAGGGCTTGCACCCAGGACAAGACTCCAGTGTCTCTCCATCAACCGAGTCTGTC
GGCATTACAGGAGGACAGGGACTACCGTGCACACGGCTCAGTCCAGCTGTTTGGCTCCAGCTGTCTCTACT
CCCCATAGTGGCTCGAGACTCCCCGGGTACCCACCCAGGGACCGGCCACCGAATCGGACTATCTTCTCCCCAAG
CCAAGCAGAGGCACTGGAGAAAGAGTTCCAGCTGGGCAGTAT(GTA)CCTGATTCAGTGGCCCGTGGAAAGCTGGC
TACTGCCACCTCTCTGCCTGAGGACACGGTGAGGGCTGGTTTTTCCAACAGAAGAGCCAAATGGCGTCGGCAAGAGA
AGTCAAGTGGGAAATGACAGTGCACAGGTGCTTCCAGGGGCTGACTGTACCAAGGTTGCCCCAGGAATCATCTCT
GCACAGCAGTCCCCTGGCAGTGTGCCACACAGCAGCC(TGCCCTGCCCTGGAAACCACTGGGTCCTCTGCTATCAGTG
TGCTGGGCAACAGCACCAGAAAGGTGTCTGAGTGACACCCACCTAAAGCCTGTCTCAAGCCCTGCTGGGGCCACTT
GCCCCACAGCCGAATTCCCTGGACTCAGGACTGCTTTCCTTCCCTTCTCCACTGTCACTGTCCAGTCTT
AGTGGCTCTAGGCCCTGCTCTGGCTGGCTGCCACTACTGTATGGCTTGGAAATGA
hPax4F5 primer
hPax4R5 primer
  
```

14. What is the epitope of PAX4 antibody?

The PAX4 antibody (RnD, AF2614) is polyclonal. Unfortunately, the binding epitope(s) of the antibody is proprietary information not revealed by RnD. During our evaluation of the specificity of the antibody we established that the AD2614 antibody is able to detect both the p.H192 and p.X186 recombinant proteins. As a V5-tag is located at the 3' end of PAX4 in the expression plasmid, p.186X introduced a premature stop codon that prevented the expression of V5. This is reflected in the missing band observed for V5 under the p.X186 lane (refer to the figure below). This would support the location of the epitope being in the N-terminus of the protein.

Image of western blot of WT PAX4 (WT), p.His192 and p.X186 proteins overexpressed in AD293 cells. PAX4 and V5 antibodies were used to assess protein expression 24 h post transfection.

15. ChIP should be performed to demonstrate interaction between PAX4 and gene promoter regions.

We would like to perform ChIP for PAX4 but unfortunately our evaluation of the specificity of the currently available PAX4 antibodies has shown that they are not suitable for ChIP. Hence, we respectfully respond that this is currently out of scope for this manuscript. We have sought to demonstrate an interaction between PAX4 and gene promoter regions through the use of luciferase assays, which are more sensitive in the detection of transcriptional regulatory activity.

16. Fig 5h shows that PAX4 kd had no effect on INS promoter activity, how to explain Fig 2g where it shows increased INS gene expression by PAX kd?

We thank the reviewer for highlighting this. PAX4 is not the only transcription factor that regulates the expression of the *INS* gene. We speculate that the non-significant increase in *INS* expression in some samples could be a compensatory effect by other beta cell determinant transcription factors in an attempt to re-navigate the cells towards beta cell fate - especially in earlier passages following PAX4

knockdown. To determine if shPAX4 does impact the expression of *INS* gene in EndoC-βH1 cells, we performed additional experiments (now with stable shPAX4 cells of older passages) and presented the updated data as Fig. 2g. These data remain consistent with our previous finding that *INS* expression is not significantly different following shPAX4-mediated knockdown (updated Fig. 2g).

In Fig. 5d, overexpression of WT PAX4 repressed *INS* promoter in EndoC-βH1 cells. When PAX4 was knocked down, there was no significant difference in luciferase activity at the *INS* promoter (Fig. 5g).

We speculate that the inconsistency in Fig. 5d and Fig. 2g is a consequence of different PAX4 dosages within the EndoC-βH1 cells. In Fig. 5d, the exogenous overexpression of PAX4 protein elicited a strong repression effect. On the other hand, in Fig. 2g, knocking down endogenous PAX4 using shRNA results in no significant difference in *INS* gene expression.

17. Fig 6e and g, there is no significant difference between the variants. Should be carefully interpreted in data presentation.

We thank the reviewer for highlighting this. The difference in the overall glycolytic function/mitochondrial respiration is not significant when comparing the endocrine progenitor cells carrying wild-type or variant PAX4 allele(s) (Fig. 6e,g). However, when we measure specific parameters of the glycolysis and mitochondrial stress tests, we did observe significant differences between PAX4 genotypes. In particular, there was significant reduction in non-glycolytic acidification, glycolytic capacity, and glycolytic reserve (Fig. 6f), consistent with PAX4 variants reducing key parameters of glycolytic flux. With respect to mitochondrial measurements, PAX4 variants had increased basal respiration, non-mitochondrial O₂ consumption, ATP production, and H⁺ (proton) leak (Fig. 6h), consistent with a bioenergetic switch from glycolysis to oxidative phosphorylation. We have updated the results section to make it clear that the differences in glycolysis and mitochondrial stress tests were only found in the aforementioned key parameters (line 407-419):

“To further assess a potential defect in metabolism, we performed a Seahorse XFe96 Glycolysis Stress Test. The glycolysis stress test showed that EPs carrying one or two p.His192 risk alleles had lower measurements of glycolytic function, including non-glycolytic acidification, glycolytic capacity, and

glycolytic reserve (Fig. 6e-f). In addition, EPs carrying the p.Tyr186X risk allele had decreased glycolytic reserve and non-glycolytic acidification (Fig. 6f). Next, we hypothesized that EPs would seek alternative metabolic processes to compensate for the reduction in energy production, such as oxidative phosphorylation through mitochondrial respiration. Mitochondrial function was measured via oxygen consumption in EPs using the Seahorse XFe96 analyzer. There was an increase in oxidative phosphorylation activity in EPs harboring PAX4 variants (Fig. 6g), including basal respiration, non-mitochondrial O₂ consumption, ATP production, and H⁺ (proton) leak (Fig. 6h). Overall, EPs carrying PAX4 diabetes risk alleles demonstrated a bioenergetic switch from glycolysis to oxidative phosphorylation.”

18. Line 411, these changes are not significant in Supplementary Fig6

We thank the reviewer for this point. In our RNA-seq data (Fig. 6d and Supplementary Fig. 6), we observed endocrine cells carrying PAX4 variants had altered metabolic signatures, suggesting that the cells were under elevated oxidative stress (Fig. 6d). This prompted us to look into the metabolic functions of these cells to evaluate if the metabolic stress could be one causal factor accounting for the reduced total insulin content in PAX4 variant carrying-BLCs (Fig. 4n). However, when we treated the cells (from EP to BLC) with the antioxidant NAC, there was no rescue in the total insulin content as highlighted by the reviewer (Supplementary Fig. 6), suggesting that the unique metabolic signature is not causal. Nonetheless, our data suggest that EPs carrying PAX4 variant allele(s) had a bioenergetic switch from glycolysis to oxidative phosphorylation (Fig. 6f,h). Therefore, we speculated (under discussion, line 426-429) that “*metabolic signature observed in our donor-derived hiPSC model reflects the physiological status of the EPs rather than being the immediate cause for the dysregulation of beta cell development and maturation*”. We hope this clarifies.

19. Fig 7f and g, Supplementary Fig 7, wt is missing in all experiments for control

We thank the reviewer for pointing this out. Regarding rescue experiments, we did not include the wild-type cell lines in our experiments. hiPSCs are highly heterogeneous in nature even across cell lines from the same donor. It is also noted that, from our experience with handling numerous donor-derived hiPSC lines, heterogeneity could be introduced during the differentiation process even within the same cell line. Hence, our experiments (Fig. 7c) were **carefully designed to make use of the gene-corrected lines (wild-type) versus uncorrected isogenic pairings for data comparison**. This will not change the overall message but tremendously helped in the reduction of background noise from genetic differences and from the CRISPR genome editing process (i.e. clonal expansion and increased passages). **We have now incorporated the explanation into the manuscript for better clarity** (line 433-446):

“Next, we used CRISPR-Cas9 to correct the donor-derived hiPSC lines and to generate PAX4 variant isogenic hiPSC lines. We designed sgRNA#3 to target the donor-derived homozygous p.His192His line and provided the homology-directed repair (HDR) template to correct the rs2233580 T2D-risk allele (Fig. 7a). The II-11 donor-derived hiPSC line that is heterozygous for a GTA duplication was corrected with sgRNA#4 and an HDR template (Fig. 7b). From the CRISPR-Cas9 genome editing pipeline, we generated two corrected p.Arg192Arg non-risk and two uncorrected p.His192His hiPSC lines (Fig. 7c). From the II-11 donor-derived line, four corrected p.Tyr186Tyr and four uncorrected p.Tyr186X hiPSC lines were derived (Fig. 7c). To control for the different genetic backgrounds of donors, we generated isogenic SB Ad3.1 hiPSCs that are homozygous for the risk alleles (p.His192His and p.X186X). All the corrected and uncorrected donor derived hiPSC lines and the isogenic PAX4 variant hiPSC lines were differentiated towards BLCs using Protocol B, followed by RNA-seq analyses and assessment of total insulin content (Fig. 7c).”

REVIEWER COMMENTS

Reviewer #1 (Remarks to the Author):

Even if the authors have done their best to respond to my comments, I unfortunately still remain confused by some of the data. Moreover, at this reading I identified some new points that I think should be corrected.

1. The authors use the abbreviation BLC (beta like cells) throughout. However, this is not a good term, in fact it is misleading since only <50% of the cells represent beta (like) cells. The term SC-islet (stem cell derived islet) is much more appropriate, and it has been increasingly used by other groups. Why not aim for unified nomenclature?

2. In my original comment #3, I made the simple point that since the mutant iPSC lines (both position 192 and 186) had lower insulin content and increased proportion of INS/GCG cells, one would expect to see this also in the PAX4 KO cells, but the data were not shown. This is a simple phenotype and if it is associated with PAX4 LOF, then it should be evident also in the KO cells. In the rebuttal, the authors now show that INS content is not reduced in the KO, and they do not show the KO effect on INS/GCG double positivity, even though this is what I requested. If the mutations lead to loss of function, it is difficult to understand why the mutations would lead to a reduced insulin content but the KO not. The authors explain that “due to the heterogeneous nature of hiPSC models, we were not able to detect any significant difference when comparing the total insulin content between the control and PAX4 KO lines”. I do not find this explanation acceptable. The comparison should be done between the parental line and the isogenic KO line. The “heterogeneous nature” of the lines should then not be a problem. Overall, it appears that the authors try to hide these discrepancies between the KO phenotype and the mutation phenotype. Perhaps it would be better to openly discuss the problems and limitations of the models.

3. I found it very difficult to get a clear message from Fig 7 which deals with the mutation corrected (or engineered) iPSC lines. Even after the detailed rebuttal, several problems remain:

- no data is presented to verify that the genome editing was actually successful. It is common practice to show Sanger sequencing of the corrected/introduced healthy/mutant alleles. Similar essential evidence should be provided for all genome edited lines used in these studies.

- the labelling in panels d-e remains confusing. The donor-derived mutant and corrected lines are of course also isogenic, not only the standard iPSCs with engineered mutations. It would be logical to label them as “donor-derived” and “engineered”

- It is virtually impossible to grasp any meaningful results from the long list of genes in the heatmaps in panels d-e. There are many examples of lower and higher beta-cell gene expression in the mutant as compared to corrected. The presentation should be improved to provide a clear message. If no clear message exists, then it would be better to not show these results at all.

Reviewer #2 (Remarks to the Author):

The authors have appropriately addressed all concerns raised by this reviewer.

The study will be of great interest to the readership of Nature Communication.

Benoit Gauthier

Reviewer #3 (Remarks to the Author):

The authors answered the questions well with additional experiments and convincing data.

One question remains on "5. CRISPR guides: what is the deletion? Between the 2 guides? There is no description of the

confirmation of deletion (e.g. by Sanger). The authors did not address off target effects either. "

The authors answered "With respect to the potential off target effects, using Off-Spotter (Pliatsika and Rigoutsos, 2015), we

did not detect any genomic sequences with 1 or 2 mismatches from either gRNAs. In addition, none of the predicted off-targets (up to 5 mismatches) are in genes relevant for islet cell biology. "

It is not enough to assume that the off-targets are not relevant because they are not relevant for islet cell biology. Sanger seq on top predicted off-target regions should be performed.

Point-by-point response to the reviewers' comments

Reviewer #1 (Remarks to the Author):

Even if the authors have done their best to respond to my comments, I unfortunately still remain confused by some of the data. Moreover, at this reading I identified some new points that I think should be corrected.

1. The authors use the abbreviation BLC (beta like cells) throughout. However, this is not a good term, in fact it is misleading since only <50% of the cells represent beta (like) cells. The term SC-islet (stem cell derived islet) is much more appropriate, and it has been increasingly used by other groups. Why not aim for unified nomenclature?

We thank the reviewer for the comment and agree that SC-islet is indeed more appropriate. We have now edited our manuscript and figures accordingly.

2. In my original comment #3, I made the simple point that since the mutant iPSC lines (both position 192 and 186) had lower insulin content and increased proportion of INS/GCG cells, one would expect to see this also in the PAX4 KO cells, but the data were not shown. This is a simple phenotype and if it is associated with PAX4 LOF, then it should be evident also in the KO cells. In the rebuttal, the authors now show that INS content is not reduced in the KO, and they do not show the KO effect on INS/GCG double positivity, even though this is what I requested.

We apologize for misunderstanding your original point as it is an important one. We have now performed additional experiments to quantify the number of CPEP/GCG double positive cells in PAX4 KO lines.

Sham control SC-islet cells (left panel) had significantly lower differentiation propensity than PAX4-KO (right panel). White arrow pointing to cell(s) co-expressing CPEP and GCG. Scale bar: 50 μ m.

While it appears that there is a robust increase in the number of co-expressing cells in the PAX4 KO lines, we have decided to not include the data in the manuscript for the following reasons:

- 1) Using protocol B, we routinely measure >20% C-PEP+ cells in SC-islets (Figure 4I).
- 2) For this particular differentiation we detected very few C-PEP+ cells in the PAX WT cells (sham panel on the left), suggesting poor differentiation of these PAX WT cells towards SC-islets. We had previously measured ~20% C-PEP+ cells in SC-islets derived from PAX4 WT cells (Supplementary Fig. 4c).

- 3) It is therefore not possible to conclude with current data that the increase in CPEP/GCG double positive cells is due exclusively to the KO genotype and not reflective of a poor differentiation of the WT cells to form SC-islets.

While our attempt to address the important question of whether PAX4 KO cells have increased C-PEP/GCG double positive cells was unsuccessful, we hope the reviewer appreciates our efforts and agrees with our decision to not include data that may not accurately reflect the biology.

If the mutations lead to loss of function, it is difficult to understand why the mutations would lead to a reduced insulin content but the KO not. The authors explain that “due to the heterogeneous nature of hiPSC models, we were not able to detect any significant difference when comparing the total insulin content between the control and PAX4 KO lines”. I do not find this explanation acceptable. The comparison should be done between the parental line and the isogenic KO line. The “heterogeneous nature” of the lines should then not be a problem. Overall, it appears that the authors try to hide these discrepancies between the KO phenotype and the mutation phenotype. Perhaps it would be better to openly discuss the problems and limitations of the models.

We apologize for our unsatisfactory answer to your original question and we certainly did not intend to hide any discrepancies in our data. As we have discussed in our manuscript, the in vitro differentiation model, while a powerful one, does suffer from some challenges, including variability, heterogeneity, and immaturity. We have tried to overcome these limitations by complementing our hiPSC models with clinical data and assays in an alternative model, EndoC-βH1 cells. Despite this, as the reviewer noted, there is a discrepancy between the KO phenotype and the variants. There are several possible explanations for this:

- 1) The reduced insulin content in the donor-derived cells is independent of the PAX4 genotype. This is unlikely to be true as the content phenotype was reversed by gene correction (Fig. 7).
- 2) There could be genetic compensation that “rescues” the insulin content phenotype in the homozygous null line, as has been shown in zebrafish and mouse models [PMID: 3094447]. This is a possibility that we cannot rule out; although, you would expect to only see this in the compound heterozygous line with premature stop codons.
- 3) The SB cell lines (and therefore the KO model) are prone to more variability using Protocol B than the donor-derived cells. As was illustrated above, we did have more difficulty getting consistency among our differentiations from the SB Ad3.1 cell lines using Protocol B. If we were to approach this project again, we would have spent some more time optimizing the protocol to improve the consistency.

Despite the inconsistencies between the KO and variants, we are confident that the variants are loss-of-function because a) knockdown of PAX4 in EndoC-BH1 cells reduces insulin content, b) we determined that the molecular consequence of the PAX4 variants is reduced function (Fig. 5 luciferase assays), and c) we can correct the content phenotype in donor-derived cells through PAX4 gene correction.

3. I found it very difficult to get a clear message from Fig 7 which deals with the mutation corrected (or engineered) iPSC lines. Even after the detailed rebuttal, several problems remain:
- no data is presented to verify that the genome editing was actually successful. It is common practice to show Sanger sequencing of the corrected/introduced healthy/mutant alleles. Similar essential evidence should be provided for all genome edited lines used in these studies.

We have now added Sanger sequencing to Fig. 3b (PAX4 KO hiPSC lines) and Fig. 7a-b (Correction of patient-derived lines).

Fig. 3b: Sanger sequencing for exon 2 and exon 5 of compound heterozygous PAX4^{KO/KO} and homozygous deletion PAX4^{KO/KO} hiPSC lines.

Fig. 7a-b: (a-b) CRISPR-Cas9 gene editing strategy to correct the (a) p.His192His genotype to p.Arg192Arg (rs2233580) and (b) p.Tyr186X genotype to p.Tyr186Tyr (GTA duplication). Protospacer adjacent motif (PAM) sequence is bolded, and the respective variant and duplication are labelled in green. Representative Sanger sequencing results for the corrected allele are depicted with nucleotide changes labelled in red.

- the labelling in panels d-e remains confusing. The donor-derived mutant and corrected lines are of course also isogenic, not only the standard iPSCs with engineered mutations. It would be logical to label them as “donor-derived” and “engineered”

Thank you for the suggestion. We updated the figure and manuscript accordingly.

- It is virtually impossible to grasp any meaningful results from the long list of genes in the heatmaps in panels d-e. There are many examples of lower and higher beta-cell gene expression in the mutant as compared to corrected. The presentation should be improved to provide a clear message. If no clear message exists, then it would be better to not show these results at all.

We thank the reviewer for their suggestion to consider removing the heatmaps in Fig. 7. By removing them, we were able to include the requested Sanger sequencing data confirming the CRISPR-Cas9 correction. The RNA-seq and DEG lists can still be found in Tables S4, S5, and S6 and the data will be publicly available. The important message, which the updated figure better highlights, is that CRISPR-Cas9 correction improved total insulin content in PAX4 patient-derived SC-islets.

Reviewer #2 (Remarks to the Author):

The authors have appropriately addressed all concerns raised by this reviewer. The study will be of great interest to the readership of Nature Communication.
Benoit Gauthier

Thank you for your helpful feedback.

Reviewer #3 (Remarks to the Author):

The authors answered the questions well with additional experiments and convincing data.

One question remains on "5. CRISPR guides: what is the deletion? Between the 2 guides? There is no description of the confirmation of deletion (e.g. by Sanger). The authors did not address off target effects either."

We have now added the Sanger sequencing data to Fig. 3b and updated the text as follows:

“Using Sanger sequencing, we confirmed that two of the independent cell lines had a homozygous deletion for amino acids 64 through 200, whilst the other cell line was compound heterozygous for two premature stop codons at amino acids 61 and 74, respectively (Fig. 3b).”

The authors answered "With respect to the potential off target effects, using Off-Spotter (Pliatsika and Rigoutsos, 2015), we did not detect any genomic sequences with 1 or 2 mismatches from either gRNAs. In addition, none of the predicted off-targets (up to 5 mismatches) are in genes relevant for islet cell biology. "

It is not enough to assume that the off-targets are not relevant because they are not relevant for islet cell biology. Sanger seq on top predicted off-target regions should be performed.

Thank you for the suggestion. Using the results from Off-Spotter, we designed primers and sequenced the top predicted off-target sites for each gRNA and their respective hiPSC lines. We have added the following sentence to the methods section:

“Primers were designed to amplify and Sanger sequence the top predicted off-target sites for each of the four gRNAs. No mutations were detected in any of the sequenced potential off-targets sites.”

We have included the Sanger sequencing results that support this statement below:

- 1) Exon 2 gRNA: CTAGGGCGTTACTACCGCAC [+ strand]
 - a. PSD4 [4 mismatches] CCAAGGCGTTGCCACCGCAC
 - i. No SNPs detected for three WT and three KO cell lines.

- 2) Exon 5 gRNA: TATCCTGATTCAAGTGGCCCG
 - a. PSM2 [4 mismatches] TAGTGTGACTCAGTGGCCCG
 - i. No SNPs detected for three WT and three KO cell lines.

ii. No SNPs detected for two p.Arg192His cell lines.

iii. No SNPs detected for one p.Tyr186X cell line.

b. ZNF704 [4 mismatches] TATGCTGCTGCTGTGGCCCG

i. No SNPs detected for three WT and three KO cell lines.

ii. No SNPs detected for two p.Arg192His cell lines.

iii. No SNPs detected for on p.Tyr186X cell line.

- c. PANK4 [4 mismatches] TATCCTGGCTGGGTGGCCCG
 - i. No SNPs detected for three WT and three KO cell lines.

- ii. No SNPs detected for two p.Arg192His cell lines.

- iii. No SNPs detected for on p.Tyr186X cell line.

- 3) sgRNA91: ATCTCCGCAGAGTTCCAGCG
 - a. RBPJL [4 mismatches] CTCTCCGTGGAGCTCCAGCG
 - i. No SNPs detected for patient-derived p.Tyr186X (two corrected, two uncorrected, and the parental) cell lines.

4) sgRNA59: GGCAGTAGCCAGCTTTCCAT

a. CHUK [4 mismatches] AACTGTAGCCAGTTTTCCAT

- i. There is one SNP downstream of the potential off-target site. The edited lines are heterozygous (three corrected, two uncorrected, and the parental), as is the parental, unedited patient-derived cell line. Therefore, this SNP did not result from the genome editing process.

b. GRM7 [4 mismatches] AGCAGAAGTCAGTTTTCCAT

- i. No SNPs detected for patient-derived p.Arg192His (three corrected and two uncorrected) cell lines.

REVIEWERS' COMMENTS

Reviewer #1 (Remarks to the Author):

I want to thank the authors for their open and frank response to my remaining concerns. It is evident that there are certain problems in the SC-islet differentiation. Consequently, it was a wise decision to leave out some data coming from experiments that did not work efficiently. However, overall there is sufficient evidence in this paper to justify the link between decreased Pax4 expression and lower beta-cell functionality which is consistent with the association of Pax4 LOF variants with the risk of diabetes. I do not have any further requests.

Reviewer #3 (Remarks to the Author):

No further comments, the reviewer thanks the authors for their efforts.

Point-by-point response (1st response letter)

Reviewer #1 (Remarks to the Author):

This study by Lau&Krentz et al. contains comprehensive clinical and experimental studies that overall support the title of the paper. The results are mostly clear and well described. However, there are several problems related with Fig.7 (the rescue experiments by mutation correction).

We thank Reviewer #1 for his/her support of our manuscript. We will address his/her concerns relating to Fig. 7 below. Thank you.

Major comments:

1. Differentiation of iPSC into islet-like cells: Two well-known but related protocols were used (refs 22 and 25). The study comes from two collaborating laboratories. Does this mean that one lab used protocol A and the other one B, or were both protocols established at both sites, or only one site? Was this planned and coordinated, or were the results of two labs combined retrospectively? It would be fair to make this clear.

This study is a close collaborative effort planned and coordinated between two laboratories. Anna Gloyn's laboratory used protocol A whereas Adrian Teo's laboratory used protocol B. We compared the two differentiation protocols, showing that *PAX4* expression peaks on endocrine progenitor stage (Figs. 3c and 3e). Also, as illustrated in Extended Data Fig. 3 (a bioinformatic comparison of RNA-Seq data from Protocols A and B), the two protocols generated highly similar expression profiles over the course of differentiation.

As Adrian Teo's laboratory had access to the donors, generated the donor-derived hiPSCs, and was using Protocol B, we subsequently decided to combine efforts and perform all experiments with Protocol B after initial comparison experiments. We thank the reviewer for pointing out the lack of clarity, and have now indicated the protocol used for each experiment clearly.

Regarding the concerns of the chosen protocols, this mostly requires additional explanations on how our team came together to collaborate, beginning with Protocols A and B, which eventually culminated in most experiments being performed using Protocol B. It does not appear that there is any requirement for additional experiments.

2. It would also be important to present at least the elementary general characterization of the differentiation efficiencies using the two protocols in the Supplementary materials. How do the yields of Pdx1/Nkx6.1 endocrine progenitors at stage 4 compare? What about the yield of beta and alpha cells?

Using RNA-Seq and qPCR assessment, we have confirmed the two widely-used protocols (A and B) to generate highly similar transcriptomic profiles over the course of the pancreatic beta cell differentiation (Refer to Extended Data Fig. 3. Inserted below for easier reference). The focus of this study has been to use the protocols to study the biology of Pax4 during differentiation and development, but not to compare how the two published protocols performed against one another. Therefore, we seek the reviewer's understanding that the comparison of the two protocols for differentiation efficiencies is beyond the scope of our project aims.

Extended Data Fig. 3 | *PAX4*^{-/-} and variant lines have similar repression of pluripotency and activation of endocrine genes as wildtype and corrected lines. (a) Key pluripotency and (b) endocrine progenitor gene expression in hiPSCs, DE cells, EPs, and BLCs differentiated using Protocols A and B of *PAX4* wildtype (*PAX4*^{+/+}), knockout (*PAX4*^{-/-}), *PAX4* variants (p.His192His and p.Tyr186X), and corrected (p.Arg192Arg and p.Tyr186Tyr) donor-derived hiPSC lines.

Additionally, we have now carefully characterized the endocrine progenitors and beta-like cells differentiated from donor-derived hiPSCs using Protocol B. The efficiencies of generating beta-like cells (30-40% - INS, PDX1), alpha cells (~10%, GCG), and delta cells (15-20%, SST) were highly comparable to Pagliuca et al., 2014. While the FACS assessment has demonstrated no significant difference in the cell populations, our immunostaining data revealed elevated expression of polyhormonal INS⁺/GCG⁺ cells (Figs. 4j-m) which led us to uncover that these cells are less mature and expressed lower total insulin content (Fig. 4n).

That said, we will perform additional characterization of the beta-like cells using the current protocols.

FACS assessment on key pancreatic beta cell development markers to characterize endocrine progenitors and beta-like cells differentiated from donor-derived hiPSC lines. hiPSC lines derived from two wildtype, two p.Arg192His, two p.His192His and one p.Tyr186X donors were subjected to differentiation. Percentage of cells stained positive for PAX4, PDX1, INS, GCG and SST on D20 endocrine progenitor (EP) stage and D35 beta-like cell (BLC) stage was consolidated and represented in graphs. At least n=4 experiments were performed.

And insulin secretion in response to glucose plus other secretagogues (like GLP1, KCl)? It is also very important to clearly indicate which protocol was used for each result presented (missing at least for Fig. 7).

Most of the subsequent experiments were performed using Protocol B, Fig. 3 onwards.

Unfortunately, we have indeed stimulated the hiPSC-derived BLCs with high glucose but they were non-functional. To evaluate the importance of PAX4 in beta cell function, we therefore made use of a well-reported human beta cell line (EndoC- β H1) to supplement our findings in hiPSC-derived BLCs.

We understand the request for functional data from hiPSC-derived BLCs. We propose to differentiate a select number of hiPSCs (PAX4-KO versus control; p.His192His donor-derived hiPSCs versus corrected lines; and p.Tyr186X donor-derived versus corrected lines) into BLCs using Protocol B and assess their response to glucose and KCl.

While we will attempt additional experiments with selected hiPSC lines (PAX4 knockout; key patient lines that were genome corrected), getting fully functional beta cells is known to be difficult, variable and possibly cell line-dependent. We propose to include conditions such as stimulated with KCl in case these beta-like cells do not prove to be glucose-responsive. We seek the agreement of the editor with this proposed approach.

3. Fig. 4 k-n shows clearly the important effects of the LOF mutations on the insulin content and presence of polyhormonal cells. Why is this information lacking for the Pax4 KO lines? This is an important part of the basic characterization that is missing (see point 3), and it would be particularly

important to include this characterization for the Pax4 KO lines in comparison with the isogenic WT controls.

We thank the reviewer for this comment. We agree that the effect of the loss of PAX4 on insulin content and presence of polyhormonal cells is an important characterization to support our main findings. We will proceed to perform more experiments to assess the insulin content and presence of polyhormonal cells in PAX4-KO lines and add the findings to the next version. (3 Sham + 3 PAX4-KO lines, SB origin).

4. An improved protocol has recently been published by Balboa et al (PMID 35241836), showing further maturation and the loss of polyhormonality during extended final stage. It would be interesting to see the impact of Pax4 LOF (both the KO and the mutations studied) on the C-Pep/GCG double positive cells after extended maturation.

We thank Reviewer #1 for this suggestion. As can be appreciated, we have already adopted many hiPSC lines and two widely-used protocols in this study. The focus of this study has been to use the protocols to study the biology of Pax4 during differentiation and development, but not to compare how the two published protocols performed against one another.

Although it could be interesting to add another protocol, this is not trivial and we feel is outside the scope of the current study. If we were to establish a new protocol based on Balboa et al. in our labs, it would take 2-3 months per experiment and we would have to first demonstrate that the protocol worked in our hands. Since the purpose of this study is not to compare differentiation protocols, we feel this is out of scope.

We seek the editor and reviewer's understanding that it is out of scope to establish another protocol from scratch (Balboa et al. takes up to 13 weeks of differentiation) and apply it to so many lines. We seek the agreement of the editor on this. Thank you very much.

Of note, we have also begun to make several of these lines available to the wider research community (<https://hpscereg.eu/search?q=singapore>; <https://skip.stemcellinformatics.org/en/>).

5. The mutation correction results presented in Fig. 7 are very difficult to understand:
- The genotypes of the lines are expressed confusingly. Clearly, in panel a, it is the p.Arg192His mutation that is corrected (and not His192His)? Similar confusing labels can be found also in other panels.

We apologise for the confusion. It is indeed H192H (mutant: CAT) that is being corrected into R192R (wildtype: CGT). We will check through and indicate the wildtype genotype in the figure, to improve the clarity in all panels.

- Panels d and e: Labelling of the lanes is very confusing. I believe that in (d) the same labelling applies for both two lanes on the left and right (uncorrected vs corrected), and in (e) the same label is supposed to cover 4 lanes (?). This is really difficult to understand, so please clarify.
- Furthermore, I simply do not understand what the "isogenic BLC" stand for? The term "isogenic" in my understanding should mean that the cells have the same background genotype, i.e- are derived from the same donor. So, what are these? And how should one interpret the result when the color scale is completely different from the "donor-derived BLC". I find it hard to identify obvious patterns in the expression that would make sense (this applies also the replicate clones particularly in panel d).

Each column represents one cell line derived from a donor. Isogenic BLC refers to cell line derived from SB (StemBanc) hPSCs, with the various mutant genotype introduced using CRISPR editing, hence the lines share isogenic background.

- Based on a lot of evidence provided in the manuscript, a hallmark of Pax4 LOF should be an increased alpha-cell expression pattern, represented by higher ARX expression. However, in panels d-e it appears that ARX is consistently higher in the corrected than in uncorrected cells?

We will address this in the revised manuscript. Thank you.

Minor:

1. Title: The title could be a bit more specific; instead of “endocrine cell”, rather “islet cell” or “pancreatic endocrine cell”; and instead of “influences”, rather “increases”). As a result, perhaps an appropriate title could be: “Pax4 loss of function increases diabetes risk by altering human pancreatic endocrine cell development”.

We have now amended the title as suggested by Reviewer #1.

2. Line 195: HbA1c was 7.1%/8.7 mmol/L. If the intention is to express the HbA1c in molar units, I suppose this should be around 54 mmol/mol. But perhaps the authors mean that the HbA1c of 7.1% corresponded to a mean b-glucose level of 8.7 mmol/L. Please clarify.

We have now clarified this in the revised manuscript.

3. Fig. 1f: In the table the glucose values are obviously in mmol/L, but the column title has mg/dL.

We thank Reviewer #1 for pointing this out. We have now corrected the table title accordingly.

Reviewer #2 (Remarks to the Author):

The study by Lau/Krentz and colleagues expand our current knowledge on the functional role of the transcription factor PAX4 in human islet development, and function, and how mutations, including a new one identified herein in a Singapore family (p.X186), may contribute to type 2 diabetes risk. Using both clinical and experimental approaches that include the generation of iPSCs-derived beta-like cells obtained from individuals with this new mutation combined with CRISP and RNAseq technologies, the authors demonstrate that PAX4 is essential to repress the alpha genetic program which favors beta cell identity hallmarked by insulin content and functional insulin secretion. The study is of interest with a massive amount of data that will be resourceful for the scientific community.

We thank Reviewer #2 for the positive comments, interest and support for our study.

General comments:

Mouse studies have shown that overexpression of the Pax4 diabetes-linked mutant variant R129W in adult mice beta cells sensitize to stress and apoptosis (Hu He et al., Diabetes 2011 and Mellado Gil et al., Cell Death Dis 2016). Did the authors contemplate this venue in BLC that may rationalize reduced beta cell function and insulin content in carriers of PAX4 mutations?

Pax4 is known to be expressed during pancreas development in the endocrine progenitors. In this study, we focused on the major effect of human PAX4 variants during pancreatic differentiation, instead of overexpression of mutant Pax4 in adult beta cells and the effects on apoptosis.

Please remove references from abstract

We have now removed all references from the abstract. We thank Reviewer #2 for this point.

Figure 1e: Please label in figures proband III-1 and 2.

We have addressed this. We thank Reviewer #2 for this point.

Figure 2b to h: Please provide the actual relative values rather than a fold change.

We will address this in the revised manuscript.

Figure 3g: Please add the reference to this panel in line 281. Furthermore, please indicate that the bold GO terms are the ones that include ARX (I guess).

We will address this in the revised manuscript.

Figure 5b and c: The authors claim that CHX stabilizes the p.X186 allele while having no effect on p.His192 (line 338-341). To this reviewer's interpretation, CHX has no effect on either (red and white dots in b and green and white dots in c). Please clarify as this may have a major impact on the conclusion that NMD of p.X186 is the mechanism of haploinsufficiency.

We have included a schematic to explain how we conclude that p.X186 undergoes NMD.

The addition of CHX resulted in an accumulation of the p.X186 allele (based on the signal detected, FAM 400 to 800). The increase in p.X186 allele represents the potentially degraded mRNA if the NMD is not inhibited by CHX.

On the other hand for the p.His192 allele, the treatment with CHX on homozygotes p.His192His (white dots) and non-treated (green dots) did not show any changes. The observation suggests that the p.His192 allele is stable, and did not undergo NMD degradation, so there is no accumulation of the p.His192 allele (FAM 1400) with or without treatment with the NMD inhibitor CHX.

The Taqman allele-specific PCR is not quantitative. The FAM and HEX signal intensities are relative and hence we are not able to report the copy number of each allele. We hope this clarifies.

Extended Data Fig 4: It is unclear whether the immunofluorescence staining in (d) and the Western blot in (f) were performed with the PAX4 or V5 antibody. Please provide this information as the PAX4 antibody may not recognize the truncated p.186X variant.

The PAX4 antibody (RnD, AF2614) is polyclonal. Unfortunately, we are unable to identify the epitope that the antibody binds to (proprietary information not revealed by RnD).

However, we have tested this antibody internally on cells overexpressing PAX4 variants, and were able to detect both the p.192H and p.186X variants. As V5 is located at the 3' end, p.186X introduced a premature stop codon that prevented the expression of V5 (Refer to the figure below). Extended Data Fig. 4F was blotted with PAX4 antibody.

We thank the author for pointing this out. We will clearly indicate the PAX4 antibody used in Extended Data Fig. 4F.

Image of western blot of WT PAX4 (WT), p.His192 and p.X186 proteins overexpressed in AD293 cells. PAX4 and V5 antibodies were used to assess protein expression 24 h post transfection.

Figure 7d: Please clarify the origin of each lane. For example, uncorrected refers to a line derived from the CRISPR correction that failed while p.His192His is the initial line used for CRISPR correction? The same goes for lanes 3 and 4. If so it may be worth emphasizing that CRISPR globally alters the transcriptome landscape. It also may be worth highlighting in bold genes which are mentioned in the text. Interestingly PDX1 was increased in the corrected version.

Uncorrected and corrected lines are donor-derived, each column represents one cell line. Fig. 7d (Column from 1 to 4): Parental, uncorrected (a line that went through CRISPR correction but remains unedited), corrected line, corrected line.

Isogenic lines (heading) are StemBanc hPSC origin, with the *PAX4* variant knock-in.

We have highlighted the genes that are mentioned in the text.

Figure 7e: Columns 1, 3, 5 and 8 lack labeling. The conclusion drawn by the authors is truly a leap of faith for the donor-derived BLC. In fact, the corrected version appears to have fewer beta cell-enriched factors as compared to the uncorrected version. Nonetheless, the isogenic data is clear-cut and conclusive.

Fig. 7e (Column from 1 to 8), all are donor-derived: Parental line 1, Uncorrected line 1, Uncorrected line 2, Parental line 2, Corrected line 1, corrected line 2, corrected line 3, corrected line 4.

We will address some of these details in the revised manuscript.

Reviewer #3 (Remarks to the Author):

The authors report interesting data with attempts to illustrate functionality of *PAX4* identified from GWAS. Cutting-edge technologies were applied including application of beta-like cells (BCLs) derived from patient iPSCs and precise genome-editing by CRISPR/Cas9. Such attempts are much desired for GWAS follow-up studies; however, the present study draws conclusions without sound causal evidence, and suffers from major methodological concerns.

We thank Reviewer #3 for noting that cutting-edge technologies have been applied and that such attempts are much desired. We have now addressed the concerns of Reviewer #3 and seek his/her support for publication. Thank you.

1. Lack of GSIS study in BCLs

This is a fundamental aspect when using BCLs derived from patient iPSCs. If the BCLs do not respond to glucose on insulin secretion, the authors cannot draw firm conclusion on the effect of any genetic manipulation (KD or genome editing) to have any causal effect on beta cell function. This aspect was completely ignored throughout the study, and there is no illustration on how BCLs were characterized as BCLs.

We thank Reviewer #3 for pointing this out. In this study, we focused on the role of human PAX4 and its variants on human pancreatic differentiation, demonstrating a developmental defect that resulted in increased polyhormonal cells with decreased total insulin content.

The functionality of hiPSC-derived BLCs is known to be variable, depending on the hiPSC line used. Hence, given this significant challenge imposed upon the many hiPSC models that we generated, we used an alternative human beta cell line derived from human fetal pancreas – EndoC-bH1, to demonstrate effects on functionality.

Unfortunately, we have indeed stimulated the hiPSC-derived BLCs with high glucose but they were non-functional. To evaluate the importance of PAX4 in beta cell function, we therefore made use of a well-reported human beta cell line (EndoC- β H1) to supplement our findings in hiPSC-derived BLCs.

We understand the request for functional data from hiPSC-derived BLCs. We propose to differentiate a select number of hiPSCs (PAX4-KO versus control; p.His192His donor-derived hiPSCs versus corrected lines; and p.Try186X donor-derived versus corrected lines) into BLCs using Protocol B and assess their response to glucose and KCl.

While we will attempt additional experiments with selected hiPSC lines (PAX4 knockout; key patient lines that were genome corrected), getting fully functional beta cells is known to be difficult, variable and possibly cell line-dependent. We propose to include conditions such as stimulated with KCl in case these beta-like cells do not prove to be glucose-responsive. We seek the agreement of the editor with this proposed approach.

We also note Reviewer #3's request for characterization of BLCs. We have now carefully characterized the endocrine progenitors and beta-like cells differentiated from donor-derived hiPSCs using Protocol B. The efficiencies of generating beta-like cells (30-40% - INS, PDX1), alpha cells (~10%, GCG), and delta cells (15-20%, SST) were highly comparable to Pagliuca et al., 2014, and not significantly different in wildtype versus *PAX4* variants. Due to space constraint, the characterization data was not included in the manuscript. We have now included the FACS characterization data as extended data.

While the FACS assessment has demonstrated no significant difference in the cell populations, our immunostaining data revealed elevated expression of polyhormonal INS+/GCG+ cells (Fig. 4j-m) which led us to uncover that these cells are less mature and expressed lower total insulin content (Fig. 4n).

That said, we will perform additional characterization of the beta-like cells using the current protocols.

FACS assessment on key pancreatic beta cell development markers to characterize endocrine progenitors and beta-like cells differentiated from donor-derived hiPSC lines. hiPSC lines derived from two wildtype, two p.Arg192His, two p.His192His and one p.Tyr186X donors were subjected to differentiation. Percentage of cells stained positive for PAX4, PDX1, INS, GCG and SST on D20 endocrine progenitor (EP) stage and D35 beta-like cell (BLC) stage was consolidated and represented in graphs. At least n=4 experiments were performed.

2. PAX protein expression in all tissues is extremely low or absent (please check in various public databases), including in the pancreas. Have the authors taken this into consideration when designed the study?

Yes, we have taken the expression profile of Pax4 during pancreatic beta cell differentiation into consideration when designing this study. Rodent Pax4 is known to exhibit peak expression in pancreatic endocrine progenitors before decreasing in expression as they develop into insulin-producing beta cells (Sosa-Pineda et al., 1997). However, Pax4 remains to be detectable in adult pancreatic beta cells (Brun et al., 2004).

Using hPSCs as a model for human beta cell development, we have demonstrated that *PAX4* expression peaks on D20 endocrine progenitor stage (Fig. 3c and 3e, EP stage) and decreases towards beta-like cell stage.

Our key aims were to evaluate how *PAX4* LOF can impact human beta cell development, which can further propagate to elevated T2D risk.

3. Page 8 line 172, reduction of HOMA-B was no longer significant after adjusted for age, gender and BMI? What about only BMI, or age, or gender, or any combination? How would the authors interpret the data?

We will address this in the revised manuscript. In brief, we reported the data as it is after appropriate adjustment for the other factors.

4. In Fig 2, gene expression data was presented as “relative to siNT/scramble”. As detailed in the “Methods” session in Page 41, only 1 housekeeping gene ACTIN was used in gene expression experiments. At least 2 housekeeping genes should be used, and what are the Ct values of PAX4? According to public databases, in general *PAX4* gene expression is very low in all tissues.

In EndoC cells, the Ct values of *PAX4* are around 26-28 (*ACTIN* Ct values between 20-22). We will re-run qPCR for another housekeeping gene (such as *GAPDH*) and provide the data accordingly.

5. CRISPR guides: what is the deletion? Between the 2 guides? There is no description of the confirmation of deletion (e.g. by Sanger). The authors did not address off target effects either.

We will address this in the revised manuscript.

6. *PAX4* Arg192His variant (chr7:127253550, rs2233580) is located in Exon 5, the location of Tyr186X is also in Exon 5. CRISPR sgRNA#2 cutting site is around R192, A191 and V190I (theoretically 3-4 bps before or after the PAM seq). What is exactly the deleted region, does it cover the two SNP loci? (Line 259: deletion for AA 64-200?) What is the rationale to design a big deletion fragment in relation to the two GWAS loci?

We will address this in the revised manuscript.

7. It would be better to use *PAX4* wt instead of *PAX4*+/+, which may be confused with insertion mutation.

We thank Reviewer #3 for this suggestion. We have edited the manuscript accordingly.

8. Page 13, why did the authors choose protocol B over A, and applied protocol B for the rest of the study? *PAX4* expression was absent in BLC by protocol B.

This study is a close collaborative effort planned and coordinated between two laboratories. Anna Gloyn's laboratory used protocol A whereas Adrian Teo's laboratory used protocol B. We compared the two differentiation protocols, showing that *PAX4* expression peaks on endocrine progenitor stage (Figs. 3c and 3e). Also, as illustrated in Extended Data Fig. 3 (a bioinformatic comparison of RNA-Seq data from Protocols A and B), the two protocols generated highly similar expression profiles over the course of differentiation.

As Adrian Teo's laboratory had access to the donors, generated the donor-derived hiPSCs, and was using Protocol B, we subsequently decided to combine efforts and perform all experiments with Protocol B after initial comparison experiments. We thank the reviewer for pointing out the lack of clarity, and have now indicated the protocol used for each experiment clearly.

Regarding the **concerns of the chosen protocols**, this mostly requires additional explanations on how our team came together to collaborate, beginning with Protocols A and B, which eventually culminated in most experiments being performed using Protocol B. It does not appear that there is any requirement for additional experiments.

Using hPSCs as a model for beta cell development, we have demonstrated that *PAX4* expression peaks on D20 endocrine progenitor stage (Fig. 3c and 3e, EP stage) and decreases towards beta-like cell stage. These observations are consistent with the studies reported in rodents whereby rodent *Pax4* expression peaks on E9.5 to drive endocrine progenitors towards the insulin-producing beta cell fate before its expression decreases towards maturation (Sosa-Pineda et al., 1997). Therefore, it is an expected observation in which *PAX4* expression is lower but not absent in BLCs.

Volcano plots in panel d and f showed completely different gene profiles generated from protocol A and B, except for *DENND2D*, any follow-up study on *DENND2D*? In panel f, *PAX4* is again absent, not among the top genes that were affected by *PAX4* KD, which makes it further questionable the choice of protocol B.

The top genes are among the most differentially expressed genes (by transcript fold change) and *PAX4* expression is low in mature beta cells (Sosa-Pineda, 1997). Therefore, it is not unexpected that *PAX4* does not appear among one of the most differentially expressed genes.

DENND2D is a Guanine nucleotide exchange factor (GEF) that plays a role in converting the inactive GDP-bound Rab proteins into the active GTP-bound form (Yoshimura et al., 2010, J Cell Biol). While

some of the members of the GEF are involved in membrane trafficking in insulin secretion (Kowluru, 2021, *ACS Pharmacol. Transl. Sci.*), beyond its well-reported role as a tumour suppressor (Marat et al., 2011, *J Biol Chem*; Ling et al., 2013, *Lung Cancer*), there is no literature supporting the role of *DENND2D* in pancreatic beta cell development/function. We thank the reviewer for highlighting this gene, but it is beyond the scope of this study to look into *DENND2D*.

9. Page 14, line 290-292, this is an overstatement: “PAX4 is not required for human beta cell differentiation in vitro. Rather, PAX4 loss-of-function results in derepression of alpha cell genes and a dysregulation of key endocrine maturation genes in hiPSC-derived BLCs.” As stated on page 13 Line 283, nonsignificant.

We thank Reviewer #3 for this point. We have toned down on the language and amended it accordingly in the revised manuscript.

10. What is the ethnic origin of the hiPSC lines used in Fig 3? Furthermore, the PAX4 variants were identified in Asian populations, and EndoC cell line is of French origin. (<https://www.ncbi.nlm.nih.gov/pmc/articles/PMC3163974/>). Indeed, many human derived cell lines do carry ethnic-specific signatures. Any thoughts?

The isogenic SB Ad3.1 hiPSC line used in Fig. 3, derived from human skin fibroblasts from a Caucasian donor with no reported diabetes (Lonza CC-2511, tissue acquisition number 23447), was obtained from the Human Biomaterials Resource Centre, University of Birmingham.

In this study, we are specifically investigating the role of PAX4 in human pancreatic and beta cell lines, irrespective of the ethnicity of the cell lines. Regardless of the Asian *PAX4* donor-specific hiPSCs, Caucasian donor-specific SB hiPSCs or the French EndoC- β H1 lines, we have identified a consistent observation regarding the importance of PAX4 in repressing human alpha cell *GCG* expression and in determining human beta cell *INS* content. The higher prevalence of PAX4 R192H in Asian cohorts can therefore influence the quality of human beta cells in Asians.

11. What is the consequence of increased *GCG* gene expression in translation? In protein expression detection? Fig 4 IF stained for GCG. The mammalian proglucagon gene (*Gcg*) encodes three glucagon like sequences, glucagon, glucagon-like peptide-1 (GLP-1), and glucagon-like peptide-2. Have the authors looked at other potential translational products?

The loss of beta cell identity and the acquisition of polyhormonal cells have been reported in the pancreatic islets of individuals with diabetes (Cinti et al., 2016; Talchai et al., 2012). The transdifferentiation of metabolically-stressed beta cells to express GCG has been proposed by several groups as a mechanism underlying beta cell failure in T2D (Brereton et al., 2014; Talchai et al., 2012; Wang et al., 2014). The GCG antibody was carefully selected for the identification of alpha cells (Santa cruz, sc-7780, used by Kroon et al., 2008, Ackermann et al., 2018; Dooley et al., 2016).

We did not look into the expression of GLP-1, GLP-2 in our hiPSC models. However, we did assess the GLP-1 measurements during OGTT in carriers of p.Arg192His variant (Extended Data Fig. 1F – H, below). There is no significance difference between the wildtype and carriers.

Clinical assessment of GLP-1 in carriers of p.Arg192His PAX4 variant during two-hour OGTT.

(F) Fasting, (G) 20-min and (H) AUC of GLP-1 secreted in subjects.

12. Line 320-322, this is an overstatement. In his192 allele carriers, no change in PAX4 or INS, GCG, SST transcription was detected in this study. Line 322, “negatively affecting endocrine cell differentiation”, this is a bold conclusion based on only GCG staining, without further data to back up endocrine cell differentiation.

We thank the reviewer for highlighting this. With the new FACS characterization data that we will be including as extended data, we will tone down our statement to:

“Heterozygous and homozygous carriers of the PAX4 p.His192 allele had no measurable differences in *INS*, *GCG*, or *SST* gene expression (Fig. 4b-i). There is also no significant difference in the cell populations expressing *INS*, *GCG*, or *SST* at the EP or BLC stages (new extended data on FACS characterization). Together, these data suggest that both *PAX4* alleles result in a loss-of-function due to reduced *PAX4* gene dosage and/or altered *PAX4* transcriptional activity, negatively affecting pancreatic beta cell differentiation.”

13. PAX4 primers: hPax4F5 (AGGACACGGTGAGGGTCTGGT) shows no blot result (please correct me if I was wrong); hPax4R5 (CAGTGGTTCCAGGGCAGGCA) is located in Exon 8. Tyf186X is supposed to introduce premature stop codon at AA186 (line 333-334). How is the amplification region in relation to the variants?

Please kindly see image below for the illustration of the primer binding sites relative to Tyr.186X (GTA duplication) and Arg.192His (G>A) mutations.

Homo sapiens paired box 4 (PAX4), transcript variant 1, mRNA

>NM_001366110.1:407-1462 Homo sapiens paired box 4 (PAX4), transcript variant 1, mRNA

```
ATGCATCAGGACGGGATCAGCAGCATGAACCAGCTTGGGGGGCTCTTTGTGAATGGCCGGCCCTGCCTCTGGATAC
CCGGCAGCAGATTGTGCGGCTAGCAGTCAGTGGAAATGCGGCCCTGTGACATCTCACGGATCCTTAAGGTATCTAATG
GCTGTGTGAGCAAGATCCTAGGGCGTACTACCGCACAGGTGTCTTGGAGCCAAAGGGCATTGGGGGAAGCAAGCC
ACGGCTGGCTACACCCCTGTGGTGGCTCGAATTGCCAGCTGAAGGGTGAGTGTCCAGCCCTCTTGCCTGGGAAA
TCCAACGCCAGCTTTGTGCTGAAGGGCTTTGCACCCAGGACAAGACTCCAGTGTCTCCTCCATCAACCGAGTCTCTGC
GGGCATTACAGGAGGACCAGGACTACCGTGCACACGGCTCAGGTCACCAGCTGTTTTGGCTCCAGCTGTCTCACT
CCCCATAGTGGCTCTGAGACTCCCCGGGGTACCCACCCAGGGACCGGCCACCGGAATCGGACTATCTTCTCCCAAG
CCAAGCAGAGGCACTGGAGAAAGAGTTCAGCGTGGGCAGTAT(GTA)CCTGATTCAAGTGGCCCGTGGAAAGCTGGC
TACTGCCACCTCTCTGCCTGAGGACACGGTGAGGGTCTGGTTTTCCAACAGAAGAGCCAAATGGCGTCGGCAAGAGA
AGCTCAAGTGGGAAATGCAGCTGCCAGGTGCTTCCAGGGGCTGACTGTACCAAGGGTTGCCCCAGGAATCATCTCT
GCACAGCAGTCCCTGGCAGTGTGCCACAGCAGCCC TGCCTGCCCTGGAACTCTGGTCCCTCTGTATCAGCTG
TGCTGGGCAACAGCACCAGAAAGGTGTCTGAGTGACACCCACCTAAAGCCTGTCTCAAGCCCTGCTGGGGCCACTT
GCCCCACAGCCGAATCCCTGGACTCAGGACTGCTTGGCTTCCCTTGCCTTCCCTCCACTGTCACTGGCCAGTCTT
AGTGGCTCTCAGGCCCTGCTCTGGCTGGCTGCCACTACTGTATGGCTTGGAAATGA
```

hPax4F5 primer

hPax4R5 primer

14. What is the epitope of PAX4 antibody?

The PAX4 antibody (RnD, AF2614) is polyclonal. Unfortunately, we are unable to identify the epitope that the antibody binds to (proprietary information not revealed by RnD).

15. CHIP should be performed to demonstrate interaction between PAX4 and gene promoter regions.

With the currently available PAX4 antibody, we are not confident that it is ChIP-grade. Hence, we respectfully respond that this is currently out of scope of this manuscript. We also seek the reviewer's understanding that luciferase assays (which is more sensitive in the detection of transcriptional regulatory activity) is a more suitable technique that we applied to understand gene promoter transcriptional regulation.

16. Fig 5h shows that PAX4 kd had no effect on INS promoter activity, how to explain Fig 2g where it shows increased INS gene expression by PAX kd?

In Fig. 5d, WT PAX4 repressed *INS* promoter in EndoC-bH1 cells. When *PAX4* was knocked down, the repression activity was partly lost (albeit not statistically significant) (Fig. 5g). In Fig. 2g, there was

variability in *INS* gene expression and hence there was no difference between shScrambled and sh*PAX4* EndoC-bH1 cells.

As *PAX4* is not the only transcription factor that regulates expression of *INS* gene, we speculate that the higher expression of *INS* could be a compensatory effect by other beta cell determinant transcription factors. We did not explore further as it is beyond the scope of this study.

17. Fig 6e and g, there is no significant difference between the variants. Should be carefully interpreted in data presentation.

We thank the reviewer for highlighting this point. Fig. 6e and 6g demonstrate the overall glycolytic function and mitochondrial respiration capacity, respectively. Fig. 6F and 6H further dissected the metabolic activities within the two main pathways for energy generation in the cells. While there was no significant difference between wildtype and *PAX4* variant lines in the overall glycolytic function (Fig. 6e) and mitochondrial respiration (Fig. 6g) at the stipulated time-points, the metabolic activities within the cell lines were perturbed (such as glycolytic capacity, glycolytic reserve, basal respiration, ATP production, etc). These changes can only be uncovered when we look deeper into the metabolic profiles (ECAR or OCR). Please kindly refer to figure below for details on how each of the parameter of measurement is derived.

Calculations for the various parameters in Glycolysis stress test

Glycolysis	(Maximum rate measurement before Oligomycin injection) – (Last rate measurement before Glucose injection)
Glycolytic Capacity	(Maximum rate measurement after Oligomycin injection) – (Last rate measurement before Glucose injection)
Glycolytic Reserve	(Glycolytic Capacity) – (Glycolysis)
Glycolytic Reserve as a %	(Glycolytic Capacity Rate) / (Glycolysis) × 100
Non-Glycolytic Acidification	Last rate measurement prior to glucose injection
Acute Response	(Last measurement rate before glucose injection – Last rate measurement before acute injection)

Calculations for the various parameters in Mitochondrial stress test

Parameter Value	Equation
Non-mitochondrial Oxygen Consumption	Minimum rate measurement after Rotenone/antimycin A injection
Basal Respiration	(Last rate measurement before first injection) – (Non-Mitochondrial Respiration Rate)
Maximal Respiration	(Maximum rate measurement after FCCP injection) – (Non-Mitochondrial Respiration)
H+ (Proton) Leak	(Minimum rate measurement after Oligomycin injection) – (Non-Mitochondrial Respiration)
ATP Production	(Last rate measurement before Oligomycin injection) – (Minimum rate measurement after Oligomycin injection)
Spare Respiratory Capacity	(Maximal Respiration) – (Basal Respiration)
Spare Respiratory Capacity as a %	(Maximal Respiration) / (Basal Respiration) × 100
Acute Response	(Last rate measurement before oligomycin Injection) – (Last rate measurement before acute injection)
Coupling Efficiency	ATP Production Rate) / (Basal Respiration Rate) × 100

18. Line 411, these changes are not significant in Extended data Fig6

In Line 411, we demonstrated that endocrine progenitors carrying *PAX4* variants had altered metabolic signatures, and we interrogated if the elevated oxidative stress could be one causal factor

accounting for reduced total insulin content (Fig. 4n). However, when we treated the cells with the antioxidant NAC, it did not result in restoration of total insulin content as highlighted by Reviewer #3 (Extended data Fig. 6).

19. Fig 7f and g, Extended Data Fig 7, wt is missing in all experiments for control

In Fig. 7f, g, and Extended Data Fig. 7., we did not include parental or WT in our experiments. Rather, we based our experimental design on Fig. 7c, where we had the ideal corrected control vs uncorrected isogenic pairings for data comparisons. We seek Reviewer #3's understanding on this experimental comparison which is sufficient to lead to conclusions.

Regarding **rescue experiments**, the reviewers mentioned wildtype cells. However, in each instance, we always paired mutant with gene-corrected hiPSCs (in essence wildtype), having an ideal isogenic pair of lines for direct comparisons. We are of the opinion that additional text clarification is needed for Fig. 7. However, additional wildtype or parental lines that were not subjected to the CRISPR genome editing are not as applicable as the isogenic pairs that we have already used. We seek the understanding and agreement of the editor on this point.

Point-by-point response (2nd response letter)

Reviewer #1 (Remarks to the Author):

This study by Lau&Krentz et al. contains comprehensive clinical and experimental studies that overall support the title of the paper. The results are mostly clear and well described. However, there are several problems related with Fig.7 (the rescue experiments by mutation correction).

We thank Reviewer #1 for his/her support of our manuscript. We have addressed the concerns relating to Fig. 7 below. Thank you.

Major comments:

1. Differentiation of iPSC into islet-like cells: Two well-known but related protocols were used (refs 22 and 25). The study comes from two collaborating laboratories. Does this mean that one lab used protocol A and the other one B, or were both protocols established at both sites, or only one site? Was this planned and coordinated, or were the results of two labs combined retrospectively? It would be fair to make this clear.

Thank you for providing us the opportunity to clarify how this collaborative study was performed, as we think it is a major strength of our study. Our laboratories (Teo and Gloyn) and collaborators embarked on this study together and coordinated our efforts so that each team led on parts of the study where they had the most expertise. The Teo group led on the generation of hiPSC-derived donor cells and differentiation of all cell lines using Protocol B. The Gloyn group focused on CRISPR-Cas9 genome correction of donor-derived cells and generation of isogenic PAX4 KO and variant hiPSC lines. The bulk of the differentiations were performed by Adrian Teo's group using Protocol B, whilst Anna Gloyn's group used Protocol A for differentiation of the PAX4 KO cells. Both labs performed experiments on EndoC-βH1 cells.

The decision to predominantly use Protocol B was made after we did the head-to-head comparison of the two protocols (Gloyn Protocol A and Teo Protocol B) on the PAX4 KO cells. We did not observe significant differences between the two protocols transcriptionally (**Supplementary Fig. 3**), with both showing peak *PAX4* expression in the endocrine progenitor stage (**Figs. 3c and 3e**). As the Teo lab was tasked with performing the bulk of the *in vitro* stem cell differentiations and had extensive experience using Protocol B (Low *et al.*, 2021; Nguyen *et al.*, 2021), the remaining differentiations were performed using Protocol B. We hope the reviewers agree that one of the strengths of our study is in the combined expertise of our two groups, along with the clinical data from our collaborators, which allowed us to more comprehensively address how genetic variation at the *PAX4* locus contributes to T2D risk.

2. It would also be important to present at least the elementary general characterization of the differentiation efficiencies using the two protocols in the Supplementary materials. How do the yields of Pdx1/Nkx6.1 endocrine progenitors at stage 4 compare? What about the yield of beta and alpha cells?

The focus of our collaborative study was to elucidate the biological mechanism of how coding variants in *PAX4* impact T2D risk by altering pancreatic beta cell development and/or function. As such, we did not intend to compare the differentiation efficiencies of Protocols A and B in deriving pancreatic beta-like cells. Because the differentiations were performed by two different labs and not done in parallel, we don't think our experimental setup is the best way to perform such protocol comparison studies.

We agree with the reviewer that the general characterization of the differentiation efficiencies is important. For Protocol A, we had previously characterized the differentiation towards the definitive endoderm stage (**Supplementary Fig. 2b-c**). For Protocol B, we had previously characterized the differentiation efficiency of BLCs on the donor-derived hiPSC lines (**Fig. 4j-m**) by immunofluorescence and cell quantification. Using this quantitative technique, we identified significant differences in the proportion of total nuclei that co-expressed C-PEP and GCG (**Fig. 4m**), with homozygous carriers of the p.His192 risk allele having significantly increased co-expression. To address the reviewer's concerns on the yield of beta and alpha cells in BLCs generated from donor-

derived hiPSCs carrying *PAX4* variants, we performed new flow cytometry experiments using Protocol B and evaluated the percentage of beta (INS+, PDX1+), alpha (GCG+) and delta (SST+) cells using flow cytometry (Supplementary Fig. 5). The efficiencies of generating beta (30-40%), alpha (~10%), and delta cells (15-20%) were comparable to the adapted protocol (Pagliuca *et al.*, 2014). Again, we did not see any distinct differences in the percentage of cell types when comparing the *PAX4* variant-carrying cells to wild-type donors.

A caveat of this data is that the wild-type cell lines are hiPSCs derived from independent donors. As we know that genetic background of individual donors can contribute to transcriptional differences (Rouhani *et al.*, 2014), the definitive experiment is to compare the effect of the genotype at the *PAX4* locus using CRISPR-corrected control cells. Therefore, we have now performed additional flow cytometry analyses on uncorrected p.Tyr186X vs corrected p.Tyr186Tyr hiPSC lines derived from the same donor, as well as uncorrected p.His192His vs corrected p.Arg192Arg hiPSC lines derived from the same donor (Supplementary Fig. 4). The new data lend support to the main message of our study, that *PAX4* variants/*PAX4* KO did not detrimentally impact the differentiation trajectory of hiPSCs towards pancreatic endocrine cells but impacted the maturation of BLCs (increased polyhormonal expression and reduced total insulin content; Fig. 4).

Supplementary Fig. 4

Supplementary Fig. 5

And insulin secretion in response to glucose plus other secretagogues (like GLP1, KCI)?

The functionality of hiPSC-derived BLCs is known to be variable and is highly dependent on the hiPSC line used. Previously, we stimulated the hiPSC-derived BLCs from Protocol B with high glucose and determined they were not functionally mature (static GSIS, data not shown). **We performed new experiments to evaluate dynamic GSIS using a perfusion system** from a selected set of cell lines (*PAX4*^{WT/WT} vs *PAX4*^{KO/KO}; uncorrected p.Tyr186X vs corrected p.Tyr186Tyr; and uncorrected p.His192His vs corrected p.Arg192Arg). Unfortunately, our hiPSC-derived BLCs using Protocol B were still not robustly functional under high glucose and secreted insulin only when treated with 30 mM potassium chloride (**Now Supplementary Fig. 11**). As expected, we also observed heterogeneity across cell lines carrying the same genotype, making it challenging to draw any conclusions from the data.

Supplementary Fig. 11 | Dynamic glucose-stimulated insulin secretion performed on beta-like cells. Donor hiPSC-derived beta-like cells carrying (a) uncorrected p.His192His (two lines); (b) corrected p.Arg192Arg (two lines); (c) uncorrected p.Tyr186X (one line); and (d) corrected p.Tyr186Tyr (one line) were stimulated at 2.8 mM (6 min), 16.7 mM (40 min), 2.8 mM (16 min) and 30 mM KCL (6 min) sequentially. Each graph represents data obtained from one hiPSC line.

To overcome the immature state of our hiPSC-derived BLCs, we included an alternative human beta cell line derived from human fetal pancreas – EndoC-βH1 – to demonstrate the consequence of *PAX4*

perturbation on beta cell function (Fig. 2). Importantly, in line with our observations in BLCs derived from hiPSCs, EndoC-βH1 cells knocked down for PAX4 also exhibited reduced maturity as witnessed by reduced total insulin content, which likely contributed to the impaired GSIS (Fig. 2c,e-f), and replicated the clinical phenotype of the recruited subjects (Fig. 1a,d, lower insulin secreted in response to glucose). We have added the following sentences to the discussion section (lines 523-526):

“Our hiPSC-derived beta cells were not robustly functional under high glucose and were demonstrated to secrete insulin only when treated with 30 mM potassium chloride (Supplementary Fig. 11). To circumvent the limitations of our study in deriving fully functional beta cells in vitro, we included the study of PAX4 in the human beta cell line EndoC-βH1.”

It is also very important to clearly indicate which protocol was used for each result presented (missing at least for Fig. 7).

We have now clearly indicated the protocol used in each experiment in the revised figure legends.

3. Fig. 4 k-n shows clearly the important effects of the LOF mutations on the insulin content and presence of polyhormonal cells. Why is this information lacking for the Pax4 KO lines? This is an important part of the basic characterization that is missing (see point 3), and it would be particularly important to include this characterization for the Pax4 KO lines in comparison with the isogenic WT controls.

We thank the reviewer for this comment and agree that it is important to characterize the effect(s) of PAX4 LOF in the KO lines. We have now performed **additional flow cytometry characterization for the PAX4 KO lines** using Protocol B (three control and three PAX4-KO lines; isogenic SB origin) and have inserted the data under **Supplementary Fig. 4**). The additional characterization revealed no distinct differences between control and PAX4 KO lines in their efficiencies in differentiating towards definitive endoderm (SOX17⁺/CXCR4⁺), endocrine progenitors (INS⁺, NKX6.1⁺ and PDX1⁺), as well as beta-like (INS⁺ and NKX6.1⁺) cells. However, due to the heterogeneous nature of hiPSC models, we were not able to detect any significant difference when comparing the total insulin content between the control and PAX4 KO lines (Figure below, data not included in the manuscript). The observations stay true with the key message of our study. Unlike in rodents, PAX4 KO did not detrimentally impact the differentiation trajectory of hiPSCs towards pancreatic endocrine cells as supported by **Supplementary Fig. 4a-c**. This has now been acknowledged in the results section (lines 293-300):

“Importantly, PAX4^{WT/WT} and PAX4^{KO/KO} lines differentiating into BLCs repress pluripotency genes and activate genes involved in endocrine cell fate in a similar manner (Supplementary Fig. 3), giving rise to similar proportions of DE, EP, and BLC (Supplementary Fig. 4a-c). These observations suggest that, unlike in mouse, PAX4 is not essential for human beta cell differentiation in vitro and its loss did not detrimentally impact the differentiation trajectory of hiPSCs towards pancreatic endocrine cells. Rather, PAX4 loss-of-function results in a dysregulation of key endocrine maturation genes in hiPSC-derived BLCs.”

Total insulin content in PAX4 wild-type and knockout hiPSC-derived beta-like cells.

4. An improved protocol has recently been published by Balboa et al (PMID 35241836), showing further maturation and the loss of polyhormonality during extended final stage. It would be interesting to see the impact of Pax4 LOF (both the KO and the mutations studied) on the C-Pep/GCG double positive cells after extended maturation.

We thank the reviewer for this suggestion. The Balboa *et al.* protocol involves initial differentiation as monolayer (definitive endoderm to posterior gut tube), transiting to microwell culture (from pancreatic to endocrine progenitor) and further maturation as suspension culture over a duration of up to six weeks (total differentiation duration adding up to over 60 days). In Balboa *et al.*, hESC (human embryonic stem cell) H1 line was predominantly used to optimize the differentiation protocol to achieve beta-like cells with remarkable insulin secretion profile. While robust insulin secretion was still observed, differentiation of two other hiPSC lines with the same protocol resulted in variable amounts of insulin being secreted (Balboa *et al.*, 2022; Supplementary Figure 1h). The difficulty of adapting differentiation protocols to new hiPSC lines is consistent with our labs' experience in using hiPSCs as a platform for disease modeling (Chan and Teo, 2020; Perez-Alcantara *et al.*, 2018). To overcome the variability in hiPSC differentiation *in vitro*, we used a combination of donor-derived hiPSCs (16 lines), CRISPR-corrected donor-derived isogenic hiPSCs (10+ lines), SB Ad3.1 (commercially available and then genome edited at the *PAX4* locus, 10+ lines), and subjected them to differentiation using two established protocols (Protocols A and B). As can be appreciated, the scale of this approach to model disease using *in vitro* differentiation is time consuming and expensive, but is necessary to overcome variability and to include proper controls. Importantly, our donor hiPSCs were generated from individuals without diabetes and we have begun to make several of these lines available to the wider research community (<https://skip.stemcellinformatics.org/en/>).

We agree that the Balboa *et al.* protocol represents a substantial update in our ability to generate functional beta-like cells *in vitro*. However, we hope that the reviewer will agree that it is out of scope of our current manuscript for the following reasons:

- 1) Our current study already used two different differentiation protocols. To adopt another differentiation protocol for the number of hiPSC lines we have would be a tremendous effort.
- 2) While Protocols A and B do not generate functional beta-like cells, we have complemented these studies with clinical data from human carriers of *PAX4* alleles (**Fig. 1**), which show lower insulin secreted in response to glucose.
- 3) Consistent with the clinical data, *PAX4* perturbation in the human beta cell line EndoC- β H1 (**Fig. 2**) reduced total insulin content, which likely contributed to the impaired GSIS (**Fig. 2e-f**) and replicated the clinical phenotype of the recruited subjects (**Fig. 1**).

5. The mutation correction results presented in Fig. 7 are very difficult to understand:

- The genotypes of the lines are expressed confusingly. Clearly, in panel a, it is the p.Arg192His mutation that is corrected (and not His192His)? Similar confusing labels can be found also in other panels.

We thank the reviewer for highlighting this point and we apologize for the lack of clarity. In panel a, it is indeed p.His192His (homozygote for His allele encoded by A nucleotide) that is being corrected into p.Arg192Arg (wildtype: **CGT**). For simplicity, we have only shown the one allele/double stranded DNA sequence, but the genotype at both alleles is being corrected from A -> G. We have now added an additional schematic indicating what the corrected genotype is. We hope this improves the clarity of the CRISPR-correction strategy.

- Panels d and e: Labelling of the lanes is very confusing. I believe that in (d) the same labelling applies for both two lanes on the left and right (uncorrected vs corrected), and in (e) the same label is supposed to cover 4 lanes (?). This is really difficult to understand, so please clarify.

We apologize for the confusion. In panels d and e, each column represents one cell line. Yes, the same labeling applies for both lanes on the left (meaning that two lines carrying the uncorrected genotype, p.His192His, were displayed) and the "corrected" label applies to the two lanes on the right (two lines with corrected genotype, p.Arg192Arg, wildtype). Similarly, there are four lines with uncorrected genotype (p.Tyr186X, mutant) and four lines with corrected genotype (p.Tyr186Tyr,

wildtype) displayed in panel e. We have added additional lines to demarcate which columns correspond to which genotype.

- Furthermore, I simply do not understand what the “isogenic BLC” stand for? The term “isogenic” in my understanding should mean that the cells have the same background genotype, i.e- are derived from the same donor. So, what are these? And how should one interpret the result when the color scale is completely different from the “donor-derived BLC”. I find it hard to identify obvious patterns in the expression that would make sense (this applies also the replicate clones particularly in panel d).

In panels d and e, each column represents one cell line. For panel d, the first lane (on the left) refers to the parental line carrying the p.His192His variant (donor-derived hiPSC), second lane is parental lane that underwent CRISPR correction but retained the p.His192His mutation (CRISPR control line), while third and fourth lanes represent successfully corrected lines that are now of the wildtype genotype p.Arg192Arg. As there is only one line each for the parental and control line, we placed them together under the same label as the two lines are carrying “Uncorrected p.His192His” variant.

In panel e, there are two donor-derived hiPSC lines naturally carrying the p.Tyr186X variant (lanes 1 and 4) and two lines that underwent CRISPR correction but retained the p.Tyr186X genotype (lanes 2 and 3, CRISPR control lines). Lanes 5 to 8 refer to CRISPR-corrected donor-derived hiPSCs. We noticed higher heterogeneity in hiPSCs derived from donor II-7 (as in panel e), making it harder to draw a conclusion. Again, we placed the parental and CRISPR control lines under the same label as they are carrying the same “Uncorrected p.Tyr186X” variant. Due to the high heterogeneity, “isogenic BLCs” were introduced to provide better clarity.

In panels d and e, the lanes under “isogenic BLC” originate from SB Ad3.1 cell line, with p.His192His/p.Tyr186X mutation knocked-in (on the left) and wildtype control (on the right). We termed them “isogenic” because the two lines (mutation knocked-in and control) were derived from the same source SB Ad3.1 hence sharing isogenic background. As we only have one line each for the *PAX4* variant knocked-in and control line, even though **the same color scale had been applied**, there will only be two colors to represent the directional expression of the genes (red and blue, in this case).

Comparing the “Uncorrected” columns composed of parental and CRISPR controls, as can be appreciated, panels d and e highlighted the heterogeneous nature of hiPSCs even in lines that were derived from the **same donors**. We appreciate the reviewer for recognizing the heterogeneous nature of hiPSCs even though the cells share an isogenic background.

To address the reviewer’s concern in missing obvious patterns in the transcript expression, **we performed additional experiments** to support the hypothesis that the dysregulated gene expression profiles in *PAX4* variant carrying BLCs contributed to impaired beta cell maturation. We have incorporated the new data under Supplementary Fig. 10 to demonstrate that BLCs with corrected genotypes had reduced number of polyhormonal cells (C-PEP⁺/GCG⁺) and hence had higher total insulin content (Fig. 7f-g).

“Importantly, correcting the PAX4 p.His192His and p.Tyr186X mutations decreased the formation of polyhormonal (C-PEP⁺/GCG⁺) cells (Supplementary Fig. 10) and significantly increased the total insulin content of the BLCs (Fig. 7f-g), indicating that the PAX4 variants were a direct cause of reduced insulin content in the donor-derived BLCs.”

- Based on a lot of evidence provided in the manuscript, a hallmark of Pax4 LOF should be an increased alpha-cell expression pattern, represented by higher ARX expression. However, in panels d-e it appears that ARX is consistently higher in the corrected than in uncorrected cells?

It is an interesting observation that *ARX* is consistently upregulated in the BLCs differentiated from donor-derived CRISPR-corrected lines. We took reference from a study by Prof. Timothy Kieffer’s team (Gage *et al.*, 2015), which reported an unexpected reduction of insulin-positive cells despite a significantly higher expression of *PAX4* seen in *ARX*-knockout pancreatic progenitors. Upon re-introduction of *ARX* into the *ARX*-knockout endocrine cells, Kieffer’s team observed a restoration of insulin-positive cells and hence confirmed the importance of *ARX* in beta cell development on top of its well-reported role in the regulation of alpha cell development.

We thank the reviewer for identifying the expression of *ARX* to be consistently higher in the gene corrected BLCs and we are equally curious to find out the effect of *PAX4* LOF in human endocrine cells. **We have now performed additional experiments to assess the expression of *ARX* in our sh*PAX4* EndoC- β H1 human beta cell line.** We observed significantly higher expression of *GCG* but lower *ARX* expression when *PAX4* was knocked down (Fig. 2i). The new data has been incorporated/replaced Fig. 2d, 2g, 2h and 2i. Together with the findings by Kieffer's team that emphasized a role for *ARX* in both human beta and alpha cell lineage, our results are consistent with the expression of *ARX* not being a direct representative of alpha cell fate, at least in these human cell models.

Fig. 2

Minor:

1. Title: The title could be a bit more specific; instead of “endocrine cell”, rather “islet cell” or “pancreatic endocrine cell”; and instead of “influences”, rather “increases”). As a result, perhaps an appropriate title could be: “Pax4 loss of function increases diabetes risk by altering human pancreatic endocrine cell development”.

We thank the reviewer for the suggestion, we have now amended the title as suggested.

“PAX4 loss of function increases diabetes risk by altering human pancreatic endocrine cell development.”

2. Line 195: HbA1c was 7.1%/8.7 mmol/L. If the intention is to express the HbA1c in molar units, I suppose this should be around 54 mmol/mol. But perhaps the authors mean that the HbA1c of 7.1% corresponded to a mean b-glucose level of 8.7 mmol/L. Please clarify.

We have now clarified this in the revised manuscript (line 196). We have updated the unit for all blood glucose measurements to ‘mg/dL’, following American standard.

“Following lifestyle modifications, she lost weight (from 53.1 kg, BMI 25.3 kg/m² to 49.5 kg, BMI 23.6 kg/m²) and nine months post-diagnosis her HbA1c was 7.1% (mean basal glucose level of 156.6 mg/dL).”

3. Fig. 1f: In the table the glucose values are obviously in mmol/L, but the column title has mg/dL.

We thank the reviewer for pointing this out. We have now corrected the table title accordingly.

9	Subject	Age	BMI	Fasting glucose (mg/dL)	1-hour glucose (mg/dL)	2-hour glucose (mg/dL)	HbA1c (%)	HOMA-IR	DI
	III-1	12	21.2	86.4	ND	ND	5.6	3.31	ND
	II-11	41	24.1	95.4	174.6	118.8	5.5	1.98	4.34
	II-7	51	23.8	109.8	286.2	144.0	6.6	1.41	1.53
	II-3	58	27.0	81.0	147.6	109.8	5.6	1.75	10.76
	II-4	57	24.9	97.2	289.8	372.6	8.1	5.33	0.02
	II-5	55	24.3	190.8	349.2	396.0	8.8	3.77	0.32
	II-8	50	33.6	122.4	311.4	221.4	7.1	9.14	0.47
	II-9	55	25.4	154.8	309.6	331.2	7.0	5.17	0.4

Reviewer #2 (Remarks to the Author):

The study by Lau/Krentz and colleagues expand our current knowledge on the functional role of the transcription factor PAX4 in human islet development, and function, and how mutations, including a new one identified herein in a Singapore family (p.X186), may contribute to type 2 diabetes risk. Using both clinical and experimental approaches that include the generation of iPSCs-derived beta-like cells obtained from individuals with this new mutation combined with CRISP and RNAseq technologies, the authors demonstrate that PAX4 is essential to repress the alpha genetic program which favors beta cell identity hallmarked by insulin content and functional insulin secretion. The study is of interest with a massive amount of data that will be resourceful for the scientific community.

We thank Reviewer #2 for the positive comments, interest and support for our study.

General comments:

Mouse studies have shown that overexpression of the Pax4 diabetes-linked mutant variant R129W in adult mice beta cells sensitize to stress and apoptosis (Hu He et al., Diabetes 2011 and Mellado Gil et al., Cell Death Dis 2016). Did the authors contemplate this venue in BLC that may rationalize reduced beta cell function and insulin content in carriers of PAX4 mutations?

Pax4 is known to be expressed during pancreas development in the endocrine progenitors. In this study, we focused on the major effect of human PAX4 variants during pancreatic differentiation, rather than its role in adult beta cells and apoptosis. It is possible that the two PAX4 variants we studied could also play a role in stress and apoptosis. However, as coding variants will impact the protein from the point at which it begins to be expressed (pancreas development), any impact in mature beta cells would be in addition to, and perhaps caused by, PAX4's important role during development.

Please remove references from abstract

We have now removed all references from the abstract.

Figure 1e: Please label in figures proband III-1 and 2.

We thank the reviewer for highlighting this point. We have added label in Fig. 1e to identify the proband.

Figure 2b to h: Please provide the actual relative values rather than a fold change.

We thank the reviewer for this comment. We acknowledge the importance of Ct values particularly in knockdown studies. We have addressed this in the revised manuscript.

The siRNA studies were performed by the Gloyn team. In their hands, the Ct value of PAX4 ranged between ~28-32. As a comparison, the expression of INS in these cells amplified at an average CT of

~13, and the housekeeping gene *TBP* amplified at ~25. Unfortunately, the range in *PAX4* expression across passages and knockdown experiments makes it difficult to plot as relative values. As you can see from the graph below, there was a 50% reduction in *PAX4* expression following knockdown across all four experimental replicates. However, there is a greater than five-fold difference in the expression of *PAX4* in the non-targeted siRNA control (siNT). As such, we have opted to present fold knockdown compared to siNT (effectively normalizing for the variability in *PAX4* expression across experiments).

PAX4 expression relative to *TBP* in siRNA treated EndoC-βH1 cells.

The shRNA studies were performed by the Teo team. In their hands, the average Ct value for *PAX4* gene was ~26.7 in shSCR; ~27.2 in shPAX4 cells, with an average *ACTIN* Ct value of 20.4. We have consolidated and included additional information on the Ct values of *PAX4* and additional housekeeping genes (*ACTIN*, *GAPDH*, *TUBB3*, *PPIA*) in the figure below for better clarity for the reviewer (figures will not be included in the manuscript).

Figure 3g: Please add the reference to this panel in line 281. Furthermore, please indicate that the bold GO terms are the ones that include ARX (I guess).

Thank you for pointing out our oversight. We have cited this in the revised manuscript and updated the figure legend to explain that bold GO terms include *ARX*.

Figure 5b and c: The authors claim that CHX stabilizes the p.X186 allele while having no effect on p.His192 (line 338-341). To this reviewer's interpretation, CHX has no effect on either (red and white dots in b and green and white dots in c). Please clarify as this may have a major impact on the conclusion that NMD of p.X186 is the mechanism of haploinsufficiency.

We thank the reviewer for pointing this out. We have included a schematic here to explain how we concluded that p.X186 undergoes NMD. With treatment of CHX (which inhibits NMD) you see a rightward shift and stabilization of the p.X186 allele (increase from ~400 to ~800) but there is no stabilization of the p.His192 allele (as indicated by the overlap of green and white samples).

Supplementary Fig 4: It is unclear whether the immunofluorescence staining in (d) and the Western blot in (f) were performed with the PAX4 or V5 antibody. Please provide this information as the PAX4 antibody may not recognize the truncated p.186X variant.

The PAX4 antibody (RnD, AF2614) is polyclonal and the binding epitope(s) of the antibody is proprietary information not revealed by the company. We have evaluated this antibody on cells overexpressing PAX4 and its variant proteins when doing preliminary antibody screening to identify one that works well. In the figure below, we have demonstrated using western blot that the AF2614 antibody is able to detect both the p.192His and p.186X proteins. As V5 is located at the 3' end of *PAX4* in the expression plasmid, p.186X introduced a premature stop codon that prevented the expression of V5. This is reflected in the missing band observed for V5 under the p.X186 lane (refer to the figure below). To clarify further, Supplementary Fig. 6f was blotted with PAX4 antibody instead, so that all the variant proteins can be captured. We thank the reviewer for identifying this point. The antibody information is now clearly annotated under the figure legend.

Image of western blot of WT PAX4 (WT), p.His192 and p.X186 proteins overexpressed in AD293 cells. PAX4 and V5 antibodies were used to assess protein expression 24 h post transfection.

Supplementary Fig. 6f: western blot using PAX4 RnD antibody.

Figure 7d: Please clarify the origin of each lane. For example, uncorrected refers to a line derived from the CRISPR correction that failed while p.His192His is the initial line used for CRISPR correction? The same goes for lanes 3 and 4. If so it may be worth emphasizing that CRISPR globally alters the transcriptome landscape. It also may be worth highlighting in bold genes which are mentioned in the text. Interestingly PDX1 was increased in the corrected version.

We apologize to the reviewer as our labeling on the original figure was unclear. We have now updated the figure to indicate that in panel d the first two columns in the donor derived BLCs include parental hiPSC line and a control line that went through the genome editing process but was unedited (both are labeled “Uncorrected p.His192His”). The second two columns are two independent hiPSC clones that were derived during the genome editing process and are corrected to be homozygous for the p.Arg192 allele (“Corrected p.Arg192Arg”). We have highlighted in bold the genes that were mentioned in the text.

Figure 7e: Columns 1, 3,5 and 8 lack labeling. The conclusion drawn by the authors is truly a leap of faith for the donor-derived BLC. In fact, the corrected version appears to have fewer beta cell-enriched factors as compared to the uncorrected version. Nonetheless, the isogenic data is clear-cut and conclusive.

We have now improved the labeling for Fig. 7e. There are two donor-derived hiPSC lines naturally carrying the p.Tyr186X variant (lanes 1 and 4) and two lines that underwent genome editing but remained unedited for the p.Tyr186X genotype (lanes 2 and 3) (“Uncorrected p.Tyr186X”). Lanes 5 to 8 refer to CRISPR-corrected donor-derived hiPSCs (“Corrected p.Tyr186Tyr”).

Comparing the “Uncorrected” columns composed of parental and CRISPR controls, we agree with the reviewer that CRISPR editing could potentially alter the transcriptome landscape. However, we do not have sufficient cell lines to support this statement. Hence, we prefer to stay conservative in making this statement. We appreciate the reviewer for recognizing the heterogeneous nature of hiPSCs even through the cells share an isogenic background. This further substantiates the need to have various models to complement the findings derived from hiPSC-based models.

Reviewer #3 (Remarks to the Author):

The authors report interesting data with attempts to illustrate functionality of PAX4 identified from GWAS. Cutting-edge technologies were applied including application of beta-like cells (BCLs) derived from patient iPSCs and precise genome-editing by CRISPR/Cas9. Such attempts are much desired for GWAS follow-up studies; however, the present study draws conclusions without sound causal evidence, and suffers from major methodological concerns.

We thank Reviewer #3 for recognizing the cutting-edge technologies deployed in our study and the need for GWAS follow up studies. Below we respond to their comments and provide new data to support our conclusions.

1. Lack of GSIS study in BCLs

This is a fundamental aspect when using BCLs derived from patient iPSCs. If the BCLs do not respond to glucose on insulin secretion, the authors cannot draw firm conclusion on the effect of any genetic manipulation (KD or genome editing) to have any causal effect on beta cell function. This aspect was completely ignored throughout the study, and there is no illustration on how BCLs were characterized as BCLs.

The reviewer is correct that most studies deploying BCLs derived from patient hiPSCs do not result in BCLs that respond to glucose. For this very reason, and the fact our gene of interest is involved in development, we have focused our evaluation on the effect of PAX4 perturbation on islet cell development. Based on our observations across multiple human beta cell models, we demonstrate a developmental defect that results in increased polyhormonal cells with decreased total insulin content.

In response to Reviewers #1 and 2, we have **now carefully characterized both the endocrine progenitors and BCLs differentiated from the donor-derived hiPSCs using Protocol B**. In brief, the additional characterization revealed no distinct differences between the control and PAX4-KO/p.Tyr186X/p.His192His-carrying lines in their efficiencies to differentiate towards definitive endoderm (SOX17⁺/CXCR4⁺), endocrine progenitors (INS⁺, NKX6.1⁺ and PDX1⁺), as well as beta-like (INS⁺ and NKX6.1⁺) cells. The new data are now included in **Supplementary Fig. 4**. The efficiencies of generating beta-like cells (30-40% - INS, PDX1), alpha cells (~10%, GCG), and delta cells (15-20%, SST) were highly comparable to Pagliuca et al., 2014 (Protocol B), and not significantly different in wild-type versus PAX4 variants (**Supplementary Fig. 4**). While the flow cytometry analyses have demonstrated no significant difference in the cell populations, our immunostaining data revealed elevated expression of polyhormonal INS⁺/GCG⁺ cells (Fig. 4j-m) which led us to uncover that these cells are less mature and expressed lower total insulin content (Fig. 4n). **The new data lend support to the main message of our study that PAX4 diabetes associated variants and complete loss-of-function (PAX4-KO) do not detrimentally impact the differentiation trajectory of hiPSCs towards pancreatic endocrine cells (Supplementary Fig. 3a-b) but impact the maturation of BCLs** as witnessed by increased polyhormonal expression and reduced total insulin content (Fig. 4).

We also note the reviewer's concerns on the lack of evaluation of GSIS in BCLs in our study. The functionality of hiPSC-derived BCLs is known to be variable and highly dependent on the hiPSC line. We have previously stimulated the hiPSC-derived BCLs with high glucose but unfortunately they were non-functional (static GSIS, data not shown). We understand the request to include functional data from hiPSC-derived BCLs. Therefore **we have now performed new experiments and re-visited the insulin secretion capabilities of the BCLs** from a selected set of cell lines (PAX4^{WT/WT} vs PAX4^{KO/KO}; uncorrected p.Tyr186X vs corrected p.Tyr186Tyr; and uncorrected p.His192His vs corrected p.Arg192Arg), this time using a perfusion system to evaluate the dynamic GSIS in response to glucose and KCl. **We have now incorporated the new data under Supplementary Fig. 11.**

Unfortunately, our hiPSC-derived BCLs were still not robustly functional under high glucose stimulation and only secreted insulin when treated with 30 mM KCl (Supplementary Fig. 11). As expected, **we also observed heterogeneity across the cell lines carrying the same genotype (Supplementary Fig. 11), making it challenging to draw any robust conclusions from the data**. We therefore included a complementary human beta cell model derived from human fetal pancreas – EndoC-βH1, to evaluate the effects of PAX4 perturbation on insulin secretion (Fig. 2). Importantly, in line with our observations in BCLs derived from hiPSCs, our EndoC-βH1 model also exhibited reduced maturity as demonstrated by reduced total insulin content which likely contributes to the impaired GSIS observed (Fig. 2e-f), recapitulating the clinical phenotype in subjects carrying the diabetes associated variants (Fig. 1a and 1d, lower insulin secreted in response to glucose). **This has now been acknowledged in the discussion section (line 523-526):**

“Our hiPSC-derived beta cells were not robustly functional under high glucose and secreted insulin only when treated with 30 mM potassium chloride (Supplementary Fig 11). To circumvent the

limitations of the hiPSC model in deriving functionally mature beta cells in vitro, we included the study of PAX4 in the human beta cell line EndoC-βH1.”

We would like to emphasize that the **major focus of our study here is the role of human PAX4 and diabetes associated variants in this gene on human pancreatic differentiation.** Thus far, we have demonstrated a developmental defect that resulted in increased polyhormonal cells with decreased total insulin content. That said, we do acknowledge the limitations and difficulty in obtaining robustly functional hiPSC-derived BLCs (that we tried) with Protocols A and B used in this study.

Supplementary Fig. 11 | Dynamic glucose-stimulated insulin secretion performed on beta-like cells. Donor hiPSC-derived beta-like cells carrying (a) uncorrected p.His192His (two lines); (b) corrected p.Arg192Arg (two lines); (c) uncorrected p.Tyr186X (one line); and (d) corrected p.Tyr186Tyr (one line) were stimulated at 2.8 mM (6 min), 16.7 mM (40 min), 2.8 mM (16 min) and 30 mM KCL (6 min) sequentially. Each graph represents data obtained from one hiPSC line.

2. PAX protein expression in all tissues is extremely low or absent (please check in various public databases), including in the pancreas. Have the authors taken this into consideration when designed the study?

We can confirm that we carefully considered the expression profile of PAX4 when designing our study. We were fortunate to have existing datasets for both RNA-seq and ATAC-seq from seven stages of pancreatic islet cell development from hiPSC, providing a rich resource to evaluate not only levels of *PAX4* expression during development but also chromatin activity at the locus. These data were key in our decision of which time points to focus on. *PAX4* expression peaks on D20 in the endocrine progenitor stage (Fig. 3c,e, EP stage) and decreases towards the BLC stage, recapitulating the expression pattern seen in rodents (Sosa-Pineda *et al.*, 1997). The expression of *PAX4* in mature human islets and in our cell model (EndoC- β H1) is indeed low. Others have reported that *Pax4* remains detectable in adult pancreatic beta cells (Brun *et al.*, 2004) and that *PAX4* expression in EndoC- β H1 is lower than human islets, but detectable (Tsonkova *et al.*, 2018). In our EndoC- β H1 models, the Ct values of *PAX4* via qPCR are approximately in the 26-32 range and vary across passages.

Taking reference from the rodent models, hiPSCs are an ideal model to evaluate *PAX4* biology during human pancreatic beta cell development *in vitro*, given that access to embryonic/fetal endocrine tissues is limited and genetic manipulation of the tissue not possible. Here, we have demonstrated using hiPSC-derived BLCs that loss of *PAX4* had no significant impact on the differentiation trajectory towards endocrine cells but played a role in driving BLCs towards their maturation. Yet, the major challenge of our study remains the derivation of fully functional cells that can be robustly evaluated for defects in GSIS. To circumvent this shortcoming, we made use of EndoC- β H1 cells to evaluate the effect of *PAX4* LOF on insulin secretion. When *PAX4* is downregulated (sh*PAX4*), EndoC- β H1 cells also exhibited reduced maturity as demonstrated by reduced total insulin content which likely contributes to the impaired GSIS (Fig. 2e-f), replicating the clinical phenotype in variant carriers (Fig. 1a,d, lower insulin secreted in response to glucose). These findings support the key aim of our study to evaluate how *PAX4* LOF can impact human beta cell development and in turn increase T2D risk.

3. Page 8 line 172, reduction of HOMA-B was no longer significant after adjusted for age, gender and BMI? What about only BMI, or age, or gender, or any combination? How would the authors interpret the data?

We thank the reviewer for bringing this to our attention. When revisiting the data, we discovered that they were not normally distributed and consequently they should have been log transformed. We have redone our analysis and HOMA-B now remains significant after adjustment for age, sex and BMI.

b	p.Arg192Arg controls n=121 (95% CI)	p.His192 allele(s) carriers (n=62)	p-adj
AIRg*	548.3 (476.3-631.3)	424.6 (349.4-516.0)	0.036
Si*	2.28 (2.09-2.48)	2.64 (2.34-2.98)	0.048
DI*	1247.4 (1078.3-1444.0)	1122.0 (916.9-1372.3)	0.396
HOMA-B*	147.2 (135.0-160.4)	124.7 (110.8-140.6)	0.027

* AIRg, Si, Disposition index and HOMA-B values obtained by back-transforming Lg10 values of the model-estimated means for controls and carriers.

“Carriers of the T2D-risk allele (p.His192) had a decreased acute insulin response to glucose (AIRg, padj=0.036) after adjusting for age, sex and BMI (Fig. 1b). Carriers of the T2D-risk allele p.His192 were more insulin sensitive (Si, p=0.048). There were no differences in disposition index (DI, p=0.396) between the two groups (Fig. 1b). HOMA-B, a measurement of beta cell function, was significantly reduced in p.His192 allele(s) carriers after adjusting for age, sex and BMI (padj=0.027).”

4. In Fig 2, gene expression data was presented as “relative to siNT/scramble”. As detailed in the “Methods” session in Page 41, only 1 housekeeping gene ACTIN was used in gene expression experiments. At least 2 housekeeping genes should be used, and what are the Ct values of PAX4? According to public databases, in general PAX4 gene expression is very low in all tissues.

We thank the reviewer for highlighting this, and we agree that additional housekeeping genes should be tested. **We have now repeated qPCR to include additional housekeeping genes (GAPDH and TUBB3) and the data is displayed below.** In brief, there is no significant difference within the expression of the three housekeeping genes between shScrambled and shPAX4 EndoC-βH1 cells. When normalized to each of the housekeeping gene, PAX4 expression in the shPAX4 EndoC-βH1 cells was consistently lower than that of shScrambled cells.

5. CRISPR guides: what is the deletion? Between the 2 guides? There is no description of the confirmation of deletion (e.g. by Sanger). The authors did not address off target effects either.

We apologize that this information was not adequately highlighted in the manuscript. We have added the following additional information in the results and methods section.

“We designed our loss-of-function strategy to mirror the well-studied Pax4^{-/-} mice where almost all of the functional domains were replaced with a beta galactosidase-neomycin resistance cassette (stated in the methods section). To delete the majority of the paired and homeodomains, sgRNAs were designed targeting exon 2 and exon 5 of PAX4 gene (ENST00000341640.6). Successfully edited clones were detected using genotyping PCR and primers flanking the exon 2 through 5 deletion. Sanger sequencing was used to confirm that two of the independent cell lines had a homozygous deletion for amino acids 64 through 200, whilst the other cell line was compound heterozygous for two premature stop codons at amino acids 61 and 74, respectively.”

With respect to the potential off target effects, using Off-Spotter (Pliatsika and Rigoutsos, 2015), we did not detect any genomic sequences with 1 or 2 mismatches from either gRNAs. In addition, none of the predicted off-targets (up to 5 mismatches) are in genes relevant for islet cell biology.

6. PAX4 Arg192His variant (chr7:127253550, rs2233580) is located in Exon 5, the location of Tyr186X is also in Exon 5. CRISPR sgRNA#2 cutting site is around R192, A191 and V190I (theoretically 3-4 bps before or after the PAM seq). What is exactly the deleted region, does it cover the two SNP loci? (Line 259: deletion for AA 64-200?) What is the rationale to design a big deletion fragment in relation to the two GWAS loci?

We previously tried to generate a PAX4 knockout hiPSC line using a single gRNA to generate indels and a premature stop codon. Unfortunately, this strategy was unsuccessful in our hands as the deletion generated an in-frame mutation and resulted in the continued production of a PAX4 protein. Therefore, we decided to adjust our strategy to mimic that of the well-studied Pax4 null mouse model, which deletes a large region of the functional domains of the protein (exons 2 through 5). Two of our lines deleted amino acids 64-200 (so would delete both variants), while the third line introduced premature stop codons that would be predicted to undergo nonsense-mediated decay.

7. It would be better to use PAX4 wt instead of PAX4^{+/+}, which may be confused with insertion mutation.

We thank Reviewer #3 for this suggestion. We have edited the manuscript accordingly using PAX4^{WT/WT} for wild-type cells, and PAX4^{KO/KO} for knockout cells.

8. Page 13, why did the authors choose protocol B over A, and applied protocol B for the rest of the study? PAX4 expression was absent in BLC by protocol B.

This study is a collaboration between two labs that both perform *in vitro* stem cell differentiation using two different protocols. The Gloyne lab uses protocol A (Rezania *et al.*, 2014), and the Teo lab uses protocol B (Pagliuca *et al.*, 2014). At the initiation of the collaboration, both groups were keen to use their own protocols and saw enormous value in comparing their data. We compared the two differentiation protocols, showing that PAX4 expression peaks at the endocrine progenitor stage (Figs. 3c,e). Also, as illustrated in Supplementary Fig. 3 (a bioinformatic comparison of RNA-seq data from Protocols A and B), the two protocols generated highly similar expression profiles over the course of differentiation. As the Teo laboratory had access to the donors, generated the donor-derived hiPSCs, and was using Protocol B, we subsequently decided to combine efforts and perform all experiments with Protocol B after initial comparison experiments. We thank the reviewer for pointing out the lack of clarity. We have now indicated the protocol used for each experiment clearly.

Using hiPSCs as a model for beta cell development, we have demonstrated that PAX4 expression peaks at D20 endocrine progenitor stage (Fig. 3c,e, EP stage) and decreases towards BLC stage. These observations are consistent with the studies reported in rodents whereby rodent Pax4 begins to be expressed at E9.5, peaks between E13.5 and E15.5 to drive endocrine progenitors towards the

insulin-producing beta cell fate before its expression decreases towards maturation (Sosa-Pineda *et al.*, 1997). In fact, shortly after birth there are few Pax4-expressing cells in the mouse (Wang *et al.*, 2004). Therefore, it is an expected observation in which PAX4 expression is lower but not absent in BLCs.

Volcano plots in panel d and f showed completely different gene profiles generated from protocol A and B, except for DENND2D, any follow-up study on DENND2D? In panel f, PAX4 is again absent, not among the top genes that were affected by PAX kD, which makes it further questionable the choice of protocol B.

We thank the reviewer for pointing this out and apologize for the confusion which arises from how our data are presented. It is important to clarify that labelled genes in the volcano plots are the top ten genes that are differentially expressed genes (by p-value) (line 1363-1368). Since PAX4 expression is low in mature beta cells (Sosa-Pineda *et al.*, 1997), it is not unexpected that PAX4 does not appear among one of the most differentially expressed genes.

DENND2D is also a differentially expressed gene in the hiPSCs (Protocol B) and definitive endoderm (Protocol A). As DENND2D expression is altered before activation of PAX4, we hypothesize that the decreased expression of DENND2D is independent of PAX4. Notably, it is not predicted to be an off-target of the gRNAs.

9. Page 14, line 290-292, this is an overstatement: "PAX4 is not required for human beta cell differentiation in vitro. Rather, PAX4 loss-of-function results in derepression of alpha cell genes and a dysregulation of key endocrine maturation genes in hiPSC-derived BLCs." As stated on page 13 Line 283, nonsignificant.

We thank the reviewer for this point. We have toned down on the language and amended it accordingly in the revised manuscript.

"These observations suggest that, unlike in mouse, PAX4 is not essential for human beta cell differentiation in vitro and its loss did not detrimentally impact the differentiation trajectory of hiPSCs towards pancreatic endocrine cells. Rather, PAX4 loss-of-function results in a dysregulation of key endocrine maturation genes in hiPSC-derived BLCs."

10. What is the ethnic origin of the hiPSC lines used in Fig 3? Furthermore, the PAX4 variants were identified in Asian populations, and EndoC cell line is of French origin. (<https://www.ncbi.nlm.nih.gov/pmc/articles/PMC3163974/>). Indeed, many human derived cell lines do carry ethnic-specific signatures. Any thoughts?

The reviewer raises an interesting point regarding the genetic background of cell lines used in research. In our study we have been able to capitalize on numerous resources providing the opportunity to evaluate the role of PAX4 in cell lines of different ancestries showing a common cellular phenotype across them. The diabetes associated variants have been found in the Singapore population and we have modeled them both in patients and in an isogenic European hiPSC model and the immortalized fetal derived EndoC-βH1 cell line with consistent observations. The use of different models is a strength of our study.

11. What is the consequence of increased GCG gene expression in translation? In protein expression detection? Fig 4 IF stained for GCG. The mammalian proglucagon gene (*Gcg*) encodes three glucagon like sequences, glucagon, glucagon-like peptide-1 (GLP-1), and glucagon-like peptide-2. Have the authors looked at other potential translational products?

We thank the reviewer for raising this point. The loss of beta cell identity and the acquisition of polyhormonal cells have been reported in the pancreatic islets of individuals with diabetes (Cinti *et al.*, 2016; Talchai *et al.*, 2012). The transdifferentiation of metabolically stressed beta cells to express GCG has been proposed by several groups as a mechanism underlying beta cell failure in T2D (Brereton *et al.*, 2014; Talchai *et al.*, 2012; Wang *et al.*, 2014). The GCG antibody was carefully selected for the identification of alpha cells (Santa cruz, sc-7780, used by Kroon *et al.*, 2008; Ackermann *et al.*, 2018; Dooley *et al.*, 2016). GLP-1 and GLP-2 are predominantly secreted by the enteroendocrine cells (Iakoubov *et al.*, 2007). Therefore, we have not considered looking into the

expression of GLP-1 and GLP-2 in our hiPSC-derived BLCs. However, in response to the reviewer's concern, we did measure the GLP-1 levels during an OGTT in carriers of the p.Arg192His variant (Supplementary Fig. 1f-h, see below). There was no significant difference between the carriers and non-carriers of the variant.

Clinical assessment of GLP-1 in carriers of p.Arg192His PAX4 variant during two-hour OGTT. (F) Fasting, (G) 20-min and (H) AUC of GLP-1 secreted in subjects.

12. Line 320-322, this is an overstatement. In his192 allele carriers, no change in PAX4 or INS, GCG, SST transcription was detected in this study. Line 322, “negatively affecting endocrine cell differentiation”, this is a bold conclusion based on only GCG staining, without further data to back up endocrine cell differentiation.

We thank the reviewer for highlighting this. With new flow cytometry characterization data that we have incorporated as Supplementary Fig. 2 and 3, we have revised our manuscript as follows:

“Heterozygous and homozygous carriers of the PAX4 p.His192 allele had no measurable differences in INS, GCG, or SST gene expression (Fig. 4b-i).” There were no significant differences in the cell populations expressing INS, GCG, or SST at the EP or BLC stages (new Supplementary on FACS characterization). *“Together, these data suggest that both PAX4 alleles result in a loss-of-function due to reduced PAX4 gene dosage and/or altered PAX4 transcriptional activity, negatively affecting pancreatic beta cell differentiation.”*

13. PAX4 primers: hPax4F5 (AGGACACGGTGAGGGTCTGGT) shows no blat result (please correct me if I was wrong); hPax4R5 (CAGTGGTTCAGGGCAGGCA) is located in Exon 8. Tyf186X is supposed to introduce premature stop codon at AA186 (line 333-334). How is the amplification region in relation to the variants?

The primers were designed against the transcript (not the genome) and the hPax4F5 spans the exon 5/6 boundary. The primer pair amplifies a 191-bp sequence that is 3' to both variants (see image below).

Homo sapiens paired box 4 (PAX4), transcript variant 1, mRNA

>NM_001366110.1:407-1462 Homo sapiens paired box 4 (PAX4), transcript variant 1, mRNA

```
ATGCATCAGGACGGGATCAGCAGCATGAACCAGCTTGGGGGGCTCTTTGTGAATGGCCGGCCCCTGCCTCTGGATAC
CCGGCAGCAGATTGTGCGGCTAGCAGTCAGTGAATGCGGCCCTGTGACATCTCACGGATCCTTAAGGTATCTAATG
GCTGTGTGAGCAAGATCCTAGGGCGTTACTACCCACAGGTGTCTTGAGCCAAAGGGCATTGGGGGAAGCAAGCC
ACGGCTGGCTACACCCCTGTGGTGGCTCGAATTGCCAGCTGAAGGGTGAGTGTCCAGCCCTCTTTGCCTGGGAAA
TCCAACGCCAGCTTTGTGCTGAAGGGCTTTGCACCCAGGACAAGACTCCAGTGTCTCCTCCATCAACCGAGTCCTGC
GGGCATTACAGGAGGACCAGGGACTACCGTGCACACGGCTCAGGTCACCAGCTGTTTTGGCTCCAGCTGTCCTACT
CCCCATAGTGGCTCTGAGACTCCCCGGGTACCCACCCAGGGACCGGCCACCGGAATCGGACTATCTTCTCCCAAG
CCAAGCAGAGGCACTGGAGAAAGAGTTCAGCGTGGGCAGTAT(GTA)CCTGATTAGTGGCCCGTGGAAAAGCTGGC
TACTGCCACCTCTCTGCTGAGGACACGGTGAGGGTCTGGTTTTCCAACAAGAAGAGCCAAATGGCGTCGGCAAGAGA
AGCTCAAGTGGGAAATGCAGCTGCCAGGTGCTTCCAGGGGCTGACTGTACCAAGGTTGCCCCAGGAATCATCTCT
GCACAGCAGTCCCCTGGCAGTGTGCCACAGCAGCCCTGCCTGCCCTGGAACTACTGGTCCCTCTGCTATCAGCTG
TGCTGGGCAACAGCACCAGAAAGGTGTCTGAGTGACACCCACCTAAAGCCTGTCTCAAGCCCTGCTGGGGCCACTT
GCCCCACAGCCGAATTCCTGGACTCAGGACTGCTTTCCTTCCCTTCCCTCCACTGTACCTGGCCAGTCTT
AGTGGCTCTCAGGCCCTGCTCTGGCCTGGCTGCCACTACTGTATGGCTTGGGAATGA
```

hPax4F5 primer

hPax4R5 primer

14. What is the epitope of PAX4 antibody?

The PAX4 antibody (RnD, AF2614) is polyclonal. Unfortunately, the binding epitope(s) of the antibody is proprietary information not revealed by RnD. During our evaluation of the specificity of the antibody we established that the AD2614 antibody is able to detect both the p.H192 and p.X186 recombinant proteins. As a V5-tag is located at the 3' end of PAX4 in the expression plasmid, p.186X introduced a premature stop codon that prevented the expression of V5. This is reflected in the missing band observed for V5 under the p.X186 lane (refer to the figure below). This would support the location of the epitope being in the N-terminus of the protein.

Image of western blot of WT PAX4 (WT), p.His192 and p.X186 proteins overexpressed in AD293 cells. PAX4 and V5 antibodies were used to assess protein expression 24 h post transfection.

15. ChIP should be performed to demonstrate interaction between PAX4 and gene promoter regions.

We would like to perform ChIP for PAX4 but unfortunately our evaluation of the specificity of the currently available PAX4 antibodies has shown that they are not suitable for ChIP. Hence, we respectfully respond that this is currently out of scope for this manuscript. We have sought to demonstrate an interaction between PAX4 and gene promoter regions through the use of luciferase assays, which are more sensitive in the detection of transcriptional regulatory activity.

16. Fig 5h shows that PAX4 kd had no effect on INS promoter activity, how to explain Fig 2g where it shows increased INS gene expression by PAX kd?

We thank the reviewer for highlighting this. PAX4 is not the only transcription factor that regulates the expression of the *INS* gene. We speculate that the non-significant increase in *INS* expression in some samples could be a compensatory effect by other beta cell determinant transcription factors in an attempt to re-navigate the cells towards beta cell fate - especially in earlier passages following *PAX4* knockdown. To determine if sh*PAX4* does impact the expression of *INS* gene in EndoC-βH1 cells, we performed additional experiments (now with stable sh*PAX4* cells of older passages) and presented the updated data as Fig. 2g. These data remain consistent with our previous finding that *INS* expression is not significantly different following sh*PAX4*-mediated knockdown (updated Fig. 2g).

In Fig. 5d, overexpression of WT *PAX4* repressed *INS* promoter in EndoC-βH1 cells. When *PAX4* was knocked down, there was no significant difference in luciferase activity at the *INS* promoter (Fig. 5g).

We speculate that the inconsistency in Fig. 5d and Fig. 2g is a consequence of different *PAX4* dosages within the EndoC-βH1 cells. In Fig. 5d, the exogenous overexpression of *PAX4* protein elicited a strong repression effect. On the other hand, in Fig. 2g, knocking down endogenous *PAX4* using shRNA results in no significant difference in *INS* gene expression.

17. Fig 6e and g, there is no significant difference between the variants. Should be carefully interpreted in data presentation.

We thank the reviewer for highlighting this. The difference in the overall glycolytic function/mitochondrial respiration is not significant when comparing the endocrine progenitor cells carrying wild-type or variant *PAX4* allele(s) (Fig. 6e,g). However, when we measure specific parameters of the glycolysis and mitochondrial stress tests, we did observe significant differences between *PAX4* genotypes. In particular, there was significant reduction in non-glycolytic acidification, glycolytic capacity, and glycolytic reserve (Fig. 6f), consistent with *PAX4* variants reducing key parameters of glycolytic flux. With respect to mitochondrial measurements, *PAX4* variants had increased basal respiration, non-mitochondrial O₂ consumption, ATP production, and H⁺ (proton) leak (Fig. 6h), consistent with a bioenergetic switch from glycolysis to oxidative phosphorylation. We have updated the results section to make it clear that the differences in glycolysis and mitochondrial stress tests were only found in the aforementioned key parameters (line 407-419):

“To further assess a potential defect in metabolism, we performed a Seahorse XFe96 Glycolysis Stress Test. The glycolysis stress test showed that EPs carrying one or two p.His192 risk alleles had lower measurements of glycolytic function, including non-glycolytic acidification, glycolytic capacity, and glycolytic reserve (Fig. 6e-f). In addition, EPs carrying the p.Tyr186X risk allele had decreased glycolytic reserve and non-glycolytic acidification (Fig. 6f). Next, we hypothesized that EPs would seek alternative metabolic processes to compensate for the reduction in energy production, such as oxidative phosphorylation through mitochondrial respiration. Mitochondrial function was measured via oxygen consumption in EPs using the Seahorse XFe96 analyzer. There was an increase in oxidative phosphorylation activity in EPs harboring PAX4 variants (Fig. 6g), including basal respiration, non-mitochondrial O₂ consumption, ATP production, and H⁺ (proton) leak (Fig. 6h). Overall, EPs carrying PAX4 diabetes risk alleles demonstrated a bioenergetic switch from glycolysis to oxidative phosphorylation.”

18. Line 411, these changes are not significant in Supplementary Fig6

We thank the reviewer for this point. In our RNA-seq data (Fig. 6d and Supplementary Fig. 6), we observed endocrine cells carrying PAX4 variants had altered metabolic signatures, suggesting that the cells were under elevated oxidative stress (Fig. 6d). This prompted us to look into the metabolic functions of these cells to evaluate if the metabolic stress could be one causal factor accounting for the reduced total insulin content in PAX4 variant carrying-BLCs (Fig. 4n). However, when we treated the cells (from EP to BLC) with the antioxidant NAC, there was no rescue in the total insulin content as highlighted by the reviewer (Supplementary Fig. 6), suggesting that the unique metabolic signature is not causal. Nonetheless, our data suggest that EPs carrying PAX4 variant allele(s) had a bioenergetic switch from glycolysis to oxidative phosphorylation (Fig. 6f,h). Therefore, we speculated (under discussion, line 426-429) that *“metabolic signature observed in our donor-derived hiPSC model reflects the physiological status of the EPs rather than being the immediate cause for the dysregulation of beta cell development and maturation”*. We hope this clarifies.

19. Fig 7f and g, Supplementary Fig 7, wt is missing in all experiments for control

We thank the reviewer for pointing this out. Regarding rescue experiments, we did not include the wild-type cell lines in our experiments. hiPSCs are highly heterogeneous in nature even across cell lines from the same donor. It is also noted that, from our experience with handling numerous donor-derived hiPSC lines, heterogeneity could be introduced during the differentiation process even within the same cell line. Hence, our experiments (Fig. 7c) were **carefully designed to make use of the gene-corrected lines (wild-type) versus uncorrected isogenic pairings for data comparison**. This will not change the overall message but tremendously helped in the reduction of background noise from genetic differences and from the CRISPR genome editing process (i.e. clonal expansion and increased passages). **We have now incorporated the explanation into the manuscript for better clarity** (line 433-446):

“Next, we used CRISPR-Cas9 to correct the donor-derived hiPSC lines and to generate PAX4 variant isogenic hiPSC lines. We designed sgRNA#3 to target the donor-derived homozygous p.His192His line and provided the homology-directed repair (HDR) template to correct the rs2233580 T2D-risk allele (Fig. 7a). The II-11 donor-derived hiPSC line that is heterozygous for a GTA duplication was corrected with sgRNA#4 and an HDR template (Fig. 7b). From the CRISPR-Cas9 genome editing pipeline, we generated two corrected p.Arg192Arg non-risk and two uncorrected p.His192His hiPSC lines (Fig. 7c). From the II-11 donor-derived line, four corrected p.Tyr186Tyr and four uncorrected p.Tyr186X hiPSC lines were derived (Fig. 7c). To control for the different genetic backgrounds of donors, we generated isogenic SB Ad3.1 hiPSCs that are homozygous for the risk alleles (p.His192His and p.X186X). All the corrected and uncorrected donor derived hiPSC lines and the isogenic PAX4 variant hiPSC lines were differentiated towards BLCs using Protocol B, followed by RNA-seq analyses and assessment of total insulin content (Fig. 7c).”

Point-by-point response (3rd response letter)

Reviewer #1 (Remarks to the Author):

Even if the authors have done their best to respond to my comments, I unfortunately still remain confused by some of the data. Moreover, at this reading I identified some new points that I think should be corrected.

1. The authors use the abbreviation BLC (beta like cells) throughout. However, this is not a good term, in fact it is misleading since only <50% of the cells represent beta (like) cells. The term SC-islet (stem cell derived islet) is much more appropriate, and it has been increasingly used by other groups. Why not aim for unified nomenclature?

We thank the reviewer for the comment and agree that SC-islet is indeed more appropriate. We have now edited our manuscript and figures accordingly.

2. In my original comment #3, I made the simple point that since the mutant iPSC lines (both position 192 and 186) had lower insulin content and increased proportion of INS/GCG cells, one would expect to see this also in the PAX4 KO cells, but the data were not shown. This is a simple phenotype and if it is associated with PAX4 LOF, then it should be evident also in the KO cells. In the rebuttal, the authors now show that INS content is not reduced in the KO, and they do not show the KO effect on INS/GCG double positivity, even though this is what I requested.

We apologize for misunderstanding your original point as it is an important one. We have now performed additional experiments to quantify the number of CPEP/GCG double positive cells in PAX4 KO lines.

Sham control SC-islet cells (left panel) had significantly lower differentiation propensity than PAX4-KO (right panel). White arrow pointing to cell(s) co-expressing CPEP and GCG. Scale bar: 50 μ m.

While it appears that there is a robust increase in the number of co-expressing cells in the PAX4 KO lines, we have decided to not include the data in the manuscript for the following reasons:

- 1) Using protocol B, we routinely measure >20% C-PEP+ cells in SC-islets (Figure 4I).
- 2) For this particular differentiation we detected very few C-PEP+ cells in the PAX WT cells (sham panel on the left), suggesting poor differentiation of these PAX WT cells towards SC-islets. We had previously measured ~20% C-PEP+ cells in SC-islets derived from PAX4 WT cells (Supplementary Fig. 4c).
- 3) It is therefore not possible to conclude with current data that the increase in CPEP/GCG double positive cells is due exclusively to the KO genotype and not reflective of a poor differentiation of the WT cells to form SC-islets.

While our attempt to address the important question of whether PAX4 KO cells have increased C-PEP/GCG double positive cells was unsuccessful, we hope the reviewer appreciates our efforts and agrees with our decision to not include data that may not accurately reflect the biology.

If the mutations lead to loss of function, it is difficult to understand why the mutations would lead to a reduced insulin content but the KO not. The authors explain that “due to the heterogeneous nature of hiPSC models, we were not able to detect any significant difference when comparing the total insulin content between the control and PAX4 KO lines”. I do not find this explanation acceptable. The comparison should be done between the parental line and the isogenic KO line. The “heterogeneous nature” of the lines should then not be a problem. Overall, it appears that the authors try to hide these discrepancies between the KO phenotype and the mutation phenotype. Perhaps it would be better to openly discuss the problems and limitations of the models.

We apologize for our unsatisfactory answer to your original question and we certainly did not intend to hide any discrepancies in our data. As we have discussed in our manuscript, the in vitro differentiation model, while a powerful one, does suffer from some challenges, including variability, heterogeneity, and immaturity. We have tried to overcome these limitations by complementing our hiPSC models with clinical data and assays in an alternative model, EndoC- β H1 cells. Despite this, as the reviewer noted, there is a discrepancy between the KO phenotype and the variants. There are several possible explanations for this:

- 1) The reduced insulin content in the donor-derived cells is independent of the PAX4 genotype. This is unlikely to be true as the content phenotype was reversed by gene correction (Fig. 7).
- 2) There could be genetic compensation that “rescues” the insulin content phenotype in the homozygous null line, as has been shown in zebrafish and mouse models [PMID: 3094447]. This is a possibility that we cannot rule out; although, you would expect to only see this in the compound heterozygous line with premature stop codons.
- 3) The SB cell lines (and therefore the KO model) are prone to more variability using Protocol B than the donor-derived cells. As was illustrated above, we did have more difficulty getting consistency among our differentiations from the SB Ad3.1 cell lines using Protocol B. If we were to approach this project again, we would have spent some more time optimizing the protocol to improve the consistency.

Despite the inconsistencies between the KO and variants, we are confident that the variants are loss-of-function because a) knockdown of *PAX4* in EndoC-BH1 cells reduces insulin content, b) we determined that the molecular consequence of the *PAX4* variants is reduced function (Fig. 5 luciferase assays), and c) we can correct the content phenotype in donor-derived cells through *PAX4* gene correction.

3. I found it very difficult to get a clear message from Fig 7 which deals with the mutation corrected (or engineered) iPSC lines. Even after the detailed rebuttal, several problems remain:

- no data is presented to verify that the genome editing was actually successful. It is common practice to show Sanger sequencing of the corrected/introduced healthy/mutant alleles. Similar essential evidence should be provided for all genome edited lines used in these studies.

We have now added Sanger sequencing to Fig. 3b (PAX4 KO hiPSC lines) and Fig. 7a-b (Correction of patient-derived lines).

Fig. 3b: Sanger sequencing for exon 2 and exon 5 of compound heterozygous *PAX4*^{KO/KO} and homozygous deletion *PAX4*^{KO/KO} hiPSC lines.

Fig. 7a-b: (a-b) CRISPR-Cas9 gene editing strategy to correct the (a) p.His192His genotype to p.Arg192Arg (rs2233580) and (b) p.Tyr186X genotype to p.Tyr186Tyr (GTA duplication). Protospacer adjacent motif (PAM) sequence is bolded, and the respective variant and duplication are labelled in green. Representative Sanger sequencing results for the corrected allele are depicted with nucleotide changes labelled in red.

- the labelling in panels d-e remains confusing. The donor-derived mutant and corrected lines are of course also isogenic, not only the standard iPSCs with engineered mutations. It would be logical to label them as “donor-derived” and “engineered”

Thank you for the suggestion. We updated the figure and manuscript accordingly.

- It is virtually impossible to grasp any meaningful results from the long list of genes in the heatmaps in panels d-e. There are many examples of lower and higher beta-cell gene expression in the mutant as compared to corrected. The presentation should be improved to provide a clear message. If no clear message exists, then it would be better to not show these results at all.

We thank the reviewer for their suggestion to consider removing the heatmaps in Fig. 7. By removing them, we were able to include the requested Sanger sequencing data confirming the CRISPR-Cas9 correction. The RNA-seq and DEG lists can still be found in Tables S4, S5, and S6 and the data will be publicly available. The important message, which the updated figure better highlights, is that CRISPR-Cas9 correction improved total insulin content in *PAX4* patient-derived SC-islets.

Reviewer #2 (Remarks to the Author):

The authors have appropriately addressed all concerns raised by this reviewer. The study will be of great interest to the readership of Nature Communication. Benoit Gauthier

Thank you for your helpful feedback.

Reviewer #3 (Remarks to the Author):

The authors answered the questions well with additional experiments and convincing data.

One question remains on "5. CRISPR guides: what is the deletion? Between the 2 guides? There is no description of the confirmation of deletion (e.g. by Sanger). The authors did not address off target effects either. "

We have now added the Sanger sequencing data to Fig. 3b and updated the text as follows: "Using Sanger sequencing, we confirmed that two of the independent cell lines had a homozygous deletion for amino acids 64 through 200, whilst the other cell line was compound heterozygous for two premature stop codons at amino acids 61 and 74, respectively (Fig. 3b)."

The authors answered "With respect to the potential off target effects, using Off-Spotter (Pliatsika and Rigoutsos, 2015), we did not detect any genomic sequences with 1 or 2 mismatches from either gRNAs. In addition, none of the predicted off-targets (up to 5 mismatches) are in genes relevant for islet cell biology. "

It is not enough to assume that the off-targets are not relevant because they are not relevant for islet cell biology. Sanger seq on top predicted off-target regions should be performed.

Thank you for the suggestion. Using the results from Off-Spotter, we designed primers and sequenced the top predicted off-target sites for each gRNA and their respective hiPSC lines. We have added the following sentence to the methods section:

"Primers were designed to amplify and Sanger sequence the top predicted off-target sites for each of the four gRNAs. No mutations were detected in any of the sequenced potential off-targets sites."

We have included the Sanger sequencing results that support this statement below:

- 1) Exon 2 gRNA: CTAGGGCGTTACTACCGCAC [+ strand]
 - a. PSD4 [4 mismatches] CCAAGGCGTTGCCACCGCAC
 - i. No SNPs detected for three WT and three KO cell lines.

- 2) Exon 5 gRNA: TATCCTGATTGAGTGGCCCG

- a. PSM2 [4 mismatches] TA**GT**TG**A**CTCAGTGG**C**CCG
 - i. No SNPs detected for three WT and three KO cell lines.

- ii. No SNPs detected for two p.Arg192His cell lines.

- iii. No SNPs detected for one p.Tyr186X cell line.

- b. ZNF704 [4 mismatches] TAT**G**CT**G**CT**G**CTGTGG**C**CCG
 - i. No SNPs detected for three WT and three KO cell lines.

ii. No SNPs detected for two p.Arg192His cell lines.

iii. No SNPs detected for on p.Tyr186X cell line.

c. PANK4 [4 mismatches] TATCCTGGCTGGGTGGCCCG

i. No SNPs detected for three WT and three KO cell lines.

ii. No SNPs detected for two p.Arg192His cell lines.

iii. No SNPs detected for on p.Tyr186X cell line.

3) sgRNA91: ATCTCCGAGAGTTCCAGCG

a. RBPJL [4 mismatches] CTCTCCGTGGAGCTCCAGCG

i. No SNPs detected for patient-derived p.Tyr186X (two corrected, two uncorrected, and the parental) cell lines.

4) sgRNA59: GGCAGTAGCCAGCTTTCCAT

a. CHUK [4 mismatches] AACTGTAGCCAGTTTCCAT

i. There is one SNP downstream of the potential off-target site. The edited lines are heterozygous (three corrected, two uncorrected, and the parental), as is the parental, unedited patient-derived cell line. Therefore, this SNP did not result from the genome editing process.

- b. GRM7 [4 mismatches] AGCAGAAGTCAGTTTTCCAT
 - i. No SNPs detected for patient-derived p.Arg192His (three corrected and two uncorrected) cell lines.

Point-by-point response (final response letter)

Reviewer #1 (Remarks to the Author):

I want to thank the authors for their open and frank response to my remaining concerns. It is evident that there are certain problems in the SC-islet differentiation. Consequently, it was a wise decision to leave out some data coming from experiments that did not work efficiently. However, overall there is sufficient evidence in this paper to justify the link between decreased Pax4 expression and lower beta-cell functionality which is consistent with the association of Pax4 LOF variants with the risk of diabetes. I do not have any further requests.

We thank Reviewer #1 for acknowledging the key challenges with SC-islet differentiation.

Reviewer #3 (Remarks to the Author):

No further comments, the reviewer thanks the authors for their efforts.

Thank you.